# Oh SnapMMD! Forecasting Stochastic Dynamics Beyond the Schrödinger Bridge's End

## Abstract

Scientists often want to make predictions beyond the observed time horizon of "snapshot" data following latent stochastic dynamics. For example, in time course single-cell mRNA profiling, scientists have access to cellular transcriptional state measurements (snapshots) from different biological replicates at different time points, but they cannot access the trajectory of any one cell because measurement destroys the cell. Researchers want to forecast (e.g.) differentiation outcomes from early state measurements of stem cells. Recent Schrödinger-bridge (SB) methods are natural for interpolating between snapshots. But past SB papers have not addressed forecasting. Some natural immediate extensions of existing methods would (1) reduce to following pre-set reference dynamics (chosen without seeing data) or (2) require the user to choose a fixed, state-independent volatility since they minimize a Kullback–Leibler divergence. Either case can lead to poor forecasting quality. In the present work, we propose a new framework, SnapMMD, that learns dynamics by directly fitting the joint distribution of both state measurements and observation time with a maximum mean discrepancy (MMD) loss. Unlike past work, our method allows us to infer unknown and state-dependent volatilities from the observed data. We show in a variety of real and synthetic experiments that our method delivers accurate forecasts. Moreover, our approach allows us to learn in the presence of incomplete state measurements and yields an $R^2$-style statistic that diagnoses fit. We also find that our method's performance at interpolation (and general velocity-field reconstruction) is at least as good as (and often better than) state-of-the-art in almost all of our experiments.

## 1 Introduction

Many scientific modeling problems require forecasting stochastic dynamics from snapshot data. Here, snapshot data represent observations taken at different time points, without access to individual trajectories. And forecasting involves predicting future states beyond the observed times. For example, single-cell RNA sequencing (scRNA-seq) is widely used to study dynamic processes such as development, immune activation, and cancer progression. A scRNA-seq measurement destroys the cell, so scientists observe independent biological replicates at discrete times rather than a single replicate across multiple times. Despite the absence of individual cell trajectories, researchers often aim to forecast future cellular states; for instance, researchers are interested in forecasting differentiation outcomes of stem cells, immune cell activity after initial signal stimulation, or long-term cancer cell response to drugs with transcriptomic snapshots taken shortly after treatments. A common additional challenge is incomplete state measurement. For instance, although protein expression level mediates the dynamics of gene expression, protein level cannot be measured by scRNA-seq.

Recent work has addressed the problem of interpolating between snapshots through Schrödinger bridge (SB) methods (Pavon et al., 2021; De Bortoli et al., 2021; Vargas et al., 2021; Koshizuka and Sato, 2022; Wang et al., 2024) and their multi-marginal extensions (Shen et al., 2025; Zhang, 2024; Guan et al., 2024; Chen et al., 2024; Lavenant et al., 2024). These methods reconstruct likely trajectories between snapshots at consecutive times. In settings where the goal is to fill in missing timepoints between observed snapshots, these techniques can perform well. However, to the best of our knowledge, no previous papers before the present work presented methodology for forecasting, ran experiments on forecasting, or had code that could immediately run for the forecasting task.

Nonetheless, we can imagine some natural extensions of existing work to forecasting. One option is to follow the SB reference dynamic beyond observed time points, but since the reference dynamic

is typically chosen without considering data, we expect poor extrapolation performance. Chen et al. (2024) and Shen et al. (2025) incorporate momentum or a learned reference, respectively, so one might hope they could extend well to extrapolation. But we expect the extrapolation ability of the latter methods to be limited by an issue shared by all existing SB methods. Namely, existing methods optimize a Kullback–Leibler divergence between data and a nominal diffusion model, and this optimization formulation requires a known volatility, fixed across states. In practice, volatility is often unknown or state-dependent, as in scRNA-seq, where noise arises from a mix of biological and technical sources. We see in our experiments that missing the state dependence or choosing an inappropriate (though standard) volatility value can lead to poor forecasts. In addition, most approaches (with the exception of Shen et al. (2025)) do not take advantage of partial mechanistic knowledge (e.g., bounded production–decay kinetics in a biological system or vortex structure in ocean currents), even though such knowledge is often available and can meaningfully constrain the dynamics. Finally, we note that SB methods often rely on iterative Sinkhorn solvers that offer few tools for diagnosing model fit. We describe additional related work in Appendix A.

In this work, we propose a new framework, SnapMMD, that shifts the modeling focus from interpolation to accurate, interpretable forecasting. Our approach begins with the observation that in typical trajectory-inference settings (Lavenant et al., 2024), each sample can be viewed as an i.i.d. draw from the *joint* distribution of a system's state (e.g., mRNA expression level) and the time of measurement. Then, we characterize a parametric family of stochastic differential equations (SDEs) and seek the member of this family whose implied joint distribution over state and time best matches the empirical joint distribution observed in the data. We perform this matching using maximum mean discrepancy (MMD) with a specific kernel choice, which we show enjoys a number of conceptual and computational benefits. Our framework allows data to guide extrapolation beyond observed times, allows us to learn volatility rather than fixing it in advance, facilitates the use of domain knowledge when available, and yields interpretable model diagnostics, including an explicit velocity field and an $R^2$-like metric that quantifies model fit. Our framework offers the added benefit of enabling robust interpolation and forecasting even with incomplete state measurements, as in the protein expression example above. We evaluate SnapMMD across a range of synthetic and real-world systems. We find that SnapMMD consistently outperforms Schrödinger bridge baselines in all forecasting tasks — while still matching or exceeding interpolation performance of both SB-based and flow-matching baselines in almost all cases.

## 2  SETUP AND BACKGROUND

Though our work has application beyond scRNA-seq, we next describe our data and goals using scRNA-seq terminology to clarify and concretize our notation.

**Data Setup.** We consider single-cell mRNA measurements collected at $I$ distinct time points, labeled $t_1 < t_2 < \cdots < t_I$. For convenience, we set $t_1 = 0$. We do not require these time points to be equally spaced. At each time $t_i$, the observed data consist of $N_i$ cells, each providing a single mRNA expression level measurement (representing a cell state) in $\mathbb{R}^d$, denoted $\boldsymbol{Y}_{t_i}^n$. After a cell's mRNA level is measured, the cell is destroyed. So each cell appears exactly once in the dataset. We therefore collect $N = \sum_{i=1}^{I} N_i$ total observations across all time points. We write $\boldsymbol{Y}_{t_i}^{\text{all}}$ for the full set of measurements taken at time $t_i$.

**Goal.** If a cell's mRNA expression level were not measured, the cell would remain alive, and its mRNA expression would evolve continuously over time along a (latent) trajectory. Formally, we denote the latent trajectory of the $n$th cell observed at the $i$th time step by $\boldsymbol{X}_t^{(i,n)}$. The observed state of the cell at time $t_i$ is a single point on this trajectory $\boldsymbol{Y}_{t_i}^n = \boldsymbol{X}_{t=t_i}^{(i,n)}$. We assume these latent trajectories are independent realizations from an underlying latent distribution. Consequently, the observed snapshot measurements are also independent. Our objective is to infer a probabilistic model, chosen from a specified parametric family, that best captures the distribution of these unobserved trajectories. Our model should provide a distribution over forecasted trajectories beyond the measured time and also over interpolated trajectories between observed times.

**Dataset and independence assumptions.** At each observation time $t_i$, the measurements $\{\boldsymbol{Y}_{t_i}^n\}_n$ are modeled as i.i.d. draws from a conditional distribution $p(Y \mid t = t_i)$. Across different time-points, say $t_i \neq t_j$, the conditional distributions $p(Y \mid t = t_i)$ and $p(Y \mid t = t_j)$ may differ, so the overall dataset is independent but not identically distributed if we treat times as fixed. But we can

model the observation times themselves as random draws from a discrete distribution $h(t)$ supported on the set of measurement times. Then the dataset of pairs $\{(\boldsymbol{Y}_{t_i}^n, t_i)\}_{i,n}$ can be regarded as i.i.d. samples from the joint distribution $p(Y, t) = p(Y \mid t)h(t)$. This joint view aligns with real biological experimental protocols. Indeed, instead of sequencing each sample immediately after collection, experimenters often tag each sample with a unique identifier that encodes the time it was collected. Then, all cells are sequenced together in a single batch, with the time information retrieved from the tags. In Appendix C we discuss situations where this i.i.d. assumption might not hold.

**Model.** We model each latent trajectory of the $(i, n)$ cell with a stochastic differential equation (SDE) driven by a $d$-dimensional Brownian motion $\boldsymbol{W}_t^{(i,n)}$, independent across particles:

$$\mathrm{d}\boldsymbol{X}_t^{(i,n)} = \boldsymbol{b}_0(\boldsymbol{X}_t^{(i,n)}, t)\mathrm{d}t + \boldsymbol{g}_0(\boldsymbol{X}_t^{(i,n)}, t)\mathrm{d}\boldsymbol{W}_t, \quad \boldsymbol{X}_{t=0}^{(i,n)} \sim \pi_0. \tag{1}$$

We assume that the drift $\boldsymbol{b}_0(\cdot, \cdot) : \mathbb{R}^d \times [0, t_I] \to \mathbb{R}^d$ and initial marginal distribution $\pi_0$ are unknown. Previous Schrödinger bridge methods typically assume fixed, known volatility (e.g. De Bortoli et al., 2021; Vargas et al., 2021; Koshizuka and Sato, 2022; Wang et al., 2024; Shen et al., 2025; Zhang, 2024; Guan et al., 2024; Chen et al., 2024). By contrast, we allow the common case where the volatility function $\boldsymbol{g}_0$ can be unknown and also state- and time-dependent. For example, in scRNA-seq, transcriptional noise varies with gene identity, cell state, and developmental stage — and is further confounded by technical artifacts such as amplification bias and stochastic capture. Volatility in these systems reflects true biological uncertainty and is rarely known in advance.

Finally, we assume standard regularity conditions on the SDE: Lipschitz continuity and linear growth for drift and volatility (to ensure existence of strong solutions to the SDE; see Pavliotis, 2016, Chapter 3, Theorem 3.1) and bounded second moments of the particle distributions (ruling out the possibility that the process exhibits unbounded variability). We state these assumptions formally in Appendix B.2.

**One natural extension of Schrödinger bridges to forecasting highlights limitations of fixed reference dynamics.** SB methods reconstruct distributions of trajectories by matching observed marginals while penalizing deviation from a predefined reference process, typically Brownian motion; see Eq. (A1) in Appendix B.1 for a standard setup. These methods are well suited for interpolation between observed marginals. Forecasting, however, has not been a focus in this line of work. In Appendix B.1, we work through one natural extension of the SB framework to forecasting by introducing an additional, future marginal. In this setup, under fixed reference dynamics, extrapolation reduces to propagating the final observed marginal forward under the reference process. But the reference is chosen without considering any data, so we expect poor extrapolation performance.

## 3 OUR METHOD

To address the forecasting limitations of SBs described above, we propose an alternative approach. SBs compare the observed data and candidate model directly via Kullback–Leibler divergence. We instead formulate an optimization problem that directly matches the joint distribution of state–time pairs predicted by a candidate model to the empirical distribution observed in the data. Below, we describe precisely how we frame and solve this optimization problem using MMD, introduce an interpretable diagnostic metric for assessing model fit, and detail how this framework naturally extends to scenarios involving incomplete state measurements.

### 3.1 A LEAST SQUARES APPROACH

We formalize our approach by (1) decomposing the joint state–time empirical distribution into marginal and conditional components and (2) detailing how the resulting matching problem reduces to a least-squares formulation when using Maximum Mean Discrepancy (MMD) for distance.

**Empirical distributions.** We let $\hat{f}(\boldsymbol{y}; t)$ denote the empirical measure over state $\boldsymbol{y}$ at any observed time $t_i$. For unobserved times, we give it a placeholder distribution. Likewise, we let $\hat{h}(t)$ denote the (marginal) empirical measure over observed times. Precisely, we have

$$\hat{f}(\boldsymbol{y}; t) = \begin{cases} (N_i)^{-1} \sum_{n=1}^{N_i} \delta_{\boldsymbol{Y}_{t_i}^n}(\boldsymbol{y}) & t = t_i \\ \delta_{\boldsymbol{0}}(\boldsymbol{y}) & \text{else} \end{cases}, \quad \text{and} \quad \hat{h}(t) = \sum_{i=1}^{I} \left( \frac{N_i}{\sum_{j=1}^{I} N_j} \right) \delta_{t_i}(t). \tag{2}$$

where $\delta$ denotes as usual the Dirac measure. Let $\hat{f}(\boldsymbol{y}, t)$ denote the empirical joint distribution over observed state-time pairs. This joint decomposes into the empirical marginal and conditional described above: $\hat{f}(\boldsymbol{y}, t) = \hat{h}(t)\hat{f}(\boldsymbol{y}; t)$.

**Directly matching the joint state–time distributions.** We plan to estimate the parameters $\boldsymbol{\theta}$ of a candidate SDE model by aligning its predicted joint distribution over state–time pairs with the empirical distribution. Let $f_{\boldsymbol{\theta}}(\boldsymbol{y}|t)$ denote the predicted state distribution at time $t$. We will minimize a discrepancy between (1) the empirical joint $\hat{f}(\boldsymbol{y}, t)$ and (2) the joint implied by the predictive conditional $f_{\boldsymbol{\theta}}(\boldsymbol{y}|t)$ together with the empirical marginal $\hat{h}(t)$: namely, $f_{\boldsymbol{\theta}}(\boldsymbol{y}, t) := \hat{h}(t)f_{\boldsymbol{\theta}}(\boldsymbol{y}|t)$.

For the discrepancy, we choose the MMD (Gretton et al., 2012). MMD computes the squared distance between the kernel mean embeddings of two distributions in a reproducing kernel Hilbert space (RKHS). While in principle other measures of divergence (e.g., Wasserstein, Kullback–Leibler) could be employed, we adopt MMD for its favorable computational and statistical properties. In particular, MMD admits a closed-form empirical estimator with efficient gradients, remains well behaved in moderately high dimensions where Wasserstein distances suffer from poor sample complexity, and naturally provides a quantitative model selection criterion, as discussed in Section 3.2. We refer to Appendix D for further discussion on the computational aspects of MMD.

To apply MMD in practice, one must specify a kernel. For our method, we choose a kernel that factors across state and time; this choice lets us break the joint MMD into a weighted sum of marginal MMDs at each time point. The resulting optimization objective is reminiscent of least-squares regression, with time acting as a discrete index. We formalize this idea in the following result.

**Proposition 3.1.** *Let $f(\boldsymbol{y}, t) = f(\boldsymbol{y} \mid t)h(t)$ and $g(\boldsymbol{y}, t) = g(\boldsymbol{y} \mid t)h(t)$ be joint distributions over $\boldsymbol{y} \in \mathbb{R}^d$ (for dimension $d$) and discrete time $t \in \mathcal{T}$, where $\mathcal{T}$ denotes a finite index set of observation times, $h(t)$ is a probability mass function and $f(\boldsymbol{y} \mid t), g(\boldsymbol{y} \mid t)$ are conditional distributions. Use the kernel $K((\boldsymbol{y}, t), (\boldsymbol{y}', t')) = K_{\boldsymbol{y}}(\boldsymbol{y}, \boldsymbol{y}')\delta(t - t')$, where $K_{\boldsymbol{y}}$ is positive definite on the state space and, for all $t \in \mathcal{T}$, $\mathbb{E}_{\boldsymbol{y} \sim f(\boldsymbol{y}|t), \boldsymbol{y}' \sim f(\boldsymbol{y}|t)}K_{\boldsymbol{y}}(\boldsymbol{y}, \boldsymbol{y}') < \infty$, $\mathbb{E}_{\boldsymbol{y} \sim f(\boldsymbol{y}|t), \boldsymbol{y}' \sim g(\boldsymbol{y}|t)}K_{\boldsymbol{y}}(\boldsymbol{y}, \boldsymbol{y}') < \infty$, and $\mathbb{E}_{\boldsymbol{y} \sim g(\boldsymbol{y}|t), \boldsymbol{y}' \sim g(\boldsymbol{y}|t)}K_{\boldsymbol{y}}(\boldsymbol{y}, \boldsymbol{y}') < \infty$.[1] Then:*

$$\mathrm{MMD}_K^2(f, g) = \sum_{t \in \mathcal{T}} h^2(t) \, \mathrm{MMD}_{K_{\boldsymbol{y}}}^2 \left(f(\cdot \mid t), g(\cdot \mid t)\right).$$

This result (proof in Appendix B.3) shows that aligning joint distributions here boils down to matching conditional state distributions across time. We apply Proposition 3.1 with the two joint distributions from above, $f_{\boldsymbol{\theta}}(y, t)$ and $\hat{f}(y, t)$, to obtain the following optimization objective:

$$\mathrm{MMD}_K^2(f_{\boldsymbol{\theta}}, \hat{f}) = \sum_{i=1}^{I} \left(\frac{N_i}{\sum_{j=1}^{I} N_j}\right)^2 \mathrm{MMD}_{K_{\boldsymbol{y}}}^2(f_{\boldsymbol{\theta}}(\cdot|t_i), \hat{f}(\cdot; t_i)). \tag{3}$$

It remains to estimate the righthand MMDs and also to choose the MMD state-space kernel, the model $f_{\boldsymbol{\theta}}(\cdot|t)$, and the optimization algorithm.

**Estimating MMD at each time point.** In practice, we approximate $\mathrm{MMD}_{K_{\boldsymbol{y}}}^2(f_{\boldsymbol{\theta}}(\cdot|t), \hat{f}(\cdot; t))$ using the MMD's U-statistic estimator (Gretton et al., 2012, Lemma 6). To that end, we simulate $M$ trajectories from the candidate model. For the $m$th trajectory, we record the state snapshot $\boldsymbol{Z}_{t_i}^m$ at time $t_i$. The U-statistic estimator, which is unbiased and consistent (Hall, 2004), is then given by

$$\mathrm{MMD}_{\mathrm{U,U}}^2(f_{\boldsymbol{\theta}}(\cdot|t_i), \hat{f}(\cdot; t_i)) = \frac{1}{N_i(N_i - 1)} \sum_{n \neq n'} K_{\boldsymbol{y}}(\boldsymbol{Y}_{t_i}^n, \boldsymbol{Y}_{t_i}^{n'}) - \frac{2}{N_i M} \sum_{n,m} K_{\boldsymbol{y}}(\boldsymbol{Y}_{t_i}^n, \boldsymbol{Z}_{t_i}^m)$$

$$+ \frac{1}{M(M - 1)} \sum_{m \neq m'} K_{\boldsymbol{y}}(\boldsymbol{Z}_{t_i}^m, \boldsymbol{Z}_{t_i}^{m'}).$$

The overall optimization problem for parameter fitting then becomes:

$$\hat{\boldsymbol{\theta}} = \arg\min_{\boldsymbol{\theta}} \sum_{i=1}^{I} w_i \, \mathrm{MMD}_{\mathrm{U,U}}^2(f_{\boldsymbol{\theta}}(\cdot|t_i), \hat{f}(\cdot; t_i)) \quad \text{with weights} \quad w_i := \left(\frac{N_i}{\sum_{j=1}^{I} N_j}\right)^2 \tag{4}$$

[1]Many practical kernels satisfy these assumptions, e.g., radial basis function, Matérn, and Laplace.

Table 1: MMD for forecast and interpolation tasks. For forecast, we report mean and standard deviation over 10 random seeds. For interpolation, we aggregate over 10 seeds and validation time points. The best method (lowest MMD) is shown in bold in a green cell; we also highlight in green any other methods whose mean is contained in the one-standard deviation interval for the best method. We show the top three interpolation methods here; full results can be found in Appendix F.4.

| Task | Forecast | | | Interpolation | | |
|---|---|---|---|---|---|---|
| | **Ours** | SBIRR-ref | SB-forward | **Ours** | SBIRR | DMSB |
| LV | $\mathbf{0.057 \pm 0.03}$ | $0.69 \pm 0.11$ | $2.99 \pm 1.88$ | $\mathbf{0.01 \pm 0.01}$ | $0.04 \pm 0.08$ | $3.31 \pm 2.18$ |
| ReprParam | $\mathbf{0.072 \pm 0.080}$ | $2.06 \pm 1.34$ | $2.09 \pm 0.74$ | $\mathbf{0.04 \pm 0.03}$ | $0.16 \pm 0.10$ | $7.23 \pm 2.00$ |
| ReprSemiparam | $\mathbf{0.32 \pm 0.15}$ | $1.46 \pm 0.55$ | $5.26 \pm 1.66$ | $\mathbf{0.38 \pm 0.38}$ | $1.11 \pm 0.82$ | $7.23 \pm 2.00$ |
| ReprProtein | $\mathbf{0.048 \pm 0.029}$ | $2.50 \pm 0.05$ | $2.42 \pm 0.13$ | $\mathbf{0.01 \pm 0.00}$ | $1.48 \pm 2.95$ | $5.15 \pm 2.63$ |
| GoM | $2.36 \pm 0.11$ | $\mathbf{1.41 \pm 0.18}$ | $2.59 \pm 0.33$ | $0.07 \pm 0.05$ | $\mathbf{0.01 \pm 0.01}$ | $0.05 \pm 0.06$ |
| PBMC | $\mathbf{0.11 \pm 0.04}$ | $0.71 \pm 0.34$ | $2.38 \pm 0.49$ | $\mathbf{0.11 \pm 0.05}$ | $0.11 \pm 0.12$ | $0.97 \pm 0.10$ |

The least squares objective from Eq. (4) naturally extends classical regression to distributional settings by measuring discrepancy in the RKHS defined by kernel mean embeddings. Specifically, this least squares framework reduces to classical Euclidean regression if each predicted distribution is a Dirac measure and the kernel is linear.

**Choosing kernel, optimizer, and SDE model.** In practice, we use the radial basis function (RBF) kernel for the state space; we determine the length scale by the median heuristic (Garreau et al., 2017) applied to pairwise distances in the data. Additionally, we scale time to lie in $[0, 1]$. For optimization, we compute gradients with respect to the parameters using the stochastic adjoint method (Li et al., 2020). We use the Adam optimizer to perform the parameter updates. We pick the candidate conditional $f_{\boldsymbol{\theta}}(\cdot | t)$ based on domain knowledge by specifying parametric components of the SDE in every experiment, reflecting the partial mechanistic understanding available in scientific settings. A strength of our method is that it can leverage this information, and in Section 4.6 we show through ablations that doing so yields clear improvements over fully neural models.

**Handling incomplete state measurements.** In many practical applications — such as our mRNA sequencing example — it is common for only a subset of relevant state variables to be observed. For instance, while mRNA concentrations are routinely measured, corresponding protein levels (which are also important for modeling the underlying dynamical system) are often unavailable. Our framework can handle these missing-data settings because it relies on matching the joint distribution of time and the observed dimensions, rather than requiring all dimensions to be measured. More precisely, since our loss (Eq. (4)) is defined over the observed state variables (together with time), the model is trained to match the marginal distribution of the observed variables along with time, without making any additional assumptions or imputations for the missing dimensions. We illustrate with an experiment in Section 4 (Fig. 2, lower row).

## 3.2 EVALUATING MODEL FIT WITH AN $R^2$-STYLE METRIC

In traditional regression, $R^2$ offers a straightforward diagnostic and basis for model comparison; we propose a similar metric for distributional data. Recall that $R^2$ quantifies how well a model explains the data by comparing residual variability against total variability around a baseline constant prediction, the mean response. Given the least-squares objective of Eq. (4), we might consider a similar metric in the present distributional case. But first we need to choose an appropriate baseline.

**A distributional baseline.** Analogous to the response-mean baseline in traditional regression, we want to find the best constant (time-independent) model within our RKHS-based least squares framework. The barycenter of distributional data is the distribution minimizing the sum of squared RKHS distances to all observed distributions. For distributions embedded in an RKHS, Cohen et al. (2020) showed that the barycenter is the weighted mixture of the empirical distributions:

$$f_{\text{bary}}(\boldsymbol{y}) := \arg\min_f \sum_{i=1}^{I} w_i \, \text{MMD}^2_{K_{\boldsymbol{y}}}(f(\boldsymbol{y} \mid t_i), \hat{f}(\boldsymbol{y}; t_i)) = \frac{1}{\sum_{i=1}^{I} w_i N_i} \sum_{i=1}^{I} \sum_{n=1}^{N_i} w_i \delta_{\boldsymbol{Y}_{t_i}^n}(\boldsymbol{y}), \quad (5)$$

with weights $w_i$ as in Eq. (4).

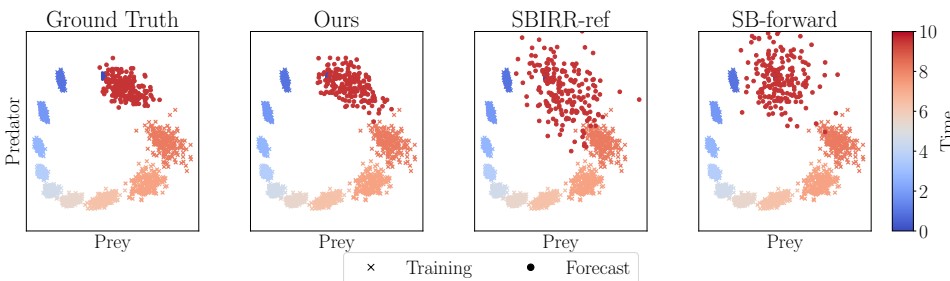

Figure 1: Lotka-Volterra results (Section 4.1). We show 200 samples at each of 10 training times and 1 forecast time (red). Forecast points overlap with the training points at time 0 (blue).

**Our RKHS-based $R^2$ metric.** With this baseline in hand, we define our metric.

**Definition 3.1** (RKHS-based $R^2$ metric). *Let $f_{\boldsymbol{\theta}}(\cdot|t)$ denote the model-predicted state distribution at time $f_{\boldsymbol{\theta}}(\cdot|t)$. Let $\hat{f}(\cdot; t_i)$ be defined as in Eq. (2), $f_{bary}$ as in Eq. (5), and $w_i$ as in Eq. (4). The RKHS-based coefficient of determination $R^2$ is*

$$R^2 = 1 - \frac{\sum_{i=1}^{I} w_i \ \text{MMD}^2_{K_{\boldsymbol{y}}}(f_{\boldsymbol{\theta}}(\cdot|t_i), \hat{f}(\cdot; t_i))}{\sum_{i=1}^{I} w_i \ \text{MMD}^2_{K_{\boldsymbol{y}}}(f_{bary}, \hat{f}(\cdot; t_i))}. \tag{6}$$

The numerator measures the discrepancy between model predictions and empirical distributions, while the denominator quantifies total variability around the barycenter (analogous to total variance in standard regression). Thus, $R^2$ captures the fraction of variability explained by the model. By construction, $R^2 \leq 1$, with values near 1 indicating good fit and values near or below 0 indicating performance no better (or worse) than the barycenter. We use this metric for model comparison and fit diagnostics; see Appendix F.2.

## 4 EXPERIMENTS

In synthetic and real-data experiments, we find that our SnapMMD method consistently provides better forecasts than competitors. We also find that, in almost all experiments, SnapMMD provides better or matching interpolation performance relative to competitors. In Appendix F.1, we further demonstrate that our method outperforms competitors on vector field reconstruction.

**Beyond cell states.** Though we use cell-state terminology above for concreteness, our experiments also include applications beyond cellular dynamics. E.g., states can instead represent predator and prey counts (Section 4.1) or spatial locations of particles following ocean currents (Section 4.4).

**Metrics of success.** To evaluate forecasting performance, we reserve a validation snapshot at a future time point beyond the training horizon. For interpolation, we retain intermediate validation snapshots between training time points. In both forecasting and interpolation tasks, we measure discrepancy between the validation data and predictions with two metrics: (1) the MMD and (2) the earth mover's distance (EMD)[2] between the forecast distribution and the held-out empirical distribution at the validation time. We also provide visual comparisons.

**Forecasting (and vector field reconstruction) baselines.** We compare SnapMMD against two baselines: (1) Schrödinger bridge with iterative reference refinement (SBIRR) (Shen et al., 2025; Zhang, 2024; Guan et al., 2024), and (2) multimarginal Schrödinger bridge with shared forward drift (SB-forward) (Shen et al., 2024). Although designed for interpolation (SBIRR) and vector field reconstruction (SB-forward), they can be adapted for forecasting by extrapolating with their fitted dynamics. Specifically, we use (1) the best fitted reference of SBIRR (SBIRR-ref) and (2) the best fitted forward drift of SB-forward. Reference fitting has also been used in prior work with linear SDE models (Zhang, 2024; Guan et al., 2024), whereas SBIRR and SBIRR-ref allow general model families. To ensure fairness, both baselines are provided with the same drift structure as SnapMMD. Their key limitation is that they rely on a fixed, known volatility; following prior work (Vargas et al., 2021; Wang et al., 2021; Shen et al., 2025), we set it to 0.1. Unlike SnapMMD, they cannot learn state-dependent diffusion.

---

[2]For EMD, we use the implementation provided by Tong et al. (2024b).

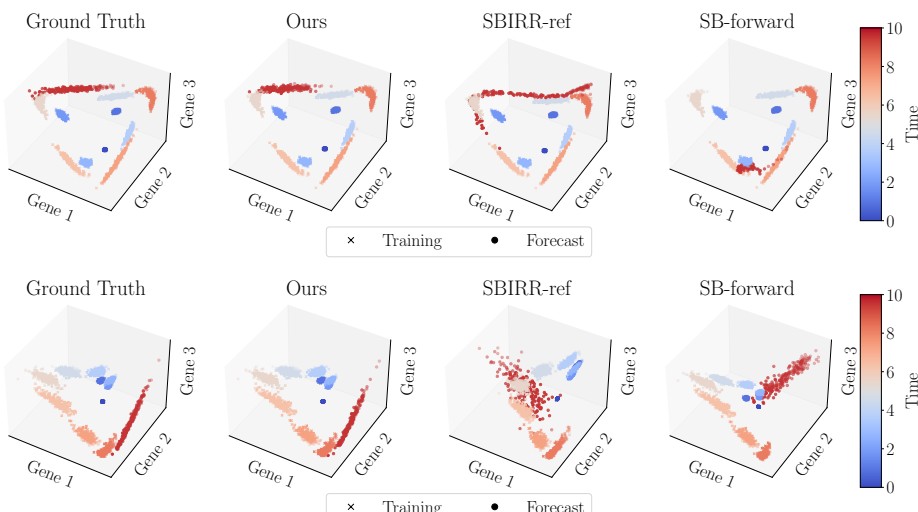

Figure 2: Repressilator results: mRNA-only (upper, Section 4.2) and mRNA and protein (lower, Section 4.3). We show 200 samples at each of 10 training times and 1 forecast time (red).

**Interpolation baselines.** For interpolation, we compare SnapMMD against five methods: optimal transport–conditional flow matching (`OT-CFM`) and Schrödinger bridge–conditional flow matching (`SB-CFM`) (Tong et al., 2024a), simulation-free Schrödinger bridge (`SF2M`) (Tong et al., 2024c), deep momentum multimarginal Schrödinger bridge (`DMSB`) (Chen et al., 2024), and again Schrödinger bridge with iterative reference refinement (`SBIRR`) (Shen et al., 2025), where here we use the main algorithm output (rather than the fitted reference as in `SBIRR-ref`). Among these, only `SBIRR` supports incorporating the same drift structure as SnapMMD. The other methods cannot accept such inductive bias by design, but we nevertheless include them, since contrasting structure-free approaches with structured ones highlights the contribution of domain knowledge. For all methods, we use default code settings.

## 4.1 LOTKA–VOLTERRA SYSTEM

**Setup.** We simulated data from a two-dimensional Lotka–Volterra predator–prey system, where each coordinate's volatility scales proportionally with its state variable. E.g., we set the volatility for the prey population $X$ to be $\sigma X$, with the same constant $\sigma$ across predator and prey. We train on 10 time points, each with 200 samples. For methods that take a model choice (ours and `SBIRR` variants), we use a parametric Lotka–Volterra model. See Appendix F.5 for full details.

**Results.** In Fig. 1, we see that our method's forecast (red dots) is closer to ground truth than the baselines are. MMD (LV, Forecast in Table 1) and EMD (LV, Forecast in Table 2) agree that our method performs best. Our method is also best in the interpolation task (LV, Interpolation in Table 1). See Appendix F.5 for further results.

## 4.2 REPRESSILATOR: MRNA ONLY

**Setup.** We simulated mRNA concentration data from a repressilator system, a biological clock composed of three genes that inhibit each other in a cyclic manner. As for Lotka–Volterra, we let each coordinate's volatility scale proportionally with its state variable. We train on 10 time points, each with 200 samples. For methods that take a model choice, we consider two options. (1) We use the same parametric model as the data-generating process; see Appendix F.6 for full results. (2) We use a semiparametric model with a multilayer perceptron; see Appendix F.7.2 for details.

**Results.** In Fig. 2 (upper row), we see that, when using the semiparametric model, our method's forecast (red dots) is closer to ground truth than the baselines are. MMD (ReprSemiparam, Forecast in Table 1) and EMD (ReprSemiparam, Forecast in Table 2) agree that our method performs best. Our method is also tied for best in the interpolation task (ReprSemiparam, Interpolation in Table 1). We find similar results when using the parametric model (Tables 1 and 2, Fig. A8). See Appendix F.6 and Appendix F.7 for further results.

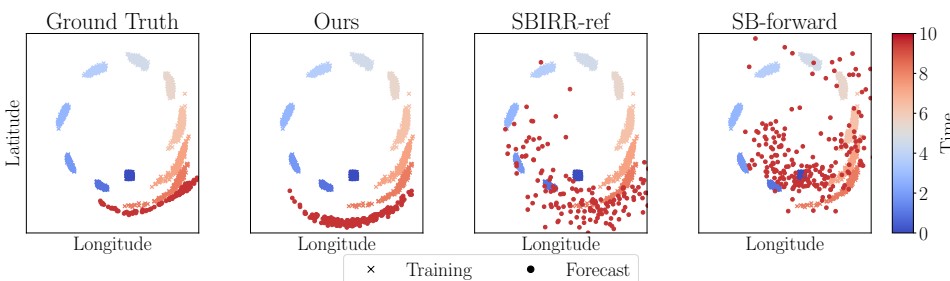

Figure 3: Gulf of Mexico results (Section 4.4). We show 200 samples at each of 10 training times and 1 forecast time (red).

## 4.3 REPRESSILATOR: MRNA AND PROTEIN

**Setup.** A more-complete biochemical model of the repressilator includes both mRNA and protein (Eq. (A12)) even though only mRNA concentration is actually observed in practice. We next generate simulated data using the more-complete model and keep only the mRNA concentrations in our observations. We again train on 10 time points, each with 200 samples. Since our method has the capacity to handle incomplete state observations, we can use the full mRNA-protein model with our method. Since SBIRR methods do not have the capacity to handle models with latent variables, we use the mRNA-only model in these methods.

**Results.** In Fig. 2 (lower row), we see that our method's forecast (red dots) is closer to ground truth than the baselines are. MMD (ReprProtein, Forecast in Table 1) and EMD (ReprProtein, Forecast in Table 2) agree that our method performs best. We emphasize that none of the methods directly observe protein levels. But since our method is aware that protein levels are also driving the underlying dynamics, it is able to better forecast mRNA concentration. Our method is also best in the interpolation task (ReprProtein, Interpolation in Table 1). See Appendix F.8 for further results.

## 4.4 OCEAN CURRENTS IN THE GULF OF MEXICO

**Setup.** We use real ocean-current data from the Gulf of Mexico: namely, high-resolution (1 km) bathymetry data from the HYbrid Coordinate Ocean Model (HYCOM) reanalysis.[3] We extract a velocity field centered on a region that appears to exhibit a vortex. We then simulate the motion of particles — representing buoys or ocean debris — evolving under this field. The training data consist of 10 time points with 400 particles each. Since the data are real in this experiment, the models used by any method must be misspecified. For methods that take a model choice, we use a physically motivated model for the vortex, where the velocity field is the sum of a Lamb-Oseen vortex and a constant divergence field. The first term accounts for swirling, rotational dynamics typical of a vortex in low viscosity fluid like water, while the divergence field accounts for vertical motion or non-conservative forces that may cause a net expansion or contraction of the flow. See Appendix F.9.3 for more details. Note that physical drifters deployed in the ocean can be tracked continuously over time. By contrast, in many remote sensing applications each observation is an image that provides only a distributional snapshot at that time, without tracking the same particles across times. For example, in oil-spill monitoring from satellite imagery, each image shows the surface oil distribution but not individual particle paths. See Appendix F.9.2 for discussion.

**Results.** In Fig. 3, we see that our method's forecast (red dots) more closely aligns with ground truth than the baselines do. EMD (GoM, Forecast in Table 2) agrees that our method performs best. However, MMD (GoM, Forecast in Table 1) prefers SBIRR-ref to our method (SnapMMD); we suspect we see this behavior because MMD using an RBF kernel can prefer a diffuse, but less accurate, cloud over a concentrated, geometrically correct one. Recall that the MMD with RBF mixes two ingredients: (i) how tightly the forecast particles cluster among themselves and (ii) how far the forecast and ground truth particles are. Our forecast points sit on a lower-dimensional curve than the baselines' points, so the "self-similarity" part of the MMD score is higher. All methods perform well visually at the interpolation task (Fig. A18), but SBIRR yields the best MMD (GoM, Interpolation in Table 1) and EMD (GoM, Interpolation in Table 2). SBIRR is built to interpolate every observed snapshot and then smooth between them, so with densely sampled times, it almost

---

[3]Dataset available at https://www.hycom.org/data/gomb0pt01/gom-reanalysis.

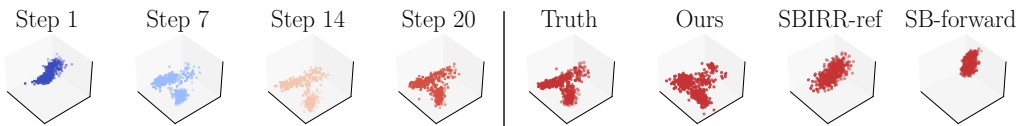

Figure 4: PBMC results (Section 4.5). The axes in every plot are the same three principal components, computed over the full data: i.e., 41 time steps of the 30-dimensional gene programs. Leftmost four panels: evolution of the training data at time steps 1, 7, 14, and 20. "Truth" panel: ground truth snapshot at time step 21. Rightmost three panels: model forecasts at time step 21.

inevitably lands near the held-out validation points. Our method's aim to recover a smooth velocity field (rather than enforce exact interpolation) can be an advantage for forecasting, but less so for interpolation. See Appendix F.9.6 for further results.

### 4.5 T CELL-MEDIATED IMMUNE ACTIVATION

**Setup.** We use a real single-cell RNA-sequencing dataset that tracks T cell–mediated immune activation in peripheral blood mononuclear cells (PBMCs) (Jiang et al., 2024). Scientists recorded gene-expression profiles every 30 minutes for 30 hours. We use the 41 snapshots collected between 0 h and 20 h — prior to the onset of steady state; we take 20 alternating snapshots (at integer hours) for training and the remaining 21 for validation. We use the 30-dimensional projection ("gene program") of the original measurements released by Jiang et al. (2024) as our data. In Fig. 4 (leftmost four), we show the training snapshots at four time steps. Fig. A20 shows the full progression of the training points over 20 time steps. Since the data is real in this experiment, the model used by any method must be misspecified. For methods that use a model, we use the same model as in the semiparametric repressilator experiment (Appendix F.7.1). Full details are in Appendix F.10.1.

**Results.** In Fig. 4 (rightmost four), we see that our method's forecast is closer to ground truth than the baselines are. MMD (PBMC, Forecast in Table 1) agrees. We do not use EMD in this experiment as it suffers from the curse of dimensionality and is thus unreliable in our 30-dimensional setting; see the discussion at the end of Section 2.5.2 in Chewi et al. (2025). Our method ties with SBIRR in the interpolation task (PMBC, Interpolation in Table 1). See Appendix F.10.5 for more results.

### 4.6 ABLATIONS: FIXED VOLATILITY AND FULLY NEURAL SDE MODEL

One of the proposed benefits of SnapMMD is the ability to learn the volatility and allow it to be state-dependent. We test the importance of this flexibility by comparing to a variant of SnapMMD with volatility fixed to the same constant as in other methods. In almost every forecasting or interpolation task, errors from the fixed-volatility variant increase by at least a factor of two and often by more than an order of magnitude, relative to SnapMMD. See Appendix G.1 for full details.

Another proposed benefit of SnapMMD is the ability to incorporate domain knowledge, which is widely available in scientific applications. We test the role of domain knowledge by replacing the (parametric or semiparametric with neural residual) SDE families in the experiments above with fully neural drift and diffusion. Again, in almost every task, performance drops sharply when the domain knowledge is not incorporated. See Appendix G.2.

## 5 DISCUSSION

In this work, we introduced a new method for learning SDEs from population-level snapshot data. Our approach is based on matching state–time distributions using a least squares scheme in a distributional space. Our proposed method handles real-life challenges such as unknown and state-dependent volatility, missing dimensions, and diagnostics for performance. Overall, our experiments indicate that our proposed framework outperforms existing methods in a wide range of applications. While many scientific applications (including those above) feature useful models, it can nonetheless be a limitation that our method requires specification of an appropriate model family. A poorly chosen model family can lead to bad performance. An overly wide family can face identifiability challenges. In fact, even the complete time series of marginal distributions need not always uniquely determine the drift and volatility functions of the SDE; see Appendix H for further discussion. Finally, our method is not simulation-free, so it requires compute that scales with sample size and state dimension; development of a simulation-free method is an interesting future direction.

**Ethics statement.** This work does not involve human subjects or personally identifiable information, and the datasets we use (PBMC, Repressilator simulations, Lotka–Volterra, Gulf of Mexico HYCOM reanalysis) are all either publicly available or simulated. We believe the main applications of our method are in scientific forecasting tasks, such as gene regulation, ecological modeling, and environmental monitoring, which we view as positive in their social impact. Nonetheless, we note that any forecasting tool could in principle be misapplied in sensitive contexts (e.g., healthcare or policy). We adhere to the ICLR Code of Ethics in the conduct and reporting of this work.

**Reproducibility statement.** We have taken several steps to ensure reproducibility. All theoretical results are stated in the main text (Section 3) with complete proofs in Appendix B.3. Experimental setups, including SDE parameterizations, training details, and evaluation metrics, are described in Section 4 as well as in Appendix F, where we have one appendix for each experiment. Code to reproduce all experiments is provided at https://anonymous.4open.science/r/snapMMD-DD84/.

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

SUPPLEMENTARY MATERIAL

## A    ADDITIONAL RELATED WORK

In this section, we discuss related work covering generative modeling, Schrödinger bridges, flow matching, and other approaches to trajectory inference.

**Generative modeling.** The machine learning community has made substantial progress in sampling from complex, unknown distributions by transporting particles using diffusion models (Ho et al., 2020; Song and Ermon, 2019; Song et al., 2021), Schrödinger bridges (De Bortoli et al., 2021; Pavon et al., 2021; Vargas et al., 2021; Wang et al., 2024), continuous normalizing flows (Chen et al., 2018; Grathwohl et al., 2019), and flow matching (Lipman et al., 2023). These methods are primarily designed for generative modeling, where the goal is to transform a simple distribution (e.g., a Gaussian) into a data distribution. While they involve a notion of "time," it typically serves as an auxiliary dimension rather than representing physical or real-world temporal dynamics. As a result, these methods focus on two-marginal transport problems and are not naturally suited to forecasting or modeling systems with multiple observed time points.

**Trajectory inference with Schrödinger bridges, flow matching and normalizing flows.** Several recent works have explored the use of Schrödinger bridges (SBs) and optimal transport for modeling trajectories across time (Schiebinger et al., 2019; Yang and Uhler, 2019). Most SB-based methods are restricted to pairwise interpolation between consecutive marginals and cannot handle multiple marginals in a principled way. Several extensions have been proposed to address this. DMSB (Chen et al., 2024) incorporates momentum into the particles to exploit local information and model multi-marginal dynamics, while Hong et al. (2025) proposed generalizations using higher-order derivatives, although these approaches are computationally feasible only in low-dimensional settings. Shen et al. (2025) introduced SBIRR, a multi-marginal Schrödinger bridge method with iterative reference refinement. By integrating information from all marginals into the reference dynamics, SBIRR enables not only interpolation but also extrapolation via learned dynamics, and serves as one of the main baselines in our experiments. Tong et al. (2024c) introduced SF2M, a simulation-free SB method based on score and flow matching, designed for efficiency but limited to Brownian motion as the reference and two-marginal settings.

Hashimoto et al. (2016) proposed a regularized recurrent neural network trained with a Wasserstein gradient flow loss for trajectory reconstruction. Bunne et al. (2022) extended this direction with JKOnet, a neural implementation of the Jordan–Kinderlehrer–Otto (JKO) scheme, enabling learning of the underlying energy landscape. More recently, Terpin et al. (2024) proposed JKOnet*, which bypasses the bilevel optimization of JKOnet by exploiting first-order optimality conditions of the JKO scheme, thereby enabling efficient and accurate recovery of potential, interaction, and internal energy components from population snapshots. While JKOnet* extends JKOnet by learning potential, interaction, and internal energies from population snapshots, its core assumption is that dynamics arise from gradient flow on an energy landscape. This makes it well suited for physical systems but less natural for regulatory feedback networks like the Repressilator, where oscillations are driven by transcriptional repression and delayed activation rather than energy minimization. Moreover, JKOnet* evaluates performance by splitting train/test data within each timepoint, whereas our setting holds out entire timepoints to assess extrapolative forecasting ability, making direct comparison challenging.

Other works use deterministic flows to infer trajectories. TrajectoryNet (Tong et al., 2020) combines dynamic optimal transport with continuous normalizing flows (CNFs) to generate continuous-time, nonlinear trajectories from snapshot data. These flows are governed by ODEs rather than SDEs, and incorporate regularization that encourages short, energy-efficient paths. Tong et al. (2024a) extended flow matching by modeling marginals as mixtures indexed by a latent variable. They proposed OT-CFM and SB-CFM, where each component evolves under a simple vector field and the overall dynamics are determined by mixing strategies based on optimal transport or SB principles. Like many flow-matching methods, these approaches are restricted to two marginals and are applied piecewise for longer time series. To address issues of geometry in the data space, Huguet et al. (2022) proposed MIOFlow, which first learns the data manifold and then solves the flow problem within that learned structure. They use neural ODEs to transport points along this manifold. Atanackovic et al. (2024) introduced methods to learn multiple flows over a Wasserstein manifold, though these too are aimed at interpolation rather than forecasting.

Overall, these methods move particles using learned vector fields and various forms of regularization, but they are generally not designed for forecasting. Like many SB-based methods, they suffer when applied to extrapolation tasks, particularly in the absence of a good prior or reference dynamic.

## B  PROOFS AND ADDITIONAL RESULTS

### B.1  A NATURAL EXTENSION OF SCHRÖDINGER BRIDGES TO FORECASTING

In this section, we analyze one natural extension of the Schrödinger bridge formulation to forecasting. This is not the only possible setup, but it illustrates the limitations that arise when the reference process is fixed a priori.

**The Schrödinger bridge problem.** Schrödinger bridges provide a framework for modeling stochastic processes conditioned on observed marginals. Formally, they solve the following constrained optimization problem: given a reference process $p$ (typically Brownian motion), we seek a trajectory distribution $q$ that is closest to $p$ in Kullback–Leibler divergence while matching observed marginals $\{\pi_1, \ldots, \pi_T\}$:

$$\arg\min_{q \in \mathcal{Q}} D_{\mathrm{KL}}(q \,\|\, p), \quad \mathcal{Q} = \{q : q_{t_i} = \pi_i, \; i = 1, \ldots, T\}. \tag{A1}$$

Typically, both $p$ and $q$ are defined via SDEs sharing the same volatility, and are supported over a finite time horizon.

**A natural forecasting extension.** To reason about forecasting, one can extend this setup by introducing an unknown future marginal $\pi_{T+1}$ and solving:

$$\pi_{T+1} = \arg\min_{\pi} \min_{q \in \mathcal{Q}} D_{\mathrm{KL}}(q \,\|\, p), \quad \mathcal{Q} = \{q : q_{t_i} = \pi_i, \; i = 1, \ldots, T, \; q_{t_{T+1}} = \pi\}. \tag{A2}$$

We emphasize that equation A2 is not the only possible formulation of SB forecasting, but it illustrates the limitations of setups that rely on fixed reference dynamics. In this setting, forecasting amounts to evolving particles forward from the last known marginal $\pi_T$ under the reference process $p$. To make this precise, we adopt notation from Lavenant et al. (2024). Let $p_{t_i, t_{i+1}}$ and $q_{t_i, t_{i+1}}$ denote the transition densities of $p$ and $q$ over $[t_i, t_{i+1}]$. Let $q_{t_i} p_{t_i, t_{i+1}}$ denote the joint trajectory distribution that begins with marginal $q_{t_i}$ and evolves forward using the dynamics of $p$ until time $t_{i+1}$.

**Proposition B.1** (Forecasting limitation of fixed-reference SB methods). *Assume that $p$ and $q$ are SDEs with the same volatility, finite time horizon and satisfying Assumption B.1 and Assumption B.2. The solution to equation A2 is the marginal at $t_{T+1}$ of $\pi_T p_{t_T, t_{T+1}}$; that is, forecasting at $t_{T+1}$ reduces to evolving $\pi_T$ forward using the reference dynamics $p$ until $t_{T+1}$.*

*Proof.* When $p$ and $q$ share volatility and finite horizon, the processes are Markovian. Using Proposition D.1 and Remark D.2 of Lavenant et al. (2024), we decompose the KL objective:

$$D_{\mathrm{KL}}(q \,\|\, p) = D_{\mathrm{KL}}(q_{t_1, t_2} \,\|\, p_{t_1, t_2}) + \sum_{i=2}^{T} D_{\mathrm{KL}}(q_{t_i, t_{i+1}} \,\|\, q_{t_i} p_{t_i, t_{i+1}}) + D_{\mathrm{KL}}(q_{t_T, t_{T+1}} \,\|\, q_{t_T} p_{t_T, t_{T+1}})$$

$$\tag{A3}$$

The first $T$ terms are fixed by the constraints on $\{\pi_1, \ldots, \pi_T\}$ and do not depend on $\pi_{T+1}$, so the final term governs the choice of $\pi_{T+1}$. This term is minimized by setting:

$$q_{t_T, t_{T+1}} = \pi_T p_{t_T, t_{T+1}} \tag{A4}$$

which yields $q_{t_{T+1}} = \pi_{T+1}$ as the marginal of $\pi_T$ evolved forward under $p$. Thus, the optimal forecast is the marginal of $\pi_T p_{t_T, t_{T+1}}$. $\qquad\square$

This result shows that in standard SB setups with fixed reference dynamics $p$, one formulation of the forecasting problem reduces to propagating the last observed marginal forward under $p$. Consequently, the quality of SB-based forecasts in this extension is determined entirely by the choice of reference and does not adapt to earlier snapshots, unless additional structure is introduced (e.g., Shen et al., 2025). One alternative approach by Chen et al. (2024) augments the dynamics with momentum terms that allow for better extrapolation. However, this approach still relies on a fixed reference process, and furthermore the released codebase does not provide a forecasting mode (which would require substantial re-engineering). Thus we benchmark this method only on interpolation tasks in our experiments.

## B.2   REGULARITY ASSUMPTIONS

For completeness, we state the standard SDE regularity assumptions used in the main text.

**Assumption B.1.** *The drifts and volatility are $L$ and $L'$-Lipschitz respectively; i.e., for all $t \in [0, t_I]$, $\boldsymbol{x_1}, \boldsymbol{x_2} \in \mathbb{R}^d$, $\|\boldsymbol{b}_0(\boldsymbol{x_1}, t) - \boldsymbol{b}_0(\boldsymbol{x_2}, t)\| \leq L\|\boldsymbol{x_1} - \boldsymbol{x_2}\|$ and $|\boldsymbol{g}_0(\boldsymbol{x_1}, t) - \boldsymbol{g}_0(\boldsymbol{x_2}, t)| \leq L'\|\boldsymbol{x_1} - \boldsymbol{x_2}\|$, where $\|\cdot\|$ denotes the usual Euclidean norm of a vector. And we have at most linear growth; i.e., there exist $K, K' < \infty$ and constant $c$ such that $\|\boldsymbol{b}_0(\boldsymbol{x_1}, t)\| < K\|\boldsymbol{x_1}\| + c$ and $\|\boldsymbol{g}_0(\boldsymbol{x_1}, t)\| < K'\|\boldsymbol{x_1}\| + c'$ for all $t \in [0, t_I]$.*

**Assumption B.2.** *At each time step $t_i$, the distribution of the $N_i$ particles has bounded second moments. Moreover, the initial distribution $\pi_0$ also has bounded second moments.*

## B.3   PROOF OF PROPOSITION 3.1

In this section, we prove our main proposition from the main text, Proposition 3.1.

*Proof of Proposition 3.1.* We start with the definition of the MMD squared between the joint distributions:

$$
\begin{aligned}
\mathrm{MMD}^2_K(f(\boldsymbol{y}, t), g(\boldsymbol{y}, t)) = {}& \mathbb{E}_{(\boldsymbol{y}, t) \sim f, (\boldsymbol{y}', t') \sim f}\left[K((\boldsymbol{y}, t), (\boldsymbol{y}', t'))\right] \\
& - 2\mathbb{E}_{(\boldsymbol{y}, t) \sim f, (\boldsymbol{y}', t') \sim g}\left[K((\boldsymbol{y}, t), (\boldsymbol{y}', t'))\right] \\
& + \mathbb{E}_{(\boldsymbol{y}, t) \sim g, (\boldsymbol{y}', t') \sim g}\left[K((\boldsymbol{y}, t), (\boldsymbol{y}', t'))\right]
\end{aligned} \tag{A5}
$$

The boundedness assumptions ensure that all these kernel expectations are finite, so that the MMD and the decomposition above are well-defined. Next we can rewrite the first term in right-hand side as follows:

$$
\begin{aligned}
\mathbb{E}_{(\boldsymbol{y}, t) \sim f, (\boldsymbol{y}', t') \sim f}\left[K((\boldsymbol{y}, t), (\boldsymbol{y}', t'))\right] &= \mathbb{E}_{(\boldsymbol{y}, t) \sim f, (\boldsymbol{y}', t') \sim f}[K_{\boldsymbol{y}}(\boldsymbol{y}, \boldsymbol{y}')\, \delta(t - t')] \\
&= \mathbb{E}_{t \sim h(t), t' \sim h(t')}\left[\delta(t - t')\, \mathbb{E}_{\substack{\boldsymbol{y} \sim f(\boldsymbol{y}|t) \\ \boldsymbol{y}' \sim f(\boldsymbol{y}|t')}}[K_{\boldsymbol{y}}(\boldsymbol{y}, \boldsymbol{y}')]\right] \\
&= \sum_{t, t' \in \mathcal{T}}\left[\delta(t - t')\, \mathbb{E}_{\substack{\boldsymbol{y} \sim f(\boldsymbol{y}|t) \\ \boldsymbol{y}' \sim f(\boldsymbol{y}|t')}}[K_{\boldsymbol{y}}(\boldsymbol{y}, \boldsymbol{y}')]\right] h(t)h(t') \\
&= \sum_{t \in \mathcal{T}} \mathbb{E}_{\boldsymbol{y}, \boldsymbol{y}' \sim f(\boldsymbol{y}|t)}[K_{\boldsymbol{y}}(\boldsymbol{y}, \boldsymbol{y}')]\, h^2(t)
\end{aligned} \tag{A6}
$$

where the first equality uses the factorized form of the kernel, the second equality is by the the law of iterated expectation conditioning on the time components.

Similarly the second term:

$$
\begin{aligned}
-2\mathbb{E}_{(\boldsymbol{y}, t) \sim f, (\boldsymbol{y}', t') \sim g}\left[K((\boldsymbol{y}, t), (\boldsymbol{y}', t'))\right] &= \mathbb{E}_{(\boldsymbol{y}, t) \sim f, (\boldsymbol{y}', t') \sim g}[K_{\boldsymbol{y}}(\boldsymbol{y}, \boldsymbol{y}')\, \delta(t - t')] \\
&= \mathbb{E}_{t \sim h(t), t' \sim h(t')}\left[\delta(t - t')\left(-2\mathbb{E}_{\substack{\boldsymbol{y} \sim f(\boldsymbol{y}|t) \\ \boldsymbol{y}' \sim g(\boldsymbol{y}|t')}}[K_{\boldsymbol{y}}(\boldsymbol{y}, \boldsymbol{y}')]\right)\right] \\
&= \sum_{t, t' \in \mathcal{T}}\left[\delta(t - t')\left(-2\mathbb{E}_{\substack{\boldsymbol{y} \sim f(\boldsymbol{y}|t) \\ \boldsymbol{y}' \sim g(\boldsymbol{y}|t')}}[K_{\boldsymbol{y}}(\boldsymbol{y}, \boldsymbol{y}')]\right)\right] h(t)h(t') \\
&= \sum_{t \in \mathcal{T}} -2\mathbb{E}_{\substack{\boldsymbol{y} \sim f(\boldsymbol{y}|t) \\ \boldsymbol{y}' \sim g(\boldsymbol{y}|t)}}[K_{\boldsymbol{y}}(\boldsymbol{y}, \boldsymbol{y}')]\, h^2(t)
\end{aligned}
$$

$$\tag{A7}$$

The third term

$$\mathbb{E}_{(\boldsymbol{y},t)\sim g,(\boldsymbol{y}',t')\sim f}\left[K((\boldsymbol{y},t),(\boldsymbol{y}',t'))\right] = \mathbb{E}_{(\boldsymbol{y},t)\sim g,\,(\boldsymbol{y}',t')\sim g}[K_{\boldsymbol{y}}(\boldsymbol{y},\boldsymbol{y}')\,\delta(t-t')]$$

$$= \mathbb{E}_{t\sim h(t),\,t'\sim h(t')}\left[\delta(t-t')\,\mathbb{E}_{\substack{\boldsymbol{y}\sim g(\boldsymbol{y}|t)\\ \boldsymbol{y}'\sim g(\boldsymbol{y}|t')}}\left[K_{\boldsymbol{y}}(\boldsymbol{y},\boldsymbol{y}')\right]\right]$$

$$= \sum_{t,t'\in\mathcal{T}}\left[\delta(t-t')\,\mathbb{E}_{\substack{\boldsymbol{y}\sim g(\boldsymbol{y}|t)\\ \boldsymbol{y}'\sim g(\boldsymbol{y}|t')}}\left[K_{\boldsymbol{y}}(\boldsymbol{y},\boldsymbol{y}')\right]\right]h(t)h(t') \quad \text{(A8)}$$

$$= \sum_{t\in\mathcal{T}}\mathbb{E}_{\boldsymbol{y},\boldsymbol{y}'\sim g(\boldsymbol{y}|t)}[K_{\boldsymbol{y}}(\boldsymbol{y},\boldsymbol{y}')]\,h^2(t)$$

Collecting terms, we have

$$\mathrm{MMD}_K^2(f(\boldsymbol{y},t),g(\boldsymbol{y},t))$$

$$= \sum_{t\in\mathcal{T}}h^2(t)\left[\mathbb{E}_{\boldsymbol{y},\boldsymbol{y}'\sim f(\boldsymbol{y}|t)}[K_{\boldsymbol{y}}(\boldsymbol{y},\boldsymbol{y}')] - 2\mathbb{E}_{\substack{\boldsymbol{y}\sim f(\boldsymbol{y}|t)\\ \boldsymbol{y}'\sim g(\boldsymbol{y}|t)}}[K_{\boldsymbol{y}}(\boldsymbol{y},\boldsymbol{y}')] + \mathbb{E}_{\boldsymbol{y},\boldsymbol{y}'\sim g(\boldsymbol{y}|t)}[K_{\boldsymbol{y}}(\boldsymbol{y},\boldsymbol{y}')]\right]$$

$$= \sum_{t\in\mathcal{T}}h^2(t)\,\mathrm{MMD}_{K_{\boldsymbol{y}}}^2(f(\cdot\mid t),g(\cdot\mid t))$$

$$\square$$

In our application, we used the empirical distribution for time, i.e., $\hat{h}(t)$ from Eq. (2). By doing so, the two distributions of interest to apply Proposition 3.1 are $f_{\boldsymbol{\theta}}(y,t)$ and $\hat{f}(y,t)$, and we can rewrite the squared MMD as

$$\mathrm{MMD}_K^2(f_{\boldsymbol{\theta}},\hat{f}) = \sum_{i=1}^{I}\left(\frac{N_i}{\sum_{j=1}^{I}N_j}\right)^2 \mathrm{MMD}_{K_{\boldsymbol{y}}}^2(f_{\boldsymbol{\theta}}(\cdot|t_i),\hat{f}(\cdot;t_i)).$$

as noted in the main text.

## C  ON I.I.D. ASSUMPTIONS ACROSS TIMEPOINTS

Our use of MMD on the joint distribution $p(Y,t)$ assumes that the dataset of pairs $(Y_{t_i}^n, t_i)$ can be treated as i.i.d. samples. Within each timepoint $t_i$, the measurements $Y_{t_i}^n$ are modeled as i.i.d. from the conditional distribution $p(Y\mid t_i)$. Across timepoints $t_i\neq t_j$, the conditional distributions may differ, but by treating the time variable itself as a discrete random variable drawn from a distribution $h(t)$, the joint pairs $(Y_{t_i}^n, t_i)$ become i.i.d. from $p(Y,t)=p(Y\mid t)h(t)$.

This modeling assumption is a tractable approximation. In practice, dependencies may arise: for example, if multiple cells derive from a shared lineage, or if experimental design enforces balanced sample counts per timepoint. Similarly, in particle simulations with deterministic observation times, samples may be structured rather than independent. Real-world datasets (e.g., single-cell RNA-seq) also exhibit technical artifacts such as batch effects. While these considerations mean that the i.i.d. assumption is not exact, it is widely adopted in the literature and provides a practical working model.

## D  ON THE COMPUTATIONAL ASPECTS OF MMD

In this appendix, we briefly discuss the computational reasons for adopting MMD as the discrepancy measure in our framework.

**Computational form of the loss.** The MMD loss in Eq. (4) reduces to a closed-form, quadratic expression over sample pairs. For a batch of $N$ samples, the cost scales as $O(N^2)$, and this can be reduced to $O(N)$ using low-rank kernel approximations or random Fourier features. The resulting gradients are straightforward to compute through the SDE solver, either via the stochastic adjoint method or standard automatic differentiation.

**Scaling with dimensionality.** MMD remains computationally and statistically robust in moderately high dimensions. The kernel can be applied component-wise or over lower-dimensional embeddings, and the U-statistic estimator used for MMD achieves a standard convergence rate independent of dimension. In contrast, Wasserstein distances exhibit poor sample complexity in high-dimensional spaces, making them less practical for datasets such as PBMC, which have embeddings of dimension 30.

**Implementation.** One more reason to use MMD is that — from a practical standpoint — MMD is simple to implement and widely supported in standard ML libraries. In our code, we use a simple implementation in `PyTorch` (Paszke et al., 2019).

## E    ALTERNATIVE VIEW OF $R^2$

An alternative way to view this metric is through the lens of comparing joint distributions to the product of their marginals. In information theory, mutual information quantifies the dependence between two random variables by measuring the divergence (typically via the Kullback–Leibler divergence) between the joint distribution and the product of the marginal distributions. Analogously, by reapplying Proposition 3.1, we can interpret our $R^2$ metric as comparing the joint distribution of state and time as predicted by the model with the distribution obtained by taking the product of the marginal (state and time) distributions. In this view, the denominator in Eq. (6) (which uses the barycenter) reflects the total variation or "spread" in the observed data. And the numerator captures the remaining error when the model-predicted joint distribution is compared to the empirical joint distribution. Thus, a higher $R^2$ indicates that the model captures more of the dependence structure between state and time—just as in regression a higher $R^2$ means the model explains a larger fraction of the variability in the data. This analogy to mutual information provides an intuitive understanding of how our metric not only assesses goodness-of-fit but also the degree to which the model captures the temporal structure of the data.

## F    FURTHER EXPERIMENTAL DETAILS

In this section, we provide additional details and results for our experiments. First we provide more details about the vector field reconstruction task. Then, we provide the summary table for the forecasting and interpolation task for all experiments using EMD. After that, for each experiment introduced in the main text, we describe: (1) the experimental setup, (2) the choice of model family used with our method, (3) forecasting results, (4) vector field reconstruction results, and (5) interpolation results. Our experiments are carried out using four cores of Intel Xeon Gold 6248 CPU and one Nvidia Volta V100 GPU with 32 GB RAM.

### F.1    VECTOR FIELD RECONSTRUCTION

In cases where the true underlying drift function is known (e.g., synthetic experiments), we also evaluate how accurately methods reconstruct the vector field driving the system dynamics. We use the same baselines as in the forecasting task, since this task also requires recovering a coherent forward-time dynamic, rather than just interpolating between marginals. We measure reconstruction accuracy visually and by computing mean squared error (MSE) between the learned drift and ground truth drift on a dense grid covering the observed data range. Overall, our method provides better or matching vector field reconstruction performance relative to competitors. Detailed results for vector field reconstruction are presented in Appendix F.5.4 (for Lotka-Volterra), Appendix F.6.4 (for repressilator with parametric model family), Appendix F.7.4 (for repressilator with semiparametric model family), and Appendix F.9.5 (for Gulf of Mexico).

### F.2    HOW WE USE $R^2$ IN OUR EXPERIMENTS

In this section, we briefly describe how the proposed $R^2$ metric is used in our experiments. The $R^2$ metric defined in Eq. (6) naturally provides a standardized and interpretable criterion to compare candidate SDE models, guiding model selection and diagnosing fit quality. Values of $R^2$ close to 1 indicate a high-quality fit, while values near or below 0 signal poor model performance relative to the simple barycenter model. Finally, note that $R^2$ is always upper bounded by 1 due to the non-negativity of the MMD, and it may become negative if the candidate model performs worse than the barycenter, analogous to regression models without an intercept. In our experiments, we apply it in two main ways. First, for early stopping: we select the number of training epochs such that $R^2$ increases by less than 0.01 over the last 20 epochs. Second, as a model selection criterion

when choosing among neural network architectures. In particular, for real-data experiments using multilayer perceptrons, we perform a small grid search over the number of layers and hidden units, selecting the model with the highest $R^2$. Specific details for each experiment are provided in the "Model family choice" subsections in Appendix F.

### F.3 EMD TABLE WITH SUMMARY RESULTS ACROSS ALL METHODS AND EXPERIMENTS

In this section, we provide the summary table (Table 2) for comparing forecasting and interpolation performance in terms of EMD over all the experiments of interest. For the interpolation results, we report the mean and standard deviation for each method, aggregated over different random seeds and interpolation held-out time points. Detailed per-time-point results are provided for each experiment in the corresponding "Interpolation results" subsubsection later in Appendix F.

Table 2: EMD for forecast and interpolation tasks. Bold green = best method; plain green = method whose mean lies within one standard deviation of the best method.

| | Forecast | | | Interpolation | | |
|---|---|---|---|---|---|---|
| **Task** | **Ours** | SBIRR-ref | SB-forward | **Ours** | SBIRR | DMSB |
| LV | **0.21 ± 0.04** | 0.79 ± 0.05 | 1.82 ± 0.9 | **0.06 ± 0.03** | 0.10 ± 0.08 | 0.82 ± 0.6 |
| ReprParam | **0.19 ± 0.08** | 1.55 ± 0.8 | 1.39 ± 0.6 | **0.10 ± 0.04** | 0.17 ± 0.06 | 1.89 ± 0.9 |
| ReprSemiparam | **0.35 ± 0.09** | 1.18 ± 0.4 | 1.16 ± 0.3 | **0.21 ± 0.11** | 0.39 ± 0.1 | 1.89 ± 0.9 |
| ReprProtein | **0.26 ± 0.04** | 6.36 ± 0.5 | 7.24 ± 0.5 | **0.08 ± 0.03** | 0.59 ± 0.8 | 1.48 ± 1.0 |
| GoM | **0.71 ± 0.01** | 0.89 ± 0.03 | 0.94 ± 0.08 | 0.10 ± 0.04 | **0.04 ± 0.01** | 0.08 ± 0.04 |
| PBMC | – | – | – | – | – | – |

### F.4 FULL RESULTS FOR INTERPOLATION TASK

In this section, we provide summary tables for all methods for the interpolation task. In Table 3 we provide the summary table for MMD. In Table 4 we provide the summary table for EMD. As in Appendix F.3, we report the mean and standard deviation for each method, aggregated over different random seeds and interpolation held-out time points. We find that in almost all experiments, SnapMMD provides better or matching interpolation performance relative to competitors.

Table 3: Global summary of MMD across all seeds and validation points

| Task | Ours | SBIRR | DMSB | OT-CFM | SB-CFM | SF2M |
|---|---|---|---|---|---|---|
| LV | **0.013 ± 0.011** | 0.048 ± 0.082 | 3.316 ± 2.175 | 2.830 ± 2.929 | 2.099 ± 2.245 | 2.055 ± 1.915 |
| ReprParam | **0.040 ± 0.034** | 0.156 ± 0.102 | 7.269 ± 2.001 | 4.639 ± 2.123 | 3.500 ± 1.878 | 3.327 ± 1.157 |
| ReprSemiparam | **0.379 ± 0.381** | 1.105 ± 0.815 | 7.269 ± 2.001 | 4.639 ± 2.123 | 3.500 ± 1.878 | 3.327 ± 1.157 |
| ReprProtein | **0.011 ± 0.006** | 1.480 ± 2.953 | 5.147 ± 2.630 | 4.092 ± 3.086 | 2.878 ± 2.643 | 3.091 ± 2.582 |
| Gulf of Mexico | 0.073 ± 0.054 | **0.008 ± 0.009** | 0.053 ± 0.064 | 0.651 ± 0.443 | 0.693 ± 0.564 | 0.624 ± 0.406 |
| PBMC | **0.114 ± 0.052** | 0.114 ± 0.122 | 0.968 ± 0.095 | 0.440 ± 0.195 | 0.488 ± 0.198 | 0.343 ± 0.094 |

Table 4: Global summary of EMD across all seeds and validation points. For the repressilator with incomplete state measurements, for some random seeds the trajectories generated by SB-CFM and SF2M diverged significantly, leading to extremely large average EMD values (1138.222 ± 8653.586 for SB-CFM and 1954.050 ± 14737.310 for SF2M). To maintain visualization clarity, we omit these entries from the summary table.

| Task | Ours | SBIRR | DMSB | OT-CFM | SB-CFM | SF2M |
|---|---|---|---|---|---|---|
| LV | **0.064 ± 0.031** | 0.096 ± 0.075 | 0.819 ± 0.559 | 0.742 ± 0.698 | 0.669 ± 0.674 | 0.770 ± 0.706 |
| ReprParam | **0.096 ± 0.041** | 0.168 ± 0.057 | 1.887 ± 0.865 | 1.367 ± 0.793 | 1.412 ± 1.377 | 23.201 ± 122.101 |
| ReprProtein | **0.080 ± 0.032** | 0.591 ± 0.783 | 1.480 ± 1.011 | 1.328 ± 1.026 | – | – |
| ReprSemiparam | **0.210 ± 0.106** | 0.394 ± 0.136 | 1.887 ± 0.865 | 1.367 ± 0.793 | 1.412 ± 1.377 | 23.201 ± 122.101 |
| Gulf of Mexico | 0.103 ± 0.040 | **0.043 ± 0.012** | 0.078 ± 0.037 | 0.252 ± 0.109 | 0.265 ± 0.136 | 0.249 ± 0.102 |
| PBMC | – | – | – | – | – | – |

## F.5 LOTKA-VOLTERRA

### F.5.1 EXPERIMENT SETUP

In this experiment, we study the stochastic Lotka-Volterra model, which describes predator-prey interactions over time. The population dynamics are governed by the following system of SDEs:

$$
\begin{aligned}
dX &= \alpha X - \beta XY + \sigma X dW_x, \\
dY &= \gamma XY - \delta Y + \sigma Y dW_y,
\end{aligned}
\tag{A9}
$$

where $[dW_x, dW_y]$ denotes a two-dimensional Brownian motion. The true parameter values are set to $\alpha = 1.0$, $\beta = 0.4$, $\gamma = 0.4$, $\delta = 0.1$, and $\sigma = 0.02$. Initial population sizes are sampled from uniform distributions: $X_0 \sim U(5, 5.1)$ and $Y_0 \sim U(4, 4.1)$. We simulate the system over 19 discrete time points using the Euler–Maruyama method (via the `torchsde` Python package), with a time step of 0.5 and 200 samples per time point. We use the 10 odd-numbered time steps as training data. The 9 even-numbered time steps are held out for evaluating interpolation performance. To assess forecasting, we simulate one additional time step beyond the final snapshot, using a larger time increment of 1.0, and hold it out as the test point.

### F.5.2 MODEL FAMILY CHOICE

For this experiment, we have access to the data-generating process, as described in Eq. (A9). Therefore, we select the model family to be the set of SDEs that satisfy this system of equations, Eq. (A9) with free parameters $\alpha, \beta, \gamma, \delta, \sigma > 0$. We learn the parameters by minimizing the proposed MMD loss using gradient descent, with a learning rate of 0.05 over 300 epochs. We choose the number of epochs such that in the last 20 epochs $R^2$ increases by less than 0.01. We implemented the model family as a Python class using in the `torchsde` (Li et al., 2020) module.

### F.5.3 FORECASTING RESULTS.

In the first row of Table 5 we show the MMD results for the forecast task. The MMD is computed using a RBF kernel with length scale 1. In each cell, the first number represent the MMD averaged across 10 different seeds, and the second number (in parenthesis) is the standard deviation over the same 10 seeds. We color in green the cell corresponding to the method with lowest MMD. We also highlight in green any other methods whose mean is contained in the one-standard deviation confidence interval for the best method. From the first row, we can see how our method is (by far) the best method at the forecasting task. In the second row, we have the same set of results, using EMD instead of MMD. We can see that our method is, also using EMD, by far the best method at the forecasting task.

### F.5.4 VECTOR FIELD RECONSTRUCTION RESULTS.

If we look at the middle row of Fig. A5, we see that from a visual perspective the reconstructed vector fields are very similar to the ground truth for all the three methods. Also from the bottom row of the same figure, we can see that the difference between the reconstructed fields and ground truth for our method and `SBIRR-ref` is very similar, whereas for `SB-forward` it is a bit worse. The same intuition is confirmed by looking at the bottom row in Table 5 where we compare the MSEs for the vector reconstruction task, and our method achieves the lowest value.

Table 5: Evaluation metric for Lotka-Volterra (mean (sd)). Drift was evaluated using MSE on a grid.

| Metric | | LV | |
| --- | --- | --- | --- |
| | Ours | SBIRR-ref | SB-forward |
| Forecast-MMD | **0.057 (0.03)** | 0.69 (0.11) | 2.99 (1.88) |
| Forecast-EMD | **0.21 (0.035)** | 0.79 (0.053) | 1.82 (0.94) |
| Drift | **0.00071 (0.000027)** | 0.079 (0.0080) | 0.59 (0.13) |

### F.5.5 INTERPOLATION RESULTS

We evaluate model performance on the interpolation task in the classic LV experiment by comparing both qualitative and quantitative results. Specifically, we assess the quality of inferred trajectories against held-out validation snapshots using MMD and EMD. In Fig. A6, the held-out validation snapshots are indicated by x-markers. An interpolation method is successful when its learned trajectories intersect these markers. For visual clarity, we omit the training snapshots: they fall between consecutive validation times and would excessively clutter the figure without adding interpretive

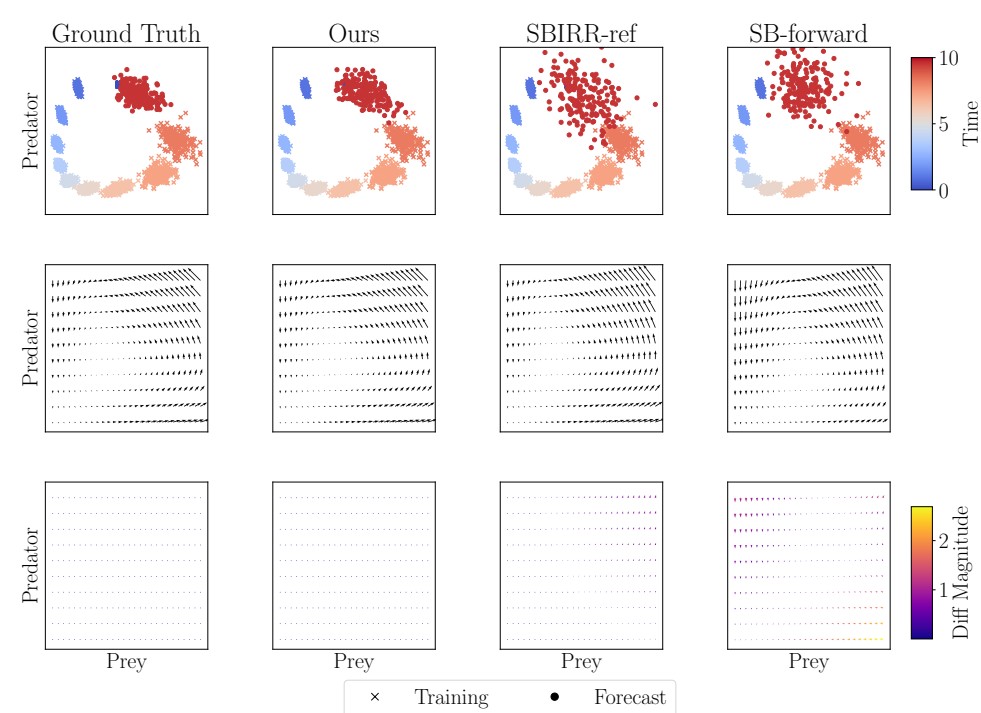

Figure A5: Experimental results for the Lotka-Volterra system. *Top row*: forecast prediction task. A method is successful if the forecast predicted points (in red) match the red points in the ground truth figure. *Middle row:* ground truth vector field (left) and reconstructed vector fields with the three methods. *Bottom row:* Difference between reconstructed vector fields and ground truth. For each point of interest on the grid, we represent the difference between the two vectors with an arrow and color it according to the magnitude of the difference (colorbar to the right).

value. As shown in Fig. A6, our method produces trajectories that interpolate closely the held-out data. Among all baselines, SBIRR and the two CFM achieve comparable visual match. This visual impression is also supported by the quantitative metrics. In Fig. A7, we plot MMD and EMD values over all validation time points. Our method consistently achieves the lowest values across time in both metrics. SBIRR performs comparably well at some validation points. In contrast, the other baselines show significantly higher errors, particularly in later time steps where the trajectory distribution becomes more complex. The corresponding tables (Table 6 and Table 7) confirm these trends. For every validation point, our method is always as good as the best baseline (SBIRR) and in most of the cases it achieves the lowest MMD and EMD values.

Table 6: MMD at each validation point for Lotka-Volterra.

| Time | Ours | SBIRR | DMSB | OT-CFM | SB-CFM | SF2M |
|---|---|---|---|---|---|---|
| 0.5 | **0.010 ± 0.003** | 0.016 ± 0.010 | 2.173 ± 0.238 | 0.188 ± 0.038 | 0.187 ± 0.027 | 0.172 ± 0.026 |
| 1.5 | **0.006 ± 0.004** | 0.014 ± 0.013 | 2.170 ± 0.871 | 0.197 ± 0.054 | 0.203 ± 0.053 | 0.190 ± 0.030 |
| 2.5 | **0.005 ± 0.003** | 0.011 ± 0.007 | 0.462 ± 0.364 | 0.366 ± 0.066 | 0.399 ± 0.102 | 0.348 ± 0.045 |
| 3.5 | 0.015 ± 0.007 | **0.007 ± 0.005** | 2.277 ± 1.094 | 0.501 ± 0.233 | 0.438 ± 0.146 | 0.433 ± 0.091 |
| 4.5 | **0.010 ± 0.007** | 0.015 ± 0.016 | 2.974 ± 0.979 | 1.254 ± 0.838 | 0.850 ± 0.392 | 1.126 ± 0.338 |
| 5.5 | **0.017 ± 0.010** | 0.019 ± 0.006 | 2.814 ± 0.969 | 3.913 ± 1.804 | 2.858 ± 1.257 | 3.389 ± 0.689 |
| 6.5 | **0.014 ± 0.009** | 0.020 ± 0.012 | 3.578 ± 1.221 | 6.755 ± 1.498 | 5.059 ± 1.679 | 5.010 ± 0.702 |
| 7.5 | **0.016 ± 0.021** | 0.080 ± 0.058 | 6.330 ± 0.995 | 6.322 ± 1.470 | 4.006 ± 1.880 | 4.044 ± 0.869 |
| 8.5 | **0.020 ± 0.015** | 0.249 ± 0.084 | 7.066 ± 0.778 | 5.970 ± 1.508 | 4.895 ± 1.414 | 3.787 ± 0.502 |

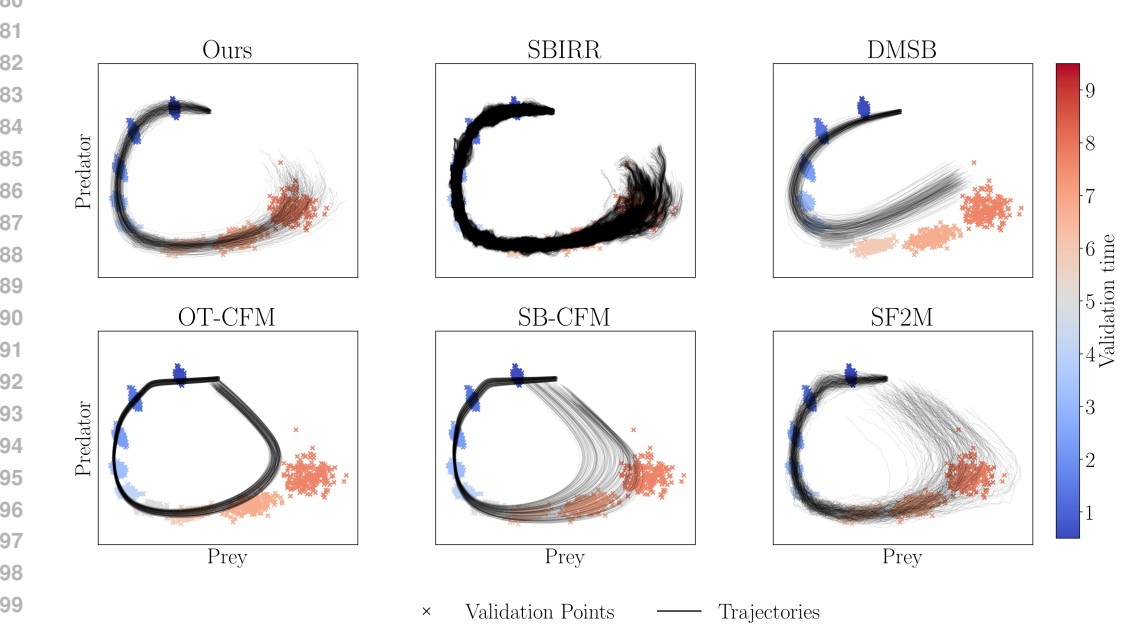

Figure A6: LV interpolation

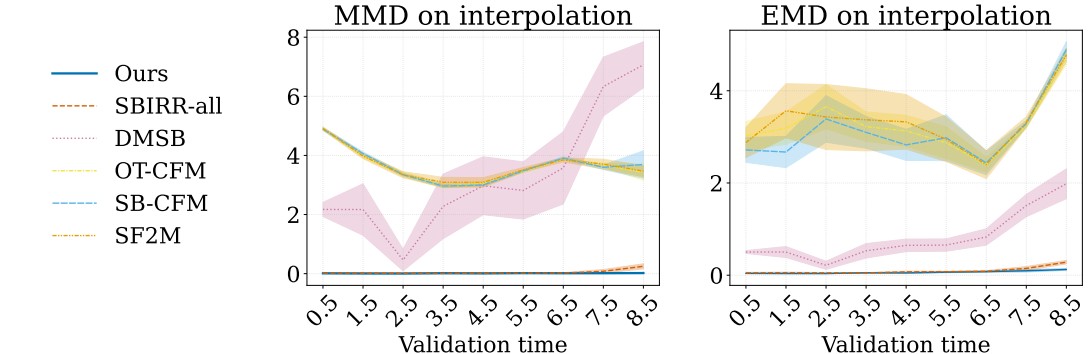

Figure A7: LV interpolation metrics

Table 7: EMD at each validation point for Lotka-Volterra.

| Time | Ours | SBIRR | DMSB | OT−CFM | SB−CFM | SF2M |
|------|------|-------|------|--------|--------|------|
| 0.5 | **0.040 ± 0.005** | 0.050 ± 0.011 | 0.503 ± 0.031 | 0.147 ± 0.014 | 0.145 ± 0.010 | 0.139 ± 0.010 |
| 1.5 | **0.037 ± 0.006** | 0.053 ± 0.012 | 0.503 ± 0.119 | 0.157 ± 0.017 | 0.155 ± 0.017 | 0.149 ± 0.010 |
| 2.5 | **0.036 ± 0.005** | 0.050 ± 0.009 | 0.215 ± 0.088 | 0.207 ± 0.018 | 0.213 ± 0.026 | 0.198 ± 0.012 |
| 3.5 | 0.052 ± 0.006 | **0.049 ± 0.007** | 0.529 ± 0.157 | 0.236 ± 0.050 | 0.222 ± 0.036 | 0.223 ± 0.023 |
| 4.5 | **0.049 ± 0.008** | 0.073 ± 0.014 | 0.647 ± 0.135 | 0.382 ± 0.133 | 0.321 ± 0.072 | 0.383 ± 0.069 |
| 5.5 | **0.065 ± 0.012** | 0.076 ± 0.010 | 0.653 ± 0.134 | 0.795 ± 0.254 | 0.654 ± 0.188 | 0.841 ± 0.173 |
| 6.5 | **0.077 ± 0.012** | 0.086 ± 0.014 | 0.829 ± 0.177 | 1.537 ± 0.479 | 1.244 ± 0.439 | 1.652 ± 0.376 |
| 7.5 | **0.096 ± 0.025** | 0.148 ± 0.040 | 1.510 ± 0.241 | 1.705 ± 0.586 | 1.246 ± 0.594 | 1.751 ± 0.477 |
| 8.5 | **0.124 ± 0.019** | 0.283 ± 0.037 | 1.983 ± 0.323 | 1.511 ± 0.454 | 1.825 ± 0.662 | 1.594 ± 0.298 |

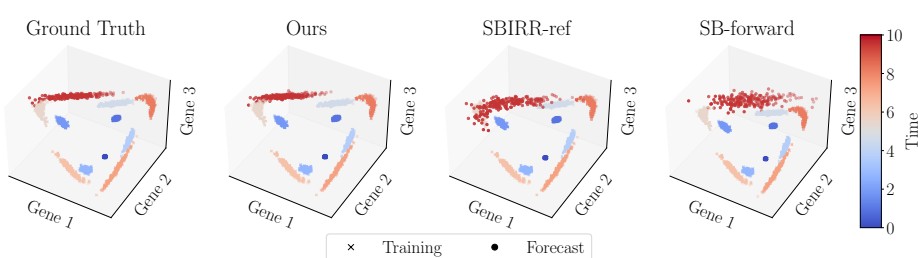

Figure A8: Forecasting task for the repressilator system with parametric model family.

### F.6 mRNA-ONLY REPRESSILATOR WITH PARAMETRIC FAMILY

#### F.6.1 EXPERIMENT SETUP

The repressilator is a synthetic genetic circuit designed to function as a biological oscillator, producing sustained periodic fluctuations in the concentrations of its components. It consists of a network of three genes arranged in a cyclic inhibitory loop: each gene encodes a protein that suppresses the expression of the next, with the last gene repressing the first, completing the feedback cycle.

The system's dynamics can be described by the following stochastic differential equations (SDEs):

$$dX_1 = \frac{\beta}{1 + (X_3/k)^n} - \gamma X_1 + \sigma X_1 dW_1,$$

$$dX_2 = \frac{\beta}{1 + (X_1/k)^n} - \gamma X_2 + \sigma X_2 dW_2, \quad (A10)$$

$$dX_3 = \frac{\beta}{1 + (X_2/k)^n} - \gamma X_3 + \sigma X_3 dW_3,$$

where $[dW_1, dW_2, dW_3]$ represents a three-dimensional Brownian motion. The inhibitory structure of the system is evident from the drift terms, which describe how each gene's expression is repressed by another in the cycle. For our simulations, we set the parameters to $\beta = 10$, $n = 3$, $k = 1$, $\gamma = 1$, and $\sigma = 0.02$. The initial conditions are sampled from uniform distributions: $X_1, X_2 \sim U(1, 1.1)$ and $X_3 \sim U(2, 2.1)$. To simulate the system, we numerically integrate the SDEs over 19 discrete time points, with sampling rate 0.5 with the Euler-Maruyama scheme(implemented via the `torchsde` Python package) with 200 samples at each step. Out of these 19 time steps, we use the 10 odd-numbered time steps as training data. The 9 even-numbered time steps are held out for evaluating interpolation performance. To assess forecasting, we simulate one additional time step beyond the final snapshot, using a larger time increment of 1.0, and hold it out as the test point.

#### F.6.2 MODEL FAMILY CHOICE

For this experiment, we have access to the data-generating process, as described in Eq. (A10). Therefore, we pick as a model family the set of SDEs that satisfy this system of equations, Eq. (A10). The learning process involves optimizing the parameters using gradient descent, with a learning rate of 0.05 over 500 epochs. We choose this number of epochs such that in the last 20 epochs $R^2$ increases by less than 0.01. We implemented the model family as a Python class using in the `torchsde` (Li et al., 2020) module.

#### F.6.3 FORECASTING RESULTS.

In this section we further discuss results for the repressilator experiment with parametric model family. In particular, we analyze the EMD and MMD in Table 8.

In the first row of Table 8, we see that for the forecasting task, our method achieves a much lower MMD compared to the two baselines. This quantitatively supports the visual intuition from Fig. A8, where our approach more accurately captures the underlying distribution of the data. In the second row, we see that also using EMD our method significantly outperforms all the baselines.

#### F.6.4 VECTOR FIELD RECONSTRUCTION RESULTS.

In the third row of Table 8, we observe that the MSE for the vector field reconstruction task is significantly lower for our method, indicating superior performance in recovering the true dynamics.

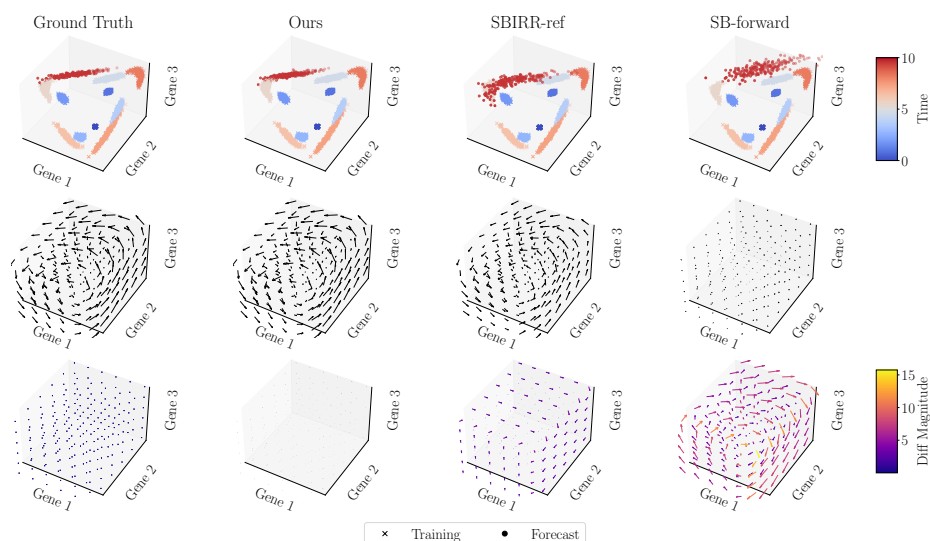

Figure A9: Experimental results for the repressilator system using parametric model as model family.*Top row*: forecast prediction task. A method is successful if the forecast predicted points (in red) match the red points in the ground truth figure. *Middle row:* ground truth vector field (left) and reconstructed vector fields with the three methods. *Bottom row:* Difference between reconstructed vector fields and ground truth. For each point of interest on the grid, we represent the difference between the two vectors with an arrow and color it according to the magnitude of the difference (colorbar to the right).

Table 8: Evaluation metric for repressilator when using the parametric model (mean(sd)). Drift was evaluated using MSE on a grid.

| | Repressilator (parametric) | | |
| Metric | Ours | SBIRR-ref | SB-forward |
|---|---|---|---|
| Forecast-MMD | **0.072 (0.080)** | 2.06 (1.34) | 2.09 (0.74) |
| Forecast-EMD | **0.19 (0.075)** | 1.55 (0.79) | 1.39 (0.62) |
| Drift | **0.027 (0.063)** | 1.71 (0.20) | 12.9 (0.21) |

This is further corroborated by the visualizations in Fig. A9: in the middle row, our reconstructed vector field closely resembles the ground truth, whereas SBIRR-ref exhibits small but notable deviations, and SB-forward fails both in magnitude and direction. The bottom row further reinforces this conclusion, showing that the magnitude of the differences between the reconstructed and true vector fields is substantially larger for the two baselines compared to our method (for which is very close to 0 everywhere on the grid).

### F.6.5 INTERPOLATION RESULTS

We next assess interpolation performance for the parametric model, again comparing inferred trajectories to held-out snapshots with MMD and EMD. Visual inspection of Fig. A10 shows that our method interpolates all the validation snapshots very closely, understanding the periodic behavior of this system. SBIRR yields a qualitatively similar plot with more noisy trajectories, while DMSB, OT-CFM, SB-CFM, and SF2M provide poor interpolations — OT-CFM fails because they are just pairwise interpolation methods and so they "connect" training points, missing the long-term behavior of the system; DMSB, SB-CFM, and SF2M fail because they start drifting outward and end up very far from the actual validation points.

The metric curves in Fig. A11 and tables Table 9–Table 10 corroborate these impressions. Across all validation times, our method always achieves the lowest MMD and EMD, whereas the second best method (SBIRR) is comparable to ours only on one validation time. All the other baselines

achieve worse performance. We note that in Fig. A11 we set the y-axis limit to 5 to show meaningful comparisons. We did this since SF2M's EMD explodes after time 5.5 — peaking at $\approx 138$ at time 8.5 — signalling complete geometric mismatch with the target distribution. We also refrain from highlighting the SF2M cell in green, despite it formally satisfying the coloring criterion. This is because the overlap with the best-performing method arises primarily from SF2M's extremely large mean and standard deviation, rather than from a meaningful proximity in performance.

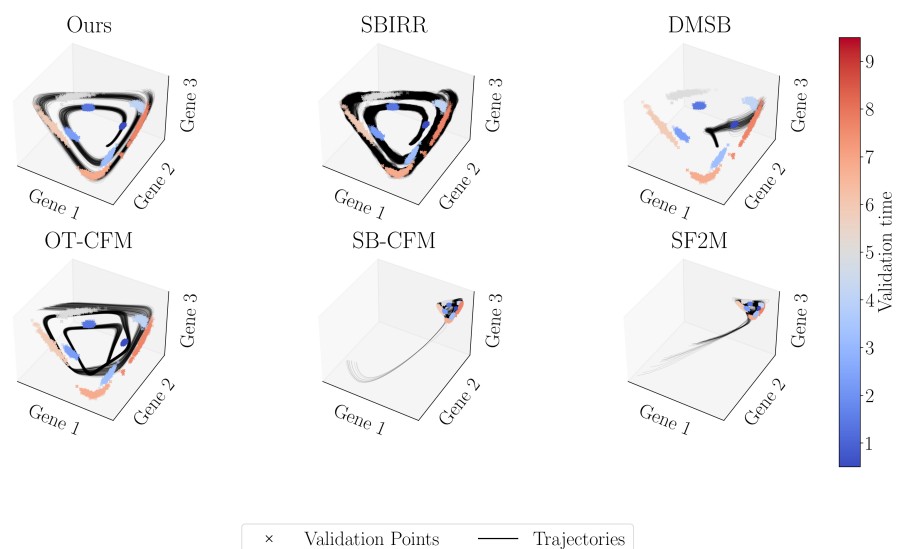

Figure A10: Parametric interpolation

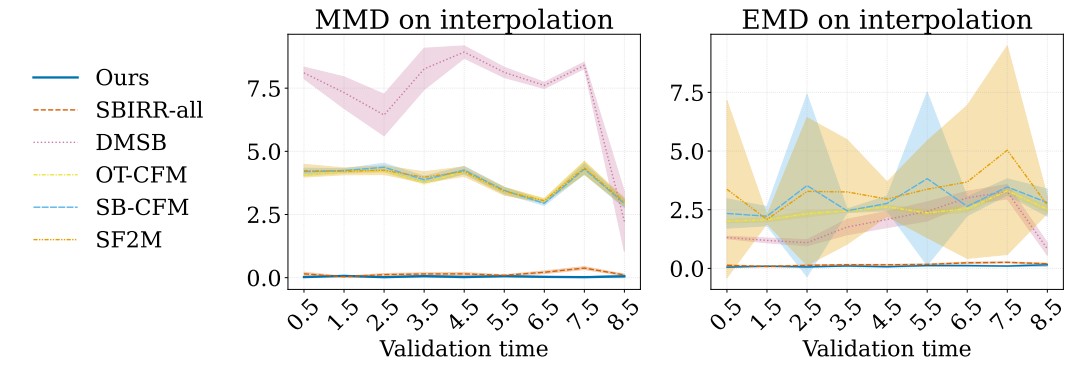

Figure A11: Parametric interpolation metric

Table 9: MMD at each validation point for Repressilator parametric.

| Time | Ours | SBIRR | DMSB | OT−CFM | SB−CFM | SF2M |
|---|---|---|---|---|---|---|
| 0.5 | **0.022 ± 0.004** | 0.151 ± 0.061 | 8.104 ± 0.230 | 2.767 ± 0.137 | 2.768 ± 0.162 | 2.639 ± 0.137 |
| 1.5 | 0.070 ± 0.020 | **0.037 ± 0.024** | 7.324 ± 0.625 | 1.280 ± 0.469 | 1.103 ± 0.498 | 1.267 ± 0.370 |
| 2.5 | **0.019 ± 0.009** | 0.120 ± 0.044 | 6.438 ± 0.823 | 5.468 ± 1.053 | 3.876 ± 0.687 | 3.661 ± 0.531 |
| 3.5 | **0.059 ± 0.036** | 0.148 ± 0.047 | 8.262 ± 0.819 | 6.943 ± 1.067 | 5.393 ± 1.852 | 4.792 ± 0.985 |
| 4.5 | **0.025 ± 0.031** | 0.150 ± 0.054 | 8.929 ± 0.245 | 4.600 ± 1.465 | 2.969 ± 1.475 | 3.633 ± 0.684 |
| 5.5 | **0.060 ± 0.029** | 0.094 ± 0.021 | 8.124 ± 0.204 | 3.818 ± 1.405 | 2.385 ± 1.362 | 2.757 ± 0.528 |
| 6.5 | **0.032 ± 0.019** | 0.214 ± 0.054 | 7.610 ± 0.124 | 5.015 ± 1.699 | 2.926 ± 1.436 | 3.134 ± 0.463 |
| 7.5 | **0.020 ± 0.013** | 0.379 ± 0.058 | 8.404 ± 0.130 | 6.561 ± 1.667 | 5.488 ± 1.984 | 4.473 ± 0.695 |
| 8.5 | **0.053 ± 0.057** | 0.108 ± 0.030 | 2.225 ± 1.179 | 5.297 ± 1.601 | 4.595 ± 0.725 | 3.589 ± 0.616 |

Table 10: EMD at each validation point for Repressilator parametric.

| Time | Ours | SBIRR | DMSB | OT-CFM | SB-CFM | SF2M |
|------|------|-------|------|--------|--------|------|
| 0.5 | **0.052 ± 0.004** | 0.129 ± 0.025 | 1.317 ± 0.053 | 0.578 ± 0.017 | 0.581 ± 0.022 | 0.581 ± 0.021 |
| 1.5 | 0.091 ± 0.013 | **0.080 ± 0.017** | 1.192 ± 0.104 | 0.384 ± 0.075 | 0.436 ± 0.165 | 0.569 ± 0.143 |
| 2.5 | **0.061 ± 0.009** | 0.128 ± 0.021 | 1.095 ± 0.131 | 0.985 ± 0.145 | 0.920 ± 0.164 | 1.023 ± 0.152 |
| 3.5 | **0.102 ± 0.023** | 0.157 ± 0.019 | 1.762 ± 0.330 | 1.424 ± 0.166 | 1.176 ± 0.229 | 1.515 ± 0.416 |
| 4.5 | **0.070 ± 0.021** | 0.153 ± 0.018 | 2.086 ± 0.348 | 1.104 ± 0.316 | 0.984 ± 0.571 | 2.618 ± 1.923 |
| 5.5 | **0.120 ± 0.025** | 0.170 ± 0.020 | 2.428 ± 0.409 | 1.193 ± 0.312 | 1.194 ± 0.986 | 5.505 ± 7.561 |
| 6.5 | **0.118 ± 0.028** | 0.238 ± 0.032 | 3.006 ± 0.279 | 2.068 ± 0.502 | 1.765 ± 1.185 | 14.974 ± 27.778 |
| 7.5 | **0.099 ± 0.018** | 0.257 ± 0.020 | 3.270 ± 0.310 | 2.556 ± 0.842 | 2.306 ± 1.469 | 44.051 ± 96.243 |
| 8.5 | **0.152 ± 0.061** | 0.198 ± 0.021 | 0.828 ± 0.266 | 2.011 ± 0.534 | 3.345 ± 2.295 | 137.973 ± 328.155 |

### F.7 MRNA-ONLY REPRESSILATOR WITH SEMIPARAMETRIC FAMILY

#### F.7.1 EXPERIMENT SETUP

The experimental setup is the same as the one for the repressilator with the parametric model choice, as discussed in Appendix F.6.1.

#### F.7.2 MODEL FAMILY CHOICE

In this experiment, we do not assume that we know the full functional form as in Eq. (A10), but only up to an unknown activation function $f_{\boldsymbol{\theta}} : \mathbb{R}_+^3 \to [0,1]^3$, that encodes the regulation among the three genes. In particular, we consider the following model:

$$d\boldsymbol{X}_t = \boldsymbol{M} f_{\boldsymbol{\theta}}(\boldsymbol{X}_t) - \boldsymbol{L}\boldsymbol{X}_t + \boldsymbol{G}\operatorname{diag}(\boldsymbol{X}_t)d\boldsymbol{W}_t \tag{A11}$$

where $\boldsymbol{M}$ is a diagonal matrix of (positive) maximum production rate, $\boldsymbol{L}$ is a diagonal matrix of (positive) degradation rate, $\boldsymbol{G}$ is a diagonal matrix of (positive) volatilities, all unknown (parameterized by their logarithm). We also parameterize the activation function using an MLP with three hidden layers of [32, 64, 32] hidden neurons each, ReLU activation, and one final sigmoid layer.

#### F.7.3 FORECASTING RESULTS.

For what concerns the experiment with the semiparametric model family, we can see in the first row of Table 11 that also with this model our method achieves a substantially lower MMD compared to the two baselines. This aligns with the visual evidence from Fig. 2 in the main text (top row), where our method's predicted points (in red) more closely match the ground truth. In the second row of Table 11, we see that EMD results are aligned with the MMD ones.

#### F.7.4 VECTOR FIELD RECONSTRUCTION RESULTS.

In the third row of Table 11, we see that for this model choice our method and `SBIRR-ref` achieve similar results, whereas `SB-forward` exhibits much higher MSE. Figure A12 confirms this intuition: our reconstructed vector field and the one for `SBIRR-ref` are quite similar and not too different from the ground truth, whereas `SB-forward` performs particularly poorly, failing to recover both the direction and magnitude of the vector field.

Table 11: Evaluation metric for Repressilator using MLP activation function (mean(sd)). Drift was evaluated using MSE on a grid, while forecast was evaluated using MMD with RBF kernel and length scale 1 as well as EMD.

| Metric | Repressilator (semiparametric) | | |
| | Ours | SBIRR-ref | SB-forward |
|---|---|---|---|
| Forecast-MMD | **0.32 (0.15)** | 1.46 (0.55) | 5.26 (1.66) |
| Forecast-EMD | **0.35 (0.091)** | 1.18 (0.44) | 1.16 (0.33) |
| Drift | **6.25 (0.37)** | **7.85 (1.85)** | 12.00 (0.74) |

#### F.7.5 INTERPOLATION RESULTS

We now evaluate interpolation performance in the more realistic semiparametric setting, where neither our method nor `SBIRR` have access to the true data-generating process. Instead, both methods rely on the same semiparametric reference family from Eq. (A11), introducing a meaningful model mismatch that more closely reflects real-world conditions. As shown in Fig. A13, interpolation quality for our method and `SBIRR` is very similar to the parametric case, as they both are still very good at interpolating all the validation snapshots. The remaining baselines are not affected by this modeling choice, so the trajectories are exactly as in Fig. A10. The quantitative results in Fig. A14 and tables Table 12–Table 13 reinforce these trends. Although all methods exhibit increased error relative to the parametric setting, our method continues to outperform all baselines across nearly all validation times. In terms of MMD, we are the best method at six of nine time points; `SBIRR` performs similarly at three time points, but is consistently worse on the rest. EMD results are even more decisive: our method achieves the lowest error at every time point.

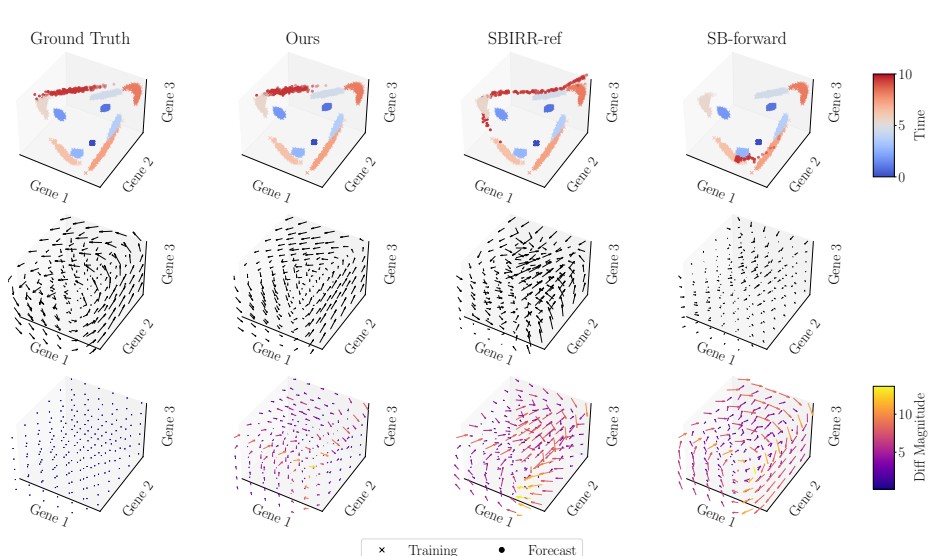

Figure A12: Experimental results for the repressilator system using semiparametric model as model family. *Top row*: forecast prediction task. A method is successful if the forecast predicted points (in red) match the red points in the ground truth figure. *Middle row:* ground truth vector field (left) and reconstructed vector fields with the three methods. *Bottom row:* Difference between reconstructed vector fields and ground truth. For each point of interest on the grid, we represent the difference between the two vectors with an arrow and color it according to the magnitude of the difference (colorbar to the right).

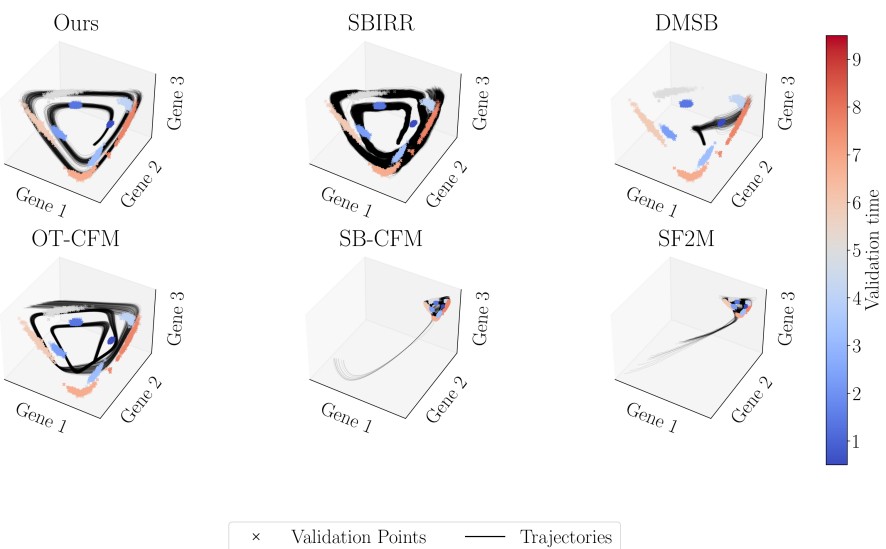

Figure A13: Semiparametric interpolation of repressilator system.

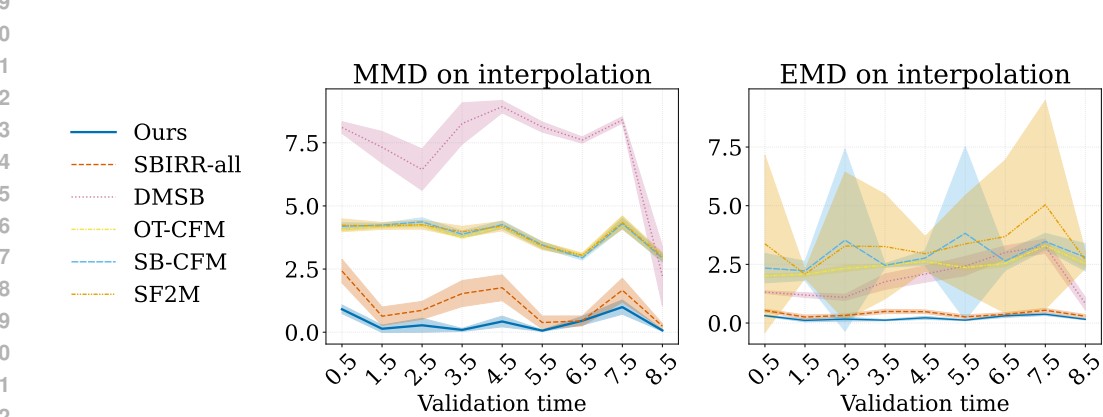

Figure A14: Metrics for Semiparametric interpolation of repressilator system.

Table 12: MMD at each validation point for Repressilator with semiparametric model family.

| Time | Ours | SBIRR | DMSB | OT-CFM | SB-CFM | SF2M |
|------|------|-------|------|--------|--------|------|
| 0.5 | **0.909 ± 0.171** | 2.422 ± 0.470 | 8.104 ± 0.230 | 2.767 ± 0.137 | 2.768 ± 0.162 | 2.639 ± 0.137 |
| 1.5 | **0.137 ± 0.146** | 0.633 ± 0.366 | 7.324 ± 0.625 | 1.280 ± 0.469 | 1.103 ± 0.498 | 1.267 ± 0.370 |
| 2.5 | **0.274 ± 0.277** | 0.865 ± 0.355 | 6.438 ± 0.823 | 5.468 ± 1.053 | 3.876 ± 0.687 | 3.661 ± 0.531 |
| 3.5 | **0.091 ± 0.061** | 1.529 ± 0.515 | 8.262 ± 0.819 | 6.943 ± 1.067 | 5.393 ± 1.852 | 4.792 ± 0.985 |
| 4.5 | **0.424 ± 0.215** | 1.761 ± 0.511 | 8.929 ± 0.245 | 4.600 ± 1.465 | 2.969 ± 1.475 | 3.633 ± 0.684 |
| 5.5 | **0.061 ± 0.031** | 0.383 ± 0.259 | 8.124 ± 0.204 | 3.818 ± 1.405 | 2.385 ± 1.362 | 2.757 ± 0.528 |
| 6.5 | 0.455 ± 0.174 | **0.449 ± 0.197** | 7.610 ± 0.124 | 5.015 ± 1.699 | 2.926 ± 1.436 | 3.134 ± 0.463 |
| 7.5 | **0.994 ± 0.283** | 1.663 ± 0.474 | 8.404 ± 0.130 | 6.561 ± 1.667 | 5.488 ± 1.984 | 4.473 ± 0.695 |
| 8.5 | **0.071 ± 0.035** | 0.241 ± 0.101 | 2.225 ± 1.179 | 5.297 ± 1.601 | 4.595 ± 0.725 | 3.589 ± 0.616 |

Table 13: EMD at each validation point for Repressilator with semiparametric model family.

| Time | Ours | SBIRR | DMSB | OT-CFM | SB-CFM | SF2M |
|------|------|-------|------|--------|--------|------|
| 0.5 | **0.312 ± 0.032** | 0.533 ± 0.061 | 1.317 ± 0.053 | 0.578 ± 0.017 | 0.581 ± 0.022 | 0.581 ± 0.021 |
| 1.5 | **0.111 ± 0.060** | 0.255 ± 0.087 | 1.192 ± 0.104 | 0.384 ± 0.075 | 0.436 ± 0.165 | 0.569 ± 0.143 |
| 2.5 | **0.167 ± 0.082** | 0.322 ± 0.065 | 1.095 ± 0.131 | 0.985 ± 0.145 | 0.920 ± 0.164 | 1.023 ± 0.152 |
| 3.5 | **0.121 ± 0.027** | 0.493 ± 0.091 | 1.762 ± 0.330 | 1.424 ± 0.166 | 1.176 ± 0.229 | 1.515 ± 0.416 |
| 4.5 | **0.222 ± 0.064** | 0.483 ± 0.078 | 2.086 ± 0.348 | 1.104 ± 0.316 | 0.984 ± 0.571 | 2.618 ± 1.923 |
| 5.5 | **0.122 ± 0.023** | 0.263 ± 0.079 | 2.428 ± 0.409 | 1.193 ± 0.312 | 1.194 ± 0.986 | 5.505 ± 7.561 |
| 6.5 | **0.307 ± 0.067** | 0.369 ± 0.075 | 3.006 ± 0.279 | 2.068 ± 0.502 | 1.765 ± 1.185 | 14.974 ± 27.778 |
| 7.5 | **0.374 ± 0.053** | 0.542 ± 0.081 | 3.270 ± 0.310 | 2.556 ± 0.842 | 2.306 ± 1.469 | 44.051 ± 96.243 |
| 8.5 | **0.157 ± 0.023** | 0.290 ± 0.074 | 0.828 ± 0.266 | 2.011 ± 0.534 | 3.345 ± 2.295 | 137.973 ± 328.155 |

## F.8 mRNA-PROTEIN REPRESSILATOR

### F.8.1 EXPERIMENT SETUP

In Appendix F.6.1 we introduced the repressilator system as a system of SDEs governing changes in mRNA concentration. A more complete model for this system takes also into account protein levels. Indeed, each gene produces a protein that represses the next gene's expression, with the last one repressing the first. So proteins play a big role in the repressilator feedback loop. And this is why scientists often consider a more complex version of this system, that evolves according to the following SDEs:

$$dX_1 = \alpha + \frac{\beta}{1 + (Y_3/k)^n} - \gamma X_1 + \sigma X_1 dW_1$$

$$dX_2 = \alpha + \frac{\beta}{1 + (Y_1/k)^n} - \gamma X_2 + \sigma X_2 dW_2$$

$$dX_3 = \alpha + \frac{\beta}{1 + (Y_2/k)^n} - \gamma X_3 + \sigma X_2 dW_3 \tag{A12}$$

$$dY_1 = \beta_p X_1 - \gamma_p Y_1 + \sigma Y_1 dW_4$$

$$dY_2 = \beta_p X_2 - \gamma_p Y_2 + \sigma Y_2 dW_5$$

$$dY_3 = \beta_p X_3 - \gamma_p Y_3 + \sigma Y_3 dW_6$$

where $[dW_1, dW_2, dW_3, dW_4, dW_5, dW_6]$ is a 6D Brownian motion. $X_1, X_2, X_3$ represents the mRNA levels while $Y_1, Y_2, Y_3$ are the corresponding proteins. As explained above, the actual system regulation is now mediated by proteins rather than mRNA themselves.

To obtain data, we fix the following parameters: $\alpha = 10^{-5}, \beta = 10, n = 3, k = 1, \gamma = 1, \beta_p = 1, \gamma_p = 1, \sigma = 0.02$. We start the dynamics with initial distribution $X_1, X_2 \sim U(1, 1.1)$ and $X_3 \sim U(2, 2.1)$, while $Y_i \sim U(0, 0.1)$. We simulate the SDEs for 10 instants of time.

To simulate the system, we numerically integrate the SDEs over 19 discrete time points, with sampling rate 0.5 with the Euler-Maruyama scheme(implemented via the `torchsde` Python package) with 200 samples at each step. Out of these 19, we used 10 odd numbered time steps as training and even numbered steps as validation for interpolation task. We further simulate one step further with time increment of 1 to hold out as test point for forecasting. In all these steps we only took $X_i$ as observations

### F.8.2 MODEL FAMILY CHOICE

**Our method.** For this experiment, we have access to the data-generating process, as described in Eq. (A12). Therefore, we select our model family to be the set of SDEs that satisfy this system of equations, Eq. (A12). We initialize the missing dimensions at all 0. The learning process involves optimizing the parameters using gradient descent, with a learning rate of 0.05 over 500 epochs. We choose this number of epochs such that in the last 20 epochs $R^2$ increases by less than 0.01.

**A Note on Baselines.** Since the two forecasting baseline methods that we consider cannot handle incomplete state observations we cannot use them to fit Eq. (A12). Instead, we fit a simpler mRNA-only model as described in Eq. (A10). We do the same for SBIRR in the interpolation experiment.

### F.8.3 FORECASTING RESULTS

In this section we give more detail on the forecasting results for mRNA-protein repressilator. We provide numerical results in EMD and MMD for forecasting in Table 14. Our method outperform baseline by a large margin, mostly because the correct account of the missing protein observation. Since the two baselines cannot make vector fields in correct dimension we did not compare vector field reconstruction.

Table 14: Evaluation metric for Repressilator forecasting with missing protein observations.

| Metric | Ours | Repressilator (with missing protein) SBIRR-ref | SB-forward |
|---|---|---|---|
| Forecast-MMD | **0.048 (0.029)** | 2.50 (0.05) | 2.42 (0.13) |
| Forecast-EMD | **0.26 (0.042)** | 6.36 (0.51) | 7.24 (0.48) |

### F.8.4 INTERPOLATION RESUTS

We finally consider the interpolation task for this more challenging incomplete state observation setting. As shown in Fig. A15, despite the mismatch between the observed variables and the true system state, our method is able to faithfully reconstruct the trajectories. It successfully captures the geometry of the limit cycle and aligns well with the validation snapshots. SBIRR also performs reasonably, although its trajectories are more dispersed. All remaining baselines fail to track the correct dynamics: DMSB and SF2M exhibit severe trajectory drift, while OT-CFM and SB-CFM overly simplify the structure, failing to represent the circular flow of the system. The lines in Fig. A16 and the numbers in Table 15–Table 16 confirm these findings. Our method consistently achieves the lowest MMD and EMD across all validation times. In contrast, baseline methods show significantly higher EMD and MMD throughout, and their confidence intervals do not overlap with ours unless for SBIRR for EMD in one validation point. As in the previous Repressilator experiment, we cap the y-axis at 5 in Fig. A16 to enable meaningful visual comparisons across methods. This is necessary because both SF2M and SB-CFM exhibit rapidly increasing EMD values after time 3.5. In particular, we observe that for certain random seeds, the inferred trajectories diverge in the wrong direction early on and continue along that path, resulting in large distributional mismatch. Accordingly, we also refrain from highlighting the corresponding cells in green, even when the coloring criterion is formally met, as the overlap with the best method arises from the extremely large variance.

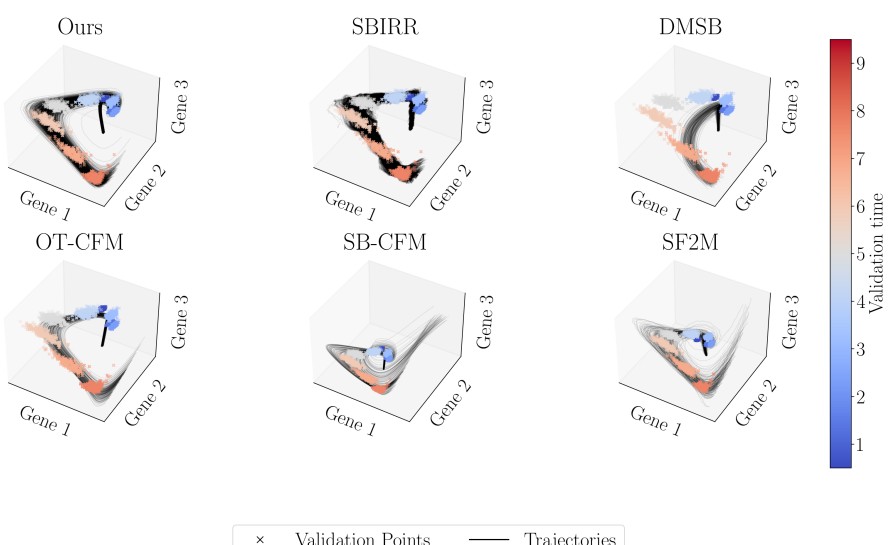

Figure A15: Interpolation of repressilator system with missing protein.

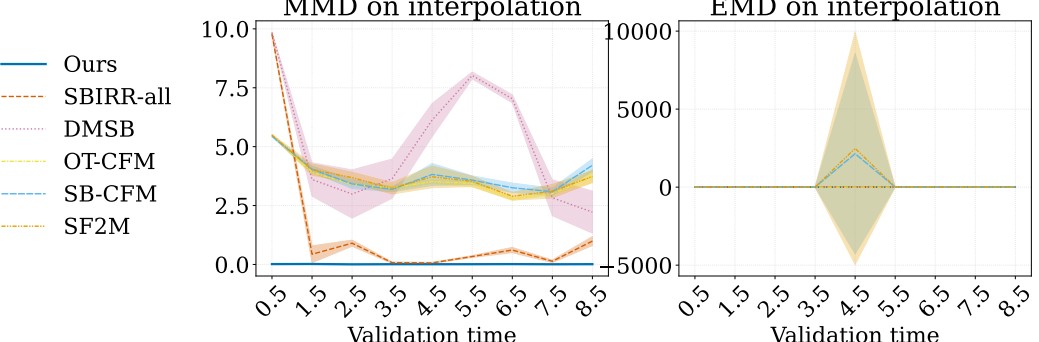

Figure A16: Metrics for Interpolation of repressilator system with missing protein.

Table 15: MMD at each validation point for Repressilator with incomplete state observations.

| Time | Ours | SBIRR | DMSB | OT-CFM | SB-CFM | SF2M |
|------|------|-------|------|--------|--------|------|
| 0.5 | $\mathbf{0.015 \pm 0.003}$ | $9.774 \pm 0.027$ | $9.847 \pm 0.002$ | $9.818 \pm 0.010$ | $9.745 \pm 0.050$ | $9.511 \pm 0.121$ |
| 1.5 | $\mathbf{0.019 \pm 0.002}$ | $0.433 \pm 0.357$ | $3.606 \pm 0.708$ | $0.192 \pm 0.158$ | $0.531 \pm 0.544$ | $0.748 \pm 0.688$ |
| 2.5 | $\mathbf{0.004 \pm 0.001}$ | $0.903 \pm 0.131$ | $2.990 \pm 1.024$ | $0.431 \pm 0.129$ | $0.688 \pm 0.396$ | $0.690 \pm 0.306$ |
| 3.5 | $\mathbf{0.009 \pm 0.002}$ | $0.075 \pm 0.032$ | $3.642 \pm 0.833$ | $1.598 \pm 0.789$ | $1.578 \pm 0.618$ | $1.538 \pm 0.503$ |
| 4.5 | $\mathbf{0.008 \pm 0.004}$ | $0.065 \pm 0.026$ | $6.131 \pm 0.702$ | $3.389 \pm 1.494$ | $2.254 \pm 0.747$ | $2.345 \pm 0.764$ |
| 5.5 | $\mathbf{0.010 \pm 0.003}$ | $0.337 \pm 0.038$ | $8.024 \pm 0.147$ | $3.473 \pm 0.950$ | $2.431 \pm 0.713$ | $2.060 \pm 0.470$ |
| 6.5 | $\mathbf{0.014 \pm 0.008}$ | $0.611 \pm 0.112$ | $7.033 \pm 0.171$ | $6.158 \pm 0.735$ | $2.953 \pm 0.504$ | $3.664 \pm 0.506$ |
| 7.5 | $\mathbf{0.007 \pm 0.005}$ | $0.135 \pm 0.053$ | $2.830 \pm 0.762$ | $6.583 \pm 0.499$ | $2.850 \pm 0.456$ | $3.954 \pm 0.553$ |
| 8.5 | $\mathbf{0.011 \pm 0.006}$ | $0.991 \pm 0.191$ | $2.223 \pm 0.892$ | $5.183 \pm 0.978$ | $2.869 \pm 0.859$ | $3.308 \pm 0.446$ |

Table 16: EMD at each validation point for Repressilator with incomplete state observations.

| Time | Ours | SBIRR | DMSB | OT-CFM | SB-CFM | SF2M |
|------|------|-------|------|--------|--------|------|
| 0.5 | $\mathbf{0.051 \pm 0.003}$ | $2.721 \pm 0.097$ | $3.583 \pm 0.071$ | $2.596 \pm 0.028$ | $2.596 \pm 0.029$ | $2.578 \pm 0.036$ |
| 1.5 | $\mathbf{0.054 \pm 0.002}$ | $0.209 \pm 0.083$ | $0.686 \pm 0.086$ | $0.166 \pm 0.050$ | $0.581 \pm 0.675$ | $0.874 \pm 1.104$ |
| 2.5 | $\mathbf{0.037 \pm 0.001}$ | $0.317 \pm 0.023$ | $0.627 \pm 0.131$ | $0.225 \pm 0.032$ | $1.045 \pm 2.139$ | $1.823 \pm 4.516$ |
| 3.5 | $\mathbf{0.059 \pm 0.002}$ | $0.120 \pm 0.013$ | $0.748 \pm 0.111$ | $0.449 \pm 0.117$ | $3.900 \pm 9.557$ | $7.221 \pm 19.895$ |
| 4.5 | $\mathbf{0.068 \pm 0.004}$ | $0.144 \pm 0.013$ | $1.192 \pm 0.145$ | $0.794 \pm 0.227$ | $16.018 \pm 44.580$ | $30.671 \pm 89.239$ |
| 5.5 | $\mathbf{0.087 \pm 0.005}$ | $0.246 \pm 0.010$ | $2.258 \pm 0.192$ | $0.826 \pm 0.163$ | $72.134 \pm 212.668$ | $136.803 \pm 407.955$ |
| 6.5 | $\mathbf{0.126 \pm 0.011}$ | $0.437 \pm 0.042$ | $2.547 \pm 0.200$ | $2.346 \pm 0.418$ | $354.032 \pm 1056.419$ | $649.221 \pm 1942.159$ |
| 7.5 | $\mathbf{0.120 \pm 0.006}$ | $0.324 \pm 0.090$ | $1.020 \pm 0.136$ | $2.614 \pm 0.291$ | $1691.087 \pm 5066.068$ | $2986.712 \pm 8953.868$ |
| 8.5 | $\mathbf{0.116 \pm 0.005}$ | $0.802 \pm 0.238$ | $0.660 \pm 0.150$ | $1.934 \pm 0.762$ | $8102.608 \pm 24292.997$ | $13770.547 \pm 41303.163$ |

### F.9 CURRENT IN THE GULF OF MEXICO

#### F.9.1 EXPERIMENTAL SETUP

We test our method in fitting and forecasting real ocean-current data from the Gulf of Mexico. We use high-resolution (1 km) bathymetry data from a HYbrid Coordinate Ocean Model (HYCOM) reanalysis[4] (Panagiotis, 2014). This dataset has been in the public domain since it was released by the US Department of Defense. The dataset provides hourly ocean current velocity fields for the region extending from 98°E to 77°E in longitude and from 18°N to 32°N in latitude, covering every day since the first day of, 2001.

We then generate particles following Shen et al. (2025). That is, we took the velocity field in a region where a vortex is observed in June 1st 2024 at 5pm. We then select an initial location near the vortex and uniformly sample 4,400 initial positions within a small radius (0.05) around this point and evolve these particles over 11 steps using the ocean current velocity field. The time step size is 1.0 and left the last time step as validation. At each time point we retain 400 particles. We approximate the velocity at each particle's position using the velocity at the nearest grid point when the particle does not align exactly with a grid point. In addition, between each training time point, we simulate another 9 intermediate steps at middle point between each pair of consecutive training time points, with 400 particle each to test for interpolation.

#### F.9.2 PARTICLE INDEPENDENCE IN THE GULF OF MEXICO EXPERIMENT

In the main text, we model particles at each time point as independent, without assuming continuity of individual trajectories across time. This reflects applications where repeated measurements of the same particle are not available.

For example, if the particles were physical buoys or drifters deployed in the ocean, they could be tracked continuously, and trajectory-based methods would be more appropriate. By contrast, in many remote sensing applications each observation is an image that provides only a distributional snapshot at that time, without identifying or tracking the same particles across times. Oil-spill monitoring from satellite imagery is a common example: each image shows the surface oil distribution but not individual particle paths. Our Gulf of Mexico experiment is designed to mimic this setting and follows the procedure described in Appendix D.5.1 of Shen et al. (2025).

#### F.9.3 MODEL FAMILY CHOICE

We employ a physically motivated model to represent the vortex by combining a Lamb-Oseen vortex — a solution of the two-dimensional viscous Navier-Stokes equations (Saffman, 1995) — with a constant divergence field. The Lamb-Oseen component captures the swirling, rotational dynamics typical of a vortex, while the divergence field is added to account for vertical motion or non-conservative forces that may cause a net expansion or contraction of the flow. In other words, this combined model enables us to represent both the core vortex behavior and the secondary effects influencing particle motion.

Formally, the particle trajectories are modeled by the following family of stochastic differential equations (SDEs):

$$dX = \left[ -\gamma \frac{(Y - y_0)r_v}{(Y - y_0)^2 + (Y - y_0)^2} \left( 1 - \exp\left( \frac{\sqrt{(Y - y_0)^2 + (Y - y_0)^2}}{r_v} \right) \right) + d\frac{X - x_{0,d}}{r_d} \right] dt + \sigma dW_x$$

$$dY = \left[ \gamma \frac{(X - x_0)r_v}{(Y - y_0)^2 + (Y - y_0)^2} \left( 1 - \exp\left( \frac{\sqrt{(Y - y_0)^2 + (Y - y_0)^2}}{r_v} \right) \right) + d(Y - y_{0,d}) \right] dt + \sigma dW_y$$

$$\text{(A13)}$$

In this formulation, the free parameters are:

- Circulation ($\gamma$): Controls the strength of the vortex.

- Vortex length scale ($r_v$): Sets the radial decay of the vortex's influence.

- Vortex center ($x_0, y_0$): Specifies the location of the vortex core.

- Divergence ($d$): Represents the magnitude of the constant divergence field.

---

[4]Dataset available at this link.

- Divergence length scale ($r_d$): Governs the spatial extent of the divergence effect in the x-direction.

- Divergence center ($x_{0,d}, y_{0,d}$): Determines the reference location for the divergence field.

- Volatility ($\sigma$): Captures the stochastic fluctuations in particle motion.

### F.9.4 FORECASTING RESULTS

In this section, we further discuss results for the Gulf of Mexico vortex experiment. In particular, we analyze the EMD, MMD.

Table 17: Evaluation metric for Gulf of Mexico experiment (mean(sd)). Drift was evaluated using MSE on a grid.

| Metric | Gulf of Mexico vortex | | |
| --- | --- | --- | --- |
| | Ours | SBIRR-ref | SB-forward |
| Forecast-MMD | 2.36 (0.11) | **1.41 (0.18)** | 2.59 (0.33) |
| Forecast-EMD | **0.71(0.014)** | 0.89(0.034) | 0.94(0.081) |
| Drift | 0.054 ($7.3 \times 10^{-5}$) | **0.031 (0.00032)** | 0.15 (0.023) |

In the second row of Table 17, we observe that our method achieves the lowest EMD for the forecasting task, indicating the closest match to the ground truth particle distribution. This aligns with the visual results in Fig. 3 from the main text, where our forecasted particles accurately capture the spatial structure of the vortex, unlike the baselines which produce more scattered and less coherent predictions. While the MMD metric (first row) slightly favors the SBIRR-ref baseline, this discrepancy may be attributed to the sensitivity of MMD to particle density and kernel choice.

### F.9.5 VECTOR FIELD RECONSTRUCTION

In the third row of Table 17, we compare the drift reconstruction error using MSE on a grid. Here SBIRR-ref achieves the lowest error, and our method performs comparably and still significantly outperforms SB-forward. Visualizations in Fig. A17 provide further insight: the reconstructed velocity fields from all the three methods exhibit a well-formed vortex structure closely resembling the ground truth (with our method and SBIRR-ref being slightly better, as also shown by the MSE results).

### F.9.6 INTERPOLATION

We now evaluate interpolation performance on the real-world drifter trajectories in the Gulf of Mexico. As shown in Fig. A18, our method captures the overall geometry of the flow and the looping structure of the trajectories. All baselines also succeed in reconstructing the large-scale circulation, with SBIRR achieving the closest match to the held-out validation points. These patterns are reflected in the quantitative metrics in Fig. A19 and tables Table 18–Table 19: across nearly all validation times, SBIRR achieves the lowest MMD and EMD values, consistently outperforming our method and often the second-best method by a considerable margin.

This performance difference highlights a key distinction between modeling objectives. SBIRR is designed to directly interpolate between training marginals, and in this setting—where particles are relatively dense and the underlying flow field is smooth—interpolating training points naturally leads to trajectories that also pass near the validation points, which lie in between. In contrast, our method is not explicitly optimized to pass through the training marginals, but rather to estimate a smooth underlying velocity field from the available data. This distinction becomes particularly relevant in tasks such as forecasting, where the goal is to recover and extrapolate the underlying dynamics rather than simply interpolate known states.

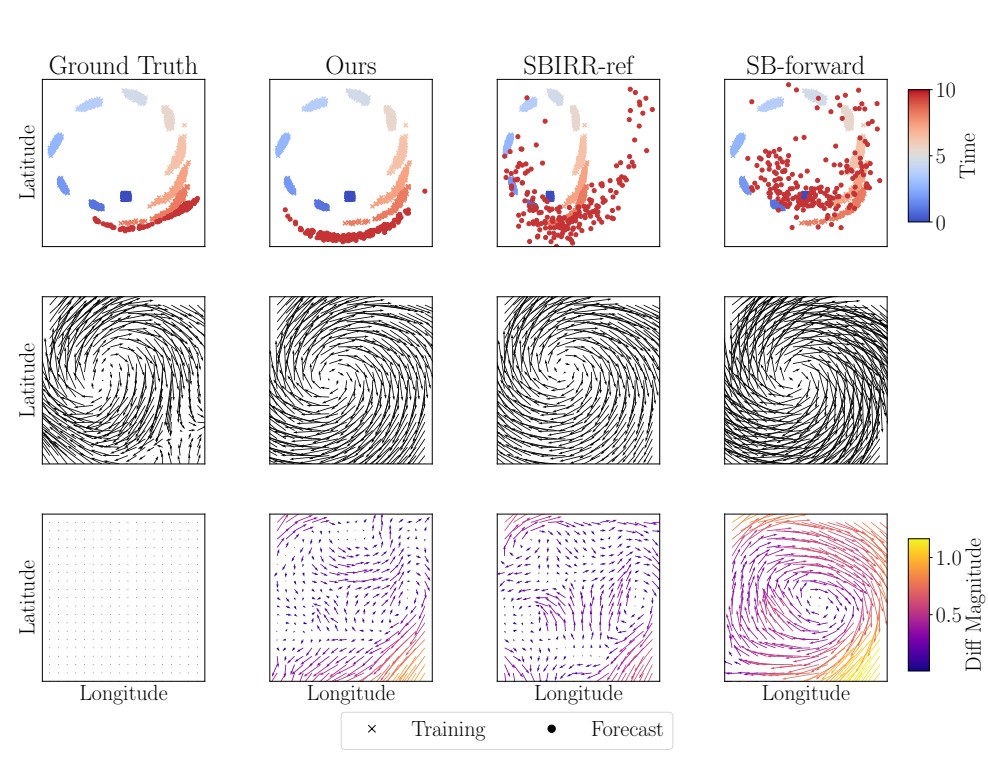

Figure A17: Experimental results for the Gulf of Mexico experiment.

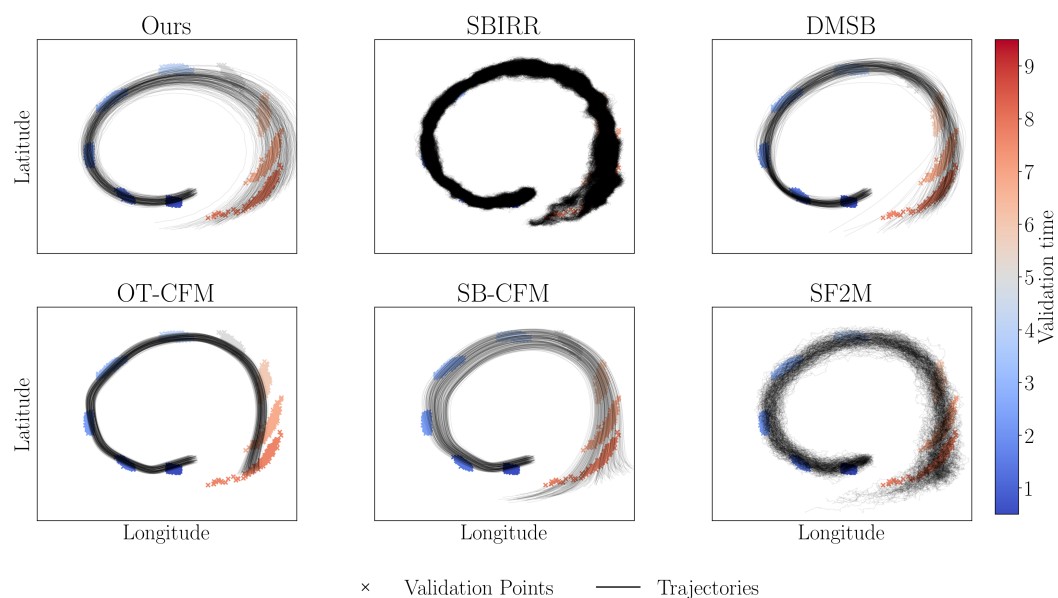

Figure A18: Interpolation of Gulf of Mexico current.

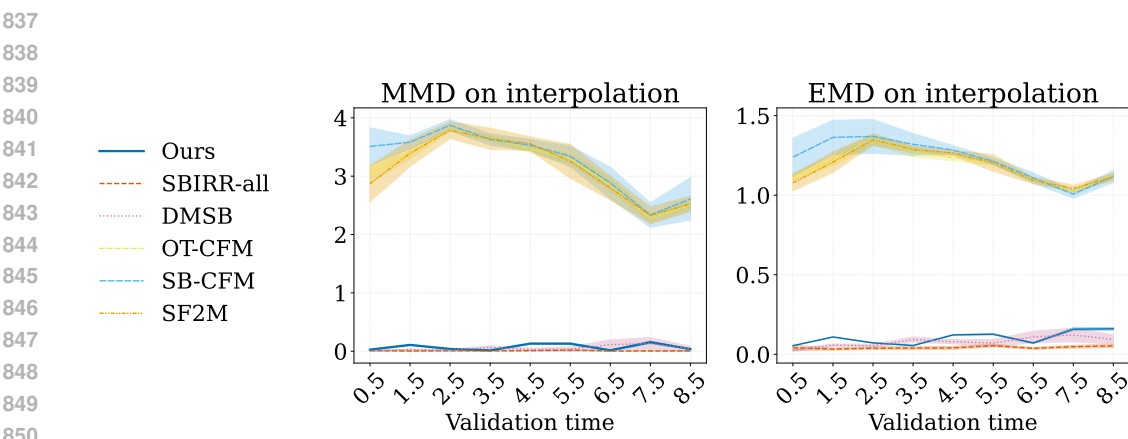

Figure A19: Metrics for Interpolation of Gulf of Mexico current.

Table 18: MMD at each validation point for Gulf of Mexico.

| Time | Ours | SBIRR | DMSB | OT−CFM | SB−CFM | SF2M |
|------|------|-------|------|--------|--------|------|
| 0.5 | $0.027 \pm 0.001$ | $0.011 \pm 0.014$ | $\mathbf{0.005 \pm 0.004}$ | $0.030 \pm 0.002$ | $0.030 \pm 0.004$ | $0.030 \pm 0.004$ |
| 1.5 | $0.108 \pm 0.004$ | $\mathbf{0.004 \pm 0.004}$ | $0.030 \pm 0.007$ | $0.109 \pm 0.013$ | $0.097 \pm 0.031$ | $0.108 \pm 0.013$ |
| 2.5 | $0.039 \pm 0.003$ | $\mathbf{0.008 \pm 0.006}$ | $0.021 \pm 0.009$ | $0.351 \pm 0.043$ | $0.353 \pm 0.136$ | $0.397 \pm 0.030$ |
| 3.5 | $0.015 \pm 0.002$ | $\mathbf{0.007 \pm 0.007}$ | $0.064 \pm 0.028$ | $0.820 \pm 0.123$ | $0.854 \pm 0.374$ | $0.900 \pm 0.090$ |
| 4.5 | $0.130 \pm 0.007$ | $\mathbf{0.009 \pm 0.005}$ | $0.031 \pm 0.017$ | $0.723 \pm 0.160$ | $0.797 \pm 0.435$ | $0.745 \pm 0.093$ |
| 5.5 | $0.130 \pm 0.010$ | $\mathbf{0.019 \pm 0.009}$ | $0.036 \pm 0.039$ | $1.022 \pm 0.205$ | $1.049 \pm 0.434$ | $1.000 \pm 0.070$ |
| 6.5 | $0.015 \pm 0.005$ | $\mathbf{0.004 \pm 0.005}$ | $0.114 \pm 0.080$ | $1.277 \pm 0.248$ | $1.192 \pm 0.383$ | $1.207 \pm 0.139$ |
| 7.5 | $0.154 \pm 0.023$ | $\mathbf{0.005 \pm 0.003}$ | $0.133 \pm 0.103$ | $1.022 \pm 0.271$ | $1.000 \pm 0.374$ | $0.884 \pm 0.192$ |
| 8.5 | $0.037 \pm 0.006$ | $\mathbf{0.005 \pm 0.007}$ | $0.040 \pm 0.044$ | $0.507 \pm 0.239$ | $0.867 \pm 0.772$ | $0.348 \pm 0.141$ |

Table 19: EMD at each validation point for Gulf of Mexico.

| Time | Ours | SBIRR | DMSB | OT−CFM | SB−CFM | SF2M |
|------|------|-------|------|--------|--------|------|
| 0.5 | $0.054 \pm 0.001$ | $0.041 \pm 0.017$ | $\mathbf{0.025 \pm 0.007}$ | $0.057 \pm 0.002$ | $0.057 \pm 0.004$ | $0.068 \pm 0.004$ |
| 1.5 | $0.109 \pm 0.002$ | $\mathbf{0.033 \pm 0.007}$ | $0.058 \pm 0.007$ | $0.109 \pm 0.006$ | $0.104 \pm 0.015$ | $0.114 \pm 0.007$ |
| 2.5 | $0.072 \pm 0.002$ | $\mathbf{0.038 \pm 0.009}$ | $0.054 \pm 0.009$ | $0.194 \pm 0.012$ | $0.194 \pm 0.037$ | $0.210 \pm 0.009$ |
| 3.5 | $0.055 \pm 0.002$ | $\mathbf{0.039 \pm 0.011}$ | $0.092 \pm 0.017$ | $0.301 \pm 0.024$ | $0.303 \pm 0.069$ | $0.318 \pm 0.016$ |
| 4.5 | $0.122 \pm 0.003$ | $\mathbf{0.041 \pm 0.010}$ | $0.079 \pm 0.013$ | $0.277 \pm 0.033$ | $0.289 \pm 0.081$ | $0.286 \pm 0.017$ |
| 5.5 | $0.126 \pm 0.004$ | $\mathbf{0.055 \pm 0.009}$ | $0.070 \pm 0.023$ | $0.333 \pm 0.036$ | $0.342 \pm 0.073$ | $0.336 \pm 0.012$ |
| 6.5 | $0.071 \pm 0.005$ | $\mathbf{0.037 \pm 0.007}$ | $0.109 \pm 0.036$ | $0.379 \pm 0.040$ | $0.374 \pm 0.062$ | $0.374 \pm 0.023$ |
| 7.5 | $0.158 \pm 0.009$ | $\mathbf{0.048 \pm 0.009}$ | $0.121 \pm 0.043$ | $0.351 \pm 0.046$ | $0.358 \pm 0.071$ | $0.324 \pm 0.038$ |
| 8.5 | $0.160 \pm 0.008$ | $\mathbf{0.053 \pm 0.011}$ | $0.094 \pm 0.029$ | $0.264 \pm 0.049$ | $0.361 \pm 0.163$ | $0.210 \pm 0.045$ |

## F.10 T CELL-MEDIATED IMMUNE ACTIVATION

In this section, we describe details of the T cell-mediated immune activation experiment contained in the main text. The dataset was released with the paper by Jiang et al. (2024) under CC-BY-NC 4.0 international license.

### F.10.1 BACKGROUND FOR THE BIOLOGICAL EXPERIMENT.

**Peripheral blood mononuclear cells** Peripheral blood mononuclear cells (PBMCs) comprise a heterogeneous mixture of immune cells, including T cells, B cells, natural killer (NK) cells, and myeloid lineages. To trigger a coordinated immune response, the PBMC pool from a single healthy donor was stimulated *in vitro* with anti-CD3/CD28 antibodies at $t = 0$ to selectively induce T cell activation. Cytokines released by activated T cells subsequently induce transcriptional changes and subsequent cytokine communications in the immune cell populations, yielding a rich, system-wide dynamical process.

**Data acquisition.** Cells were sampled every 30 min for 30 h (61 time points) and profiled with multiplexed single-cell mRNA sequencing (Jiang et al., 2024). Raw counts were library-size normalised and log-transformed following standard scRNA-seq workflows. Each snapshot is very high-dimensional (there are hundreds of cells and thousands of genes for each cell). Because modelling all genes directly is infeasible and biologically redundant, we adopt the widely used *gene-program* formulation: groups of co-expressed genes are collapsed into latent variables that capture coordinated transcriptional activity.

### F.10.2 EXPERIMENT SETUP

We use the 30 biologically annotated programs in Jiang et al. (2024) together with the dataset, which was computed from orthogonal non-negative matrix factorisation (oNMF) (Ding et al., 2006). Projecting each cell onto this 30-dimensional program space yields a low-noise, interpretable representation that is well suited for dynamical modelling. We took data from 0-20 hours (41 snapshots in total), before the cells reached steady state. We train our model using data at $0, 1, \ldots, 19$th hours for training, left measurement at 20th hour to test for forecast, and left $0.5, 1.5, \ldots, 18.5$th hour to test for interpolation. We show in Fig. A20 the evolution of the 20 training points and the forecast validation time point (bottom-left).

### F.10.3 MODEL FAMILY CHOICE

We employ the architecture of Eq. (A11), instantiated with a 30-dimensional state space. The drift is parameterised by a three-layer MLP with hidden widths $[128, 128, 128]$, ReLU activations and a sigmoid output that keeps gene-program values within biologically plausible bounds. Hyper-parameters were chosen via a small grid search (two vs. three layers, and 32 vs. 64 vs. 128 hidden nodes per layer) on the $R^2$ score.

### F.10.4 FORECASTING RESULTS

Table 20 reports quantitative forecasting performance. Because the ground-truth vector field is unknown, we restrict evaluation to distributional metrics. EMD solver failed to converge in 30 dimensions, so we use MMD with an RBF kernel of bandwidth 1. Our method attains an MMD that is an order of magnitude lower than either baseline, confirming the qualitative superiority observed in Fig. 4.

Table 20: One-step-ahead forecasting error on the T-cell activation dataset (mean $\pm$ s.d. over 10 seeds). MMD is computed with an RBF kernel of bandwidth 1 after scaling each gene program to unit variance. EMD is not reported because the solver failed to converge in 30 dimensions.

| Metric | pbmc | | |
| --- | --- | --- | --- |
| | Ours | SBIRR-ref | SB-forward |
| Forecast-MMD | **0.11 (0.04)** | 0.69 (0.11) | 2.99 (1.88) |

### F.10.5 INTERPOLATION

We now assess interpolation performance on this real-world single-cell dataset. As shown in Fig. A21, our method produces biologically plausible trajectories that span the principal components of the data and remain well-aligned with the progression of held-out validation points. SBIRR, OT-CFM , and SB-CFM perform similarly well in this task, whereas DMSB produces notably erratic

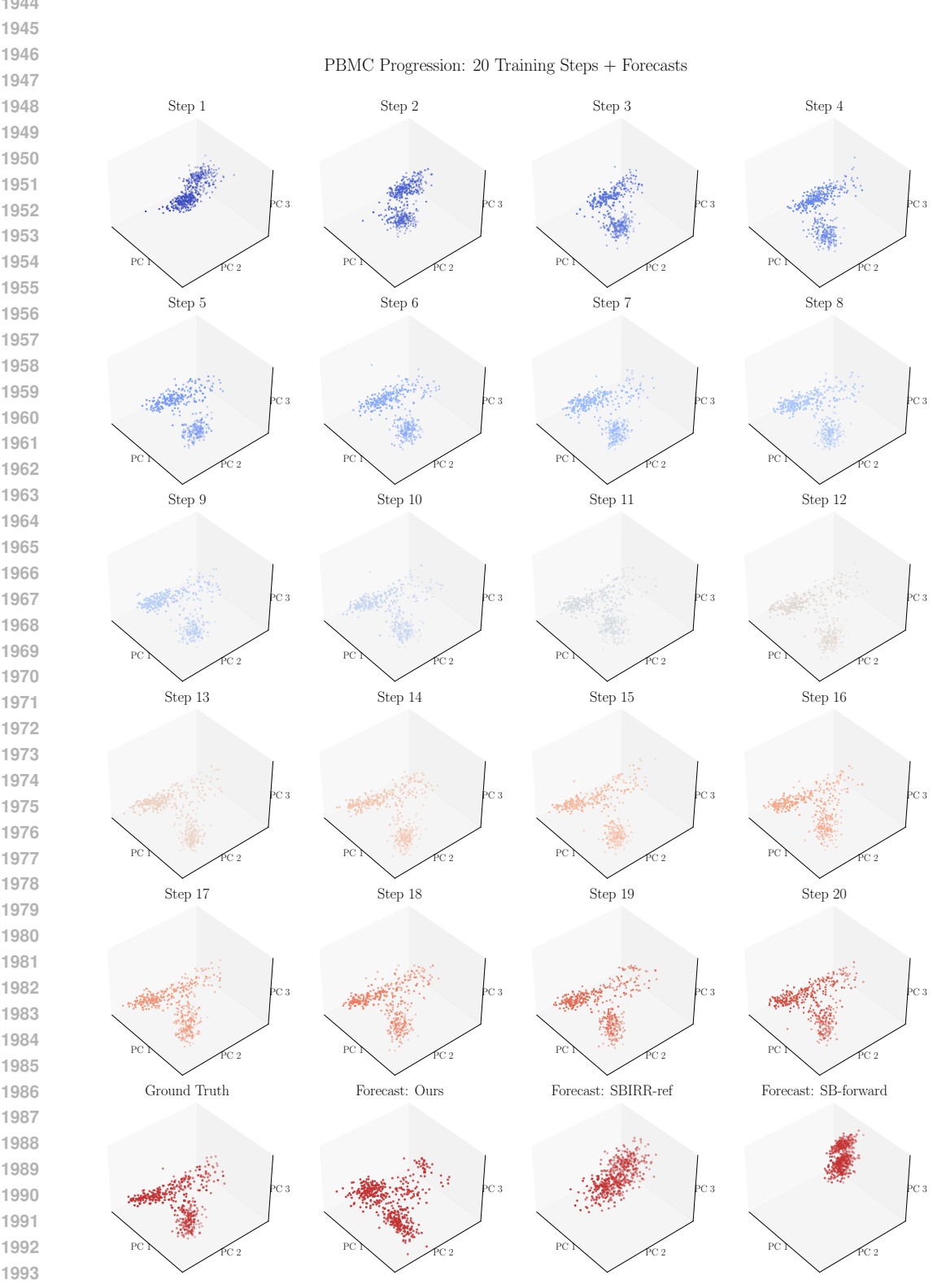

Figure A20: Training data for the PBMC experiment and forecasts with the three methods.

paths with high variance, and SF2M displays more diffuse interpolations. The same pattern appears if we look at interpolation predictions across the validation time steps, as shown in Fig. A22, Fig. A23, Fig. A24, and Fig. A25. In particular, we see that our method and SBIRR produce quite accurate interpolations for most of the time steps. Quantitative results in Fig. A26 and Table 21 support these impressions. Across the full time course, our method consistently achieves the lowest MMD at most validation points, often by a statistically significant margin. SBIRR performs competitively, particularly at later times, while the performance of DMSB degrades substantially, as reflected by persistently high MMD values throughout the trajectory. We note that, in contrast to previous experiments, we do not report EMD in this setting. Due to the high dimensionality of the latent space (30 dimensions), EMD becomes increasingly unreliable as a metric, suffering from the curse of dimensionality and producing unstable estimates. For this reason, we focus our evaluation on MMD, which remains well-behaved in high-dimensional settings and provides a more robust comparison of distributional fidelity across methods.

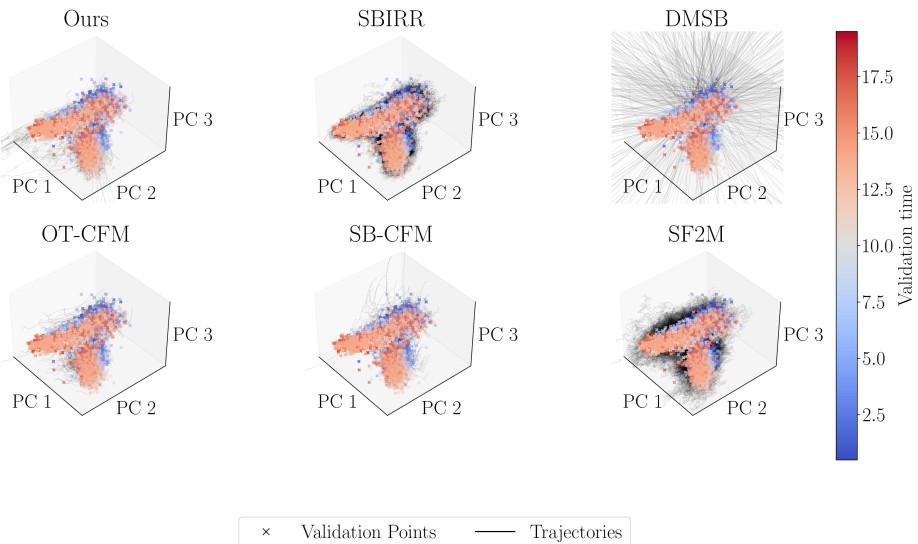

Figure A21: Interpolation of pbmc dataset.

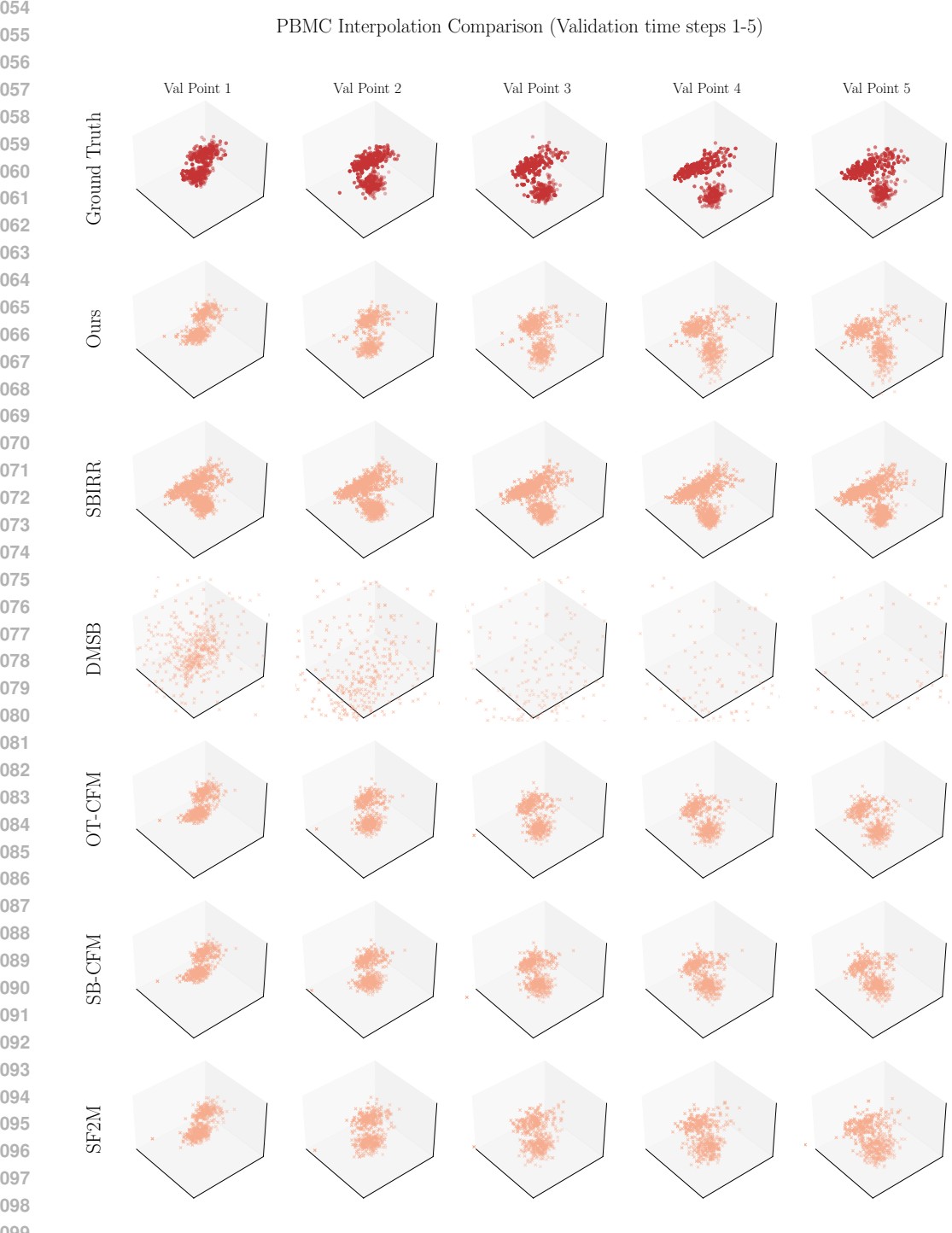

Figure A22: PBMC interpolation results for the first 5 validation time points. The axis are the first three principal components as in the forecasting experiment. The first row shows the evolution of cells for ground truth. The other six rows show the predicted cells at the validation time points for the our method and the five baselines.

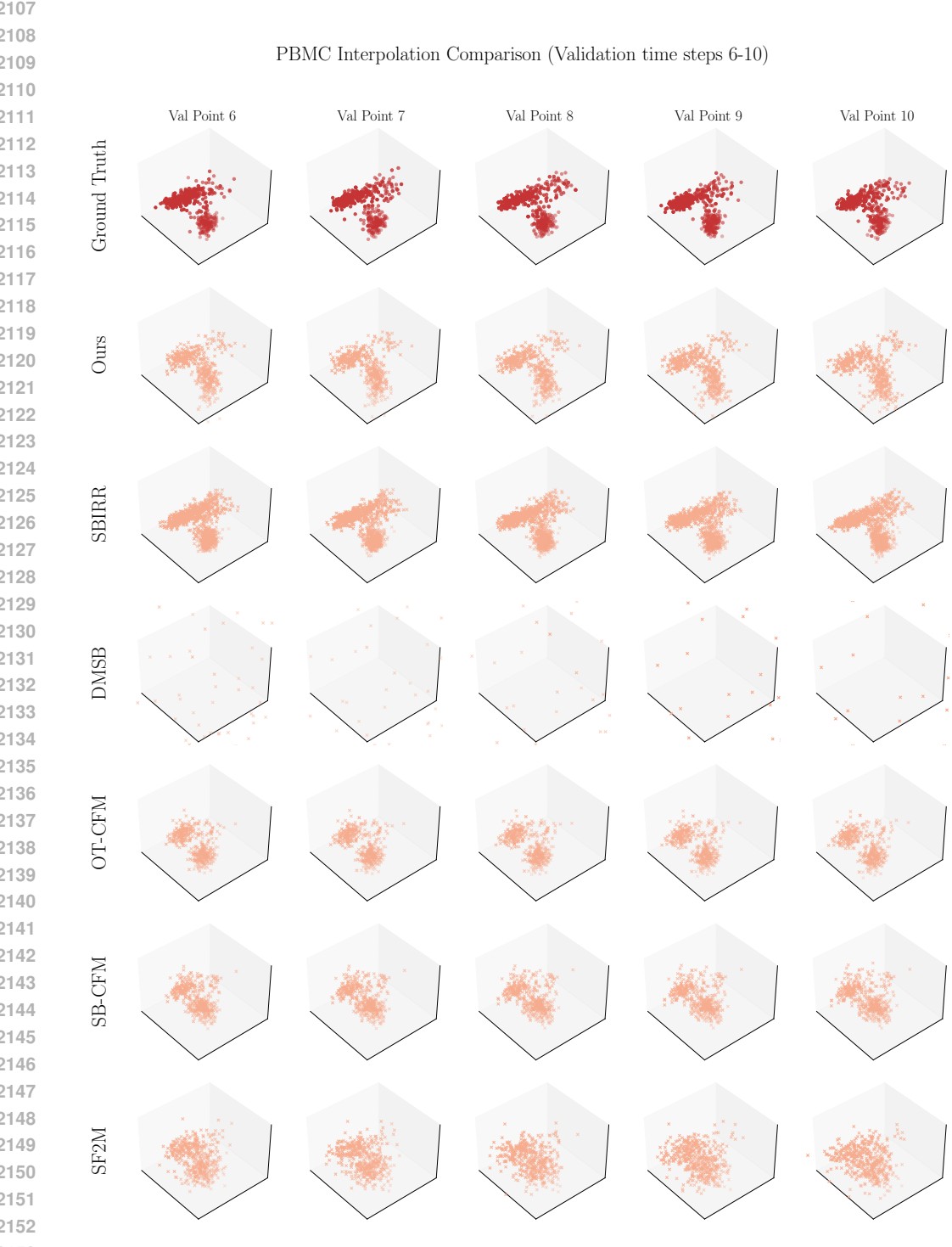

Figure A23: PBMC interpolation results for validation time points 6 to 10. The axis are the first three principal components as in the forecasting experiment. The first row shows the evolution of cells for ground truth. The other six rows show the predicted cells at the validation time points for the our method and the five baselines.

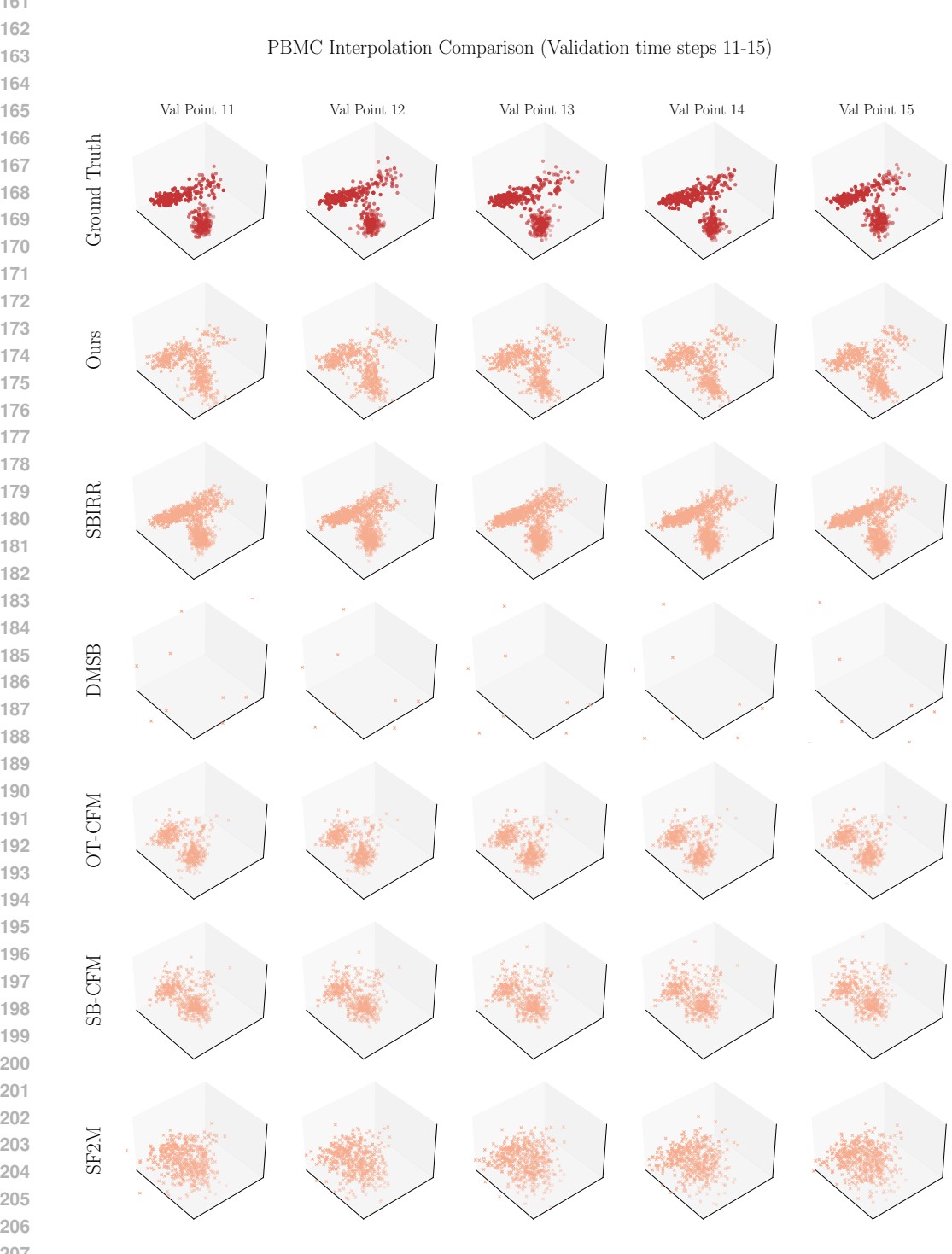

Figure A24: PBMC interpolation results for validation time points 11 to 15. The axis are the first three principal components as in the forecasting experiment. The first row shows the evolution of cells for ground truth. The other six rows show the predicted cells at the validation time points for the our method and the five baselines.

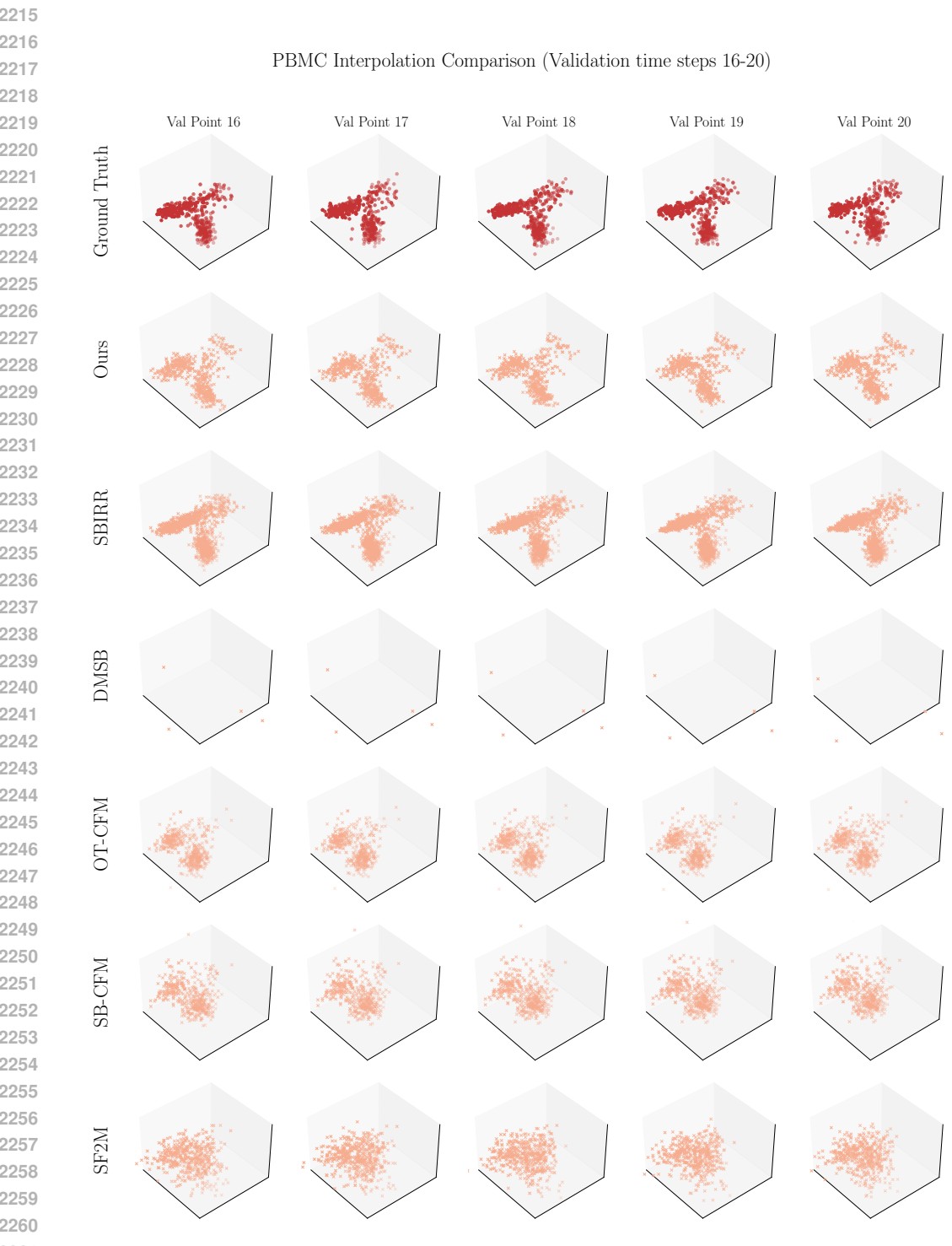

Figure A25: PBMC interpolation results for validation time points 16 to 20. The axis are the first three principal components as in the forecasting experiment. The first row shows the evolution of cells for ground truth. The other six rows show the predicted cells at the validation time points for the our method and the five baselines.

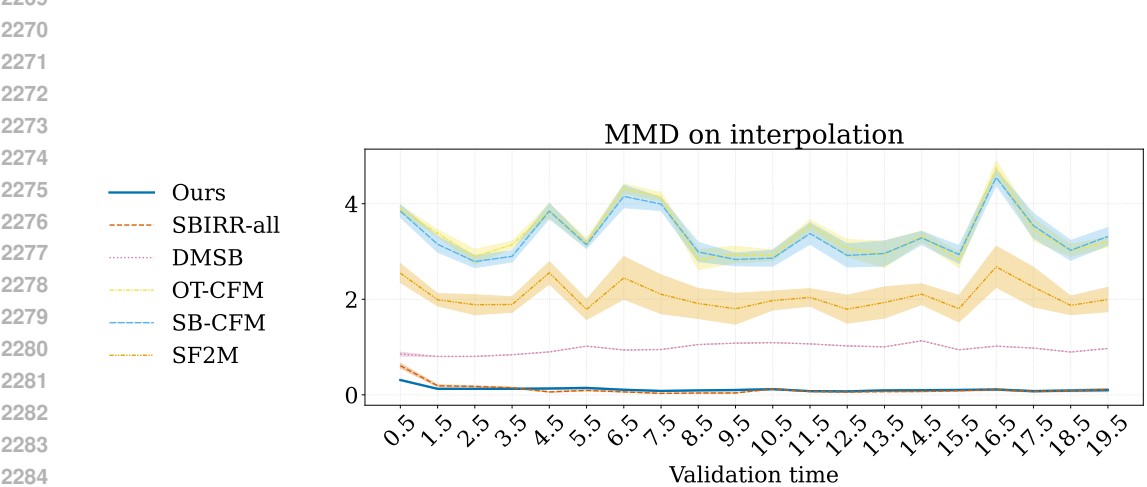

Figure A26: Metrics for Interpolation of pbmc dataset.

Table 21: MMD at each validation point for PBMC.

| Time | Ours | SBIRR | DMSB | OT−CFM | SB−CFM | SF2M |
|---|---|---|---|---|---|---|
| 0.5 | $0.310 \pm 0.013$ | $0.610 \pm 0.052$ | $0.853 \pm 0.041$ | $0.318 \pm 0.010$ | $0.328 \pm 0.009$ | $\mathbf{0.306 \pm 0.008}$ |
| 1.5 | $\mathbf{0.125 \pm 0.010}$ | $0.189 \pm 0.022$ | $0.803 \pm 0.001$ | $0.183 \pm 0.029$ | $0.201 \pm 0.018$ | $0.163 \pm 0.019$ |
| 2.5 | $\mathbf{0.125 \pm 0.011}$ | $0.175 \pm 0.012$ | $0.804 \pm 0.000$ | $0.264 \pm 0.053$ | $0.270 \pm 0.053$ | $0.236 \pm 0.050$ |
| 3.5 | $\mathbf{0.126 \pm 0.012}$ | $0.148 \pm 0.015$ | $0.839 \pm 0.000$ | $0.309 \pm 0.064$ | $0.308 \pm 0.059$ | $0.265 \pm 0.050$ |
| 4.5 | $0.132 \pm 0.019$ | $\mathbf{0.060 \pm 0.009}$ | $0.898 \pm 0.000$ | $0.275 \pm 0.069$ | $0.294 \pm 0.059$ | $0.258 \pm 0.051$ |
| 5.5 | $0.144 \pm 0.016$ | $\mathbf{0.092 \pm 0.010}$ | $1.019 \pm 0.000$ | $0.318 \pm 0.081$ | $0.339 \pm 0.063$ | $0.301 \pm 0.049$ |
| 6.5 | $0.106 \pm 0.013$ | $\mathbf{0.061 \pm 0.007}$ | $0.939 \pm 0.000$ | $0.335 \pm 0.096$ | $0.357 \pm 0.072$ | $0.295 \pm 0.049$ |
| 7.5 | $0.081 \pm 0.009$ | $\mathbf{0.030 \pm 0.005}$ | $0.948 \pm 0.000$ | $0.342 \pm 0.107$ | $0.375 \pm 0.087$ | $0.292 \pm 0.056$ |
| 8.5 | $0.093 \pm 0.011$ | $\mathbf{0.037 \pm 0.003}$ | $1.052 \pm 0.000$ | $0.374 \pm 0.122$ | $0.422 \pm 0.097$ | $0.329 \pm 0.054$ |
| 9.5 | $0.100 \pm 0.016$ | $\mathbf{0.039 \pm 0.002}$ | $1.081 \pm 0.000$ | $0.396 \pm 0.130$ | $0.456 \pm 0.105$ | $0.353 \pm 0.056$ |
| 10.5 | $\mathbf{0.118 \pm 0.018}$ | $0.122 \pm 0.010$ | $1.092 \pm 0.000$ | $0.528 \pm 0.150$ | $0.569 \pm 0.126$ | $0.425 \pm 0.068$ |
| 11.5 | $0.077 \pm 0.010$ | $\mathbf{0.069 \pm 0.007}$ | $1.066 \pm 0.000$ | $0.505 \pm 0.154$ | $0.554 \pm 0.131$ | $0.397 \pm 0.074$ |
| 12.5 | $0.073 \pm 0.010$ | $\mathbf{0.059 \pm 0.006}$ | $1.023 \pm 0.000$ | $0.519 \pm 0.156$ | $0.567 \pm 0.129$ | $0.388 \pm 0.073$ |
| 13.5 | $0.093 \pm 0.014$ | $\mathbf{0.066 \pm 0.004}$ | $1.004 \pm 0.000$ | $0.573 \pm 0.163$ | $0.621 \pm 0.130$ | $0.417 \pm 0.070$ |
| 14.5 | $0.096 \pm 0.016$ | $\mathbf{0.071 \pm 0.010}$ | $1.131 \pm 0.000$ | $0.591 \pm 0.196$ | $0.679 \pm 0.151$ | $0.448 \pm 0.080$ |
| 15.5 | $0.101 \pm 0.016$ | $\mathbf{0.085 \pm 0.008}$ | $0.942 \pm 0.000$ | $0.533 \pm 0.163$ | $0.601 \pm 0.125$ | $0.360 \pm 0.058$ |
| 16.5 | $0.111 \pm 0.015$ | $\mathbf{0.108 \pm 0.017}$ | $1.019 \pm 0.000$ | $0.594 \pm 0.183$ | $0.683 \pm 0.130$ | $0.414 \pm 0.060$ |
| 17.5 | $0.077 \pm 0.019$ | $\mathbf{0.071 \pm 0.015}$ | $0.978 \pm 0.000$ | $0.598 \pm 0.190$ | $0.688 \pm 0.141$ | $0.404 \pm 0.060$ |
| 18.5 | $0.089 \pm 0.023$ | $\mathbf{0.089 \pm 0.015}$ | $0.896 \pm 0.000$ | $0.616 \pm 0.185$ | $0.698 \pm 0.134$ | $0.385 \pm 0.055$ |
| 19.5 | $\mathbf{0.099 \pm 0.034}$ | $0.107 \pm 0.022$ | $0.971 \pm 0.000$ | $0.637 \pm 0.204$ | $0.753 \pm 0.146$ | $0.425 \pm 0.063$ |

# G   ABLATION STUDIES FULL RESULTS

In this appendix, we present extended results for the ablation studies from Section 4.6. We consider two settings: (i) replacing learned, state-dependent volatility with a fixed scalar volatility, and (ii) replacing structured SDE drift/volatility with a fully neural parameterization. Tables report mean, standard deviation, and range across seeds for both SnapMMD and the ablated variants.

## G.1   IMPACT OF LEARNING STATE-DEPENDENT VOLATILITY

In our main experiments, SnapMMD learns a volatility term that may vary with state. Here we ablate this feature by fixing volatility to a constant scalar across states and times (set to $0.1$ following prior SB work).

Across tasks, learning volatility yields clear benefits: in Lotka–Volterra and parametric Repressilator, MMD errors fall by more than an order of magnitude relative to fixed volatility, reflecting the importance of capturing heterogeneous noise levels. In PBMC, the fixed-volatility variant shows large instability across seeds (variance over an order of magnitude), while learned volatility produces consistently low error. The only exception is the Gulf of Mexico forecasting task, where fixed volatility yields more diffuse predictions that happen to align better with the kernel geometry used in the MMD metric. Importantly, even in this case, SnapMMD with learned volatility performs better on the interpolation task, where the goal is reconstructing observed marginals rather than extrapolating.

Table 22: Forecasting performance: SnapMMD with learned state-dependent diffusion vs. fixed diffusion.

|  | SnapMMD (learned volatility) | | | Fixed volatility | | |
|---|---|---|---|---|---|---|
| Experiment | Mean | SD | Range | Mean | SD | Range |
| LV | 0.057 | 0.030 | [0.025, 0.122] | 0.156 | 0.020 | [0.128, 0.189] |
| ReprParam | 0.072 | 0.080 | [0.005, 0.243] | 7.557 | 0.001 | [7.555, 7.559] |
| ReprSemiparam | 0.320 | 0.150 | [0.150, 0.640] | 0.286 | 0.191 | [0.035, 0.656] |
| ReprProtein | 0.067 | 0.034 | [0.018, 0.116] | 0.072 | 0.023 | [0.041, 0.128] |
| GoM | 2.360 | 0.110 | [2.192, 2.582] | 0.969 | 0.398 | [0.654, 2.079] |
| PBMC | 0.110 | 0.040 | [0.077, 0.236] | 0.376 | 0.542 | [0.077, 1.667] |

Table 23: Interpolation performance: SnapMMD with learned state-dependent diffusion vs. fixed volatility. We report mean, standard deviation, and range across seeds.

|  | SnapMMD (learned volatility) | | | Fixed volatility | | |
|---|---|---|---|---|---|---|
| Experiment | Mean | SD | Range | Mean | SD | Range |
| LV | 0.013 | 0.011 | [0.000, 0.075] | 8.771 | 2.257 | [1.599, 9.917] |
| ReprParam | 0.040 | 0.034 | [0.002, 0.166] | 8.461 | 0.588 | [7.693, 9.570] |
| ReprSemiparam | 0.379 | 0.381 | [0.005, 1.459] | 8.423 | 1.770 | [2.643, 9.919] |
| ReprProtein | 0.011 | 0.006 | [0.002, 0.034] | 8.768 | 1.350 | [4.074, 9.889] |
| GoM | 0.073 | 0.054 | [0.007, 0.200] | 6.716 | 3.156 | [0.312, 9.904] |
| PBMC | 0.114 | 0.052 | [0.058, 0.327] | 4.636 | 0.880 | [3.011, 6.121] |

## G.2   ROLE OF INDUCTIVE BIAS

We next ablate the use of domain-informed structure by replacing the SDE drift and diffusion with fully neural parameterizations. This removes all mechanistic scaffolding and corresponds to fitting an unconstrained neural SDE directly to the snapshot data.

Across tasks, the structured SnapMMD models outperform their fully neural counterparts, often dramatically. For example, in Lotka–Volterra and Repressilator, the neural SDE fails to recover qualitative dynamics and produces MMD errors more than two orders of magnitude worse than SnapMMD. In PBMC, where prior mechanistic knowledge is weaker, the gap is smaller but still consistent in favor of structured models. The Gulf of Mexico forecasting task again provides an

exception: here the neural variant produces more diffuse forecasts that achieve a lower MMD score, though interpolation still favors SnapMMD.

Taken together, these ablations highlight that inductive bias is essential when reliable domain knowledge is available, while semiparametric combinations of structured drift with neural residuals are most useful when knowledge is partial. Fully neural SDEs, in contrast, often fail to capture meaningful dynamics from snapshot data alone, lacking the guidance that domain structure provides.

Table 24: Forecasting performance: SnapMMD with domain-informed structure vs. fully neural SDE. We report mean, standard deviation, and range across seeds.

| Experiment | SnapMMD (structured) | | | Fully neural | | |
|---|---|---|---|---|---|---|
| | Mean | SD | Range | Mean | SD | Range |
| LV | 0.057 | 0.030 | [0.025, 0.122] | 7.230 | 0.035 | [7.173, 7.288] |
| ReprSemiparam | 0.320 | 0.150 | [0.150, 0.640] | 3.670 | 0.467 | [2.900, 4.282] |
| ReprProtein | 0.048 | 0.029 | [0.018, 0.116] | 3.423 | 0.387 | [2.711, 3.917] |
| GoM | 2.360 | 0.110 | [2.192, 2.582] | 0.607 | 0.113 | [0.324, 0.739] |
| PBMC | 0.110 | 0.040 | [0.077, 0.236] | 0.292 | 0.004 | [0.286, 0.298] |

Table 25: Interpolation performance: SnapMMD with domain-informed structure vs. fully neural SDE. We report mean, standard deviation, and range across seeds.

| Experiment | SnapMMD (structured) | | | Fully neural | | |
|---|---|---|---|---|---|---|
| | Mean | SD | Range | Mean | SD | Range |
| LV | 0.013 | 0.011 | [0.000, 0.075] | 8.667 | 1.734 | [1.426, 9.934] |
| ReprSemiparam | 0.379 | 0.381 | [0.005, 1.459] | 8.754 | 2.273 | [1.786, 9.917] |
| ReprProtein | 0.011 | 0.006 | [0.002, 0.034] | 8.654 | 1.691 | [2.817, 9.889] |
| GoM | 0.073 | 0.054 | [0.007, 0.200] | 6.530 | 3.327 | [0.197, 9.709] |
| PBMC | 0.114 | 0.052 | [0.058, 0.327] | 4.877 | 0.413 | [3.999, 5.796] |

## H    IDENTIFIABILITY ANALYSIS

In this appendix, we provide further details on the identifiability problem from the the main text discussion.

**Why drift and volatility are not identified in general.** Even with complete access to the marginal distributions $\pi_t$ over time, the pair $(\boldsymbol{b}_0, \boldsymbol{g}_0)$ is not uniquely determined by the Fokker–Planck equation

$$\frac{\partial \pi_t}{\partial t} = \nabla \cdot \left[ -\boldsymbol{b}_0 \, \pi_t + \frac{1}{2} \, \boldsymbol{g}_0 \, \boldsymbol{g}_0^\top \nabla \pi_t \right]$$

For example, suppose $(\boldsymbol{b}_0, \boldsymbol{g}_0)$ satisfies the equation for a given $\pi_t$. Then, for any vector field $\boldsymbol{h}$ that satisfies the continuity condition $\nabla \cdot (\boldsymbol{h} \, \pi_t) = 0$, the modified drift $\boldsymbol{b}_0' = \boldsymbol{b}_0 + \boldsymbol{h}$ with the same volatility $\boldsymbol{g}_0$ also satisfies the Fokker–Planck equation. This observation indicates that an infinite family of drift functions can generate the same evolution of the marginal distribution if no further constraints are imposed. Furthermore, let $\boldsymbol{A}$ be any orthogonal matrix (i.e., $\boldsymbol{A}\boldsymbol{A}^\top = \boldsymbol{I}$). Then, the pair $(\boldsymbol{b}_0, \boldsymbol{g}_0 \, \boldsymbol{A})$ also satisfies the Fokker–Planck equation. These examples illustrate the inherent non-uniqueness (or non-identifiability) of the drift and volatility functions based solely on the evolution of the marginal distributions.

In practice, to achieve identifiability, one must restrict the candidate function classes for $\boldsymbol{b}_0$ and $\boldsymbol{g}_0$. For instance, assuming that $\boldsymbol{b}_0$ is a gradient field (i.e., $\boldsymbol{b}_0 = \nabla \Phi$ for some potential $\Phi$ and that $\boldsymbol{g}_0$ is constant is known to yield identifiability under suitable conditions (Lavenant et al., 2024; Guan et al., 2024). A complete characterization of identifiability in more general settings is beyond the scope of this work and constitutes an important direction for future research.

# I  LARGE LANGUAGE MODEL (LLM) USE

We used LLM for grammar checks and to polish writing.

