# OpenReview forum: "Oh SnapMMD! Forecasting stochastic dynamics beyond the Schrödinger bridge’s end"
_ICLR.cc/2026/Conference — Submitted to ICLR 2026_

### Official Review · Reviewer_odXk · 2025-10-29

**Soundness:** 2
**Presentation:** 3
**Contribution:** 2
**Rating:** 4
**Confidence:** 4

**Summary:**

This paper considers the problem of forecasting stochastic dynamics from snapshot data, such as single-cell RNA-seq or ecological systems, where only population-level states at discrete times are observed. The authors propose SnapMMD, a framework that learns both drift and state-dependent diffusion of an SDE by directly matching the joint distribution of states and times using a Maximum Mean Discrepancy (MMD) loss. Unlike Schrödinger-Bridge (SB) methods that minimize a KL divergence to a fixed reference process, SnapMMD dispenses with the reference dynamics, allowing the g_\theta(x,t) to be learned from data. It also introduces an RKHS-based R^2 statistic for model fit and can handle partially observed states.

**Strengths:**

- Clear motivation for moving beyond interpolation toward genuine forecasting.

**Weaknesses:**

1. Confounded gains from structural priors vs. objective/volatility learning. Only SBIRR is configured to incorporate the same drift-structure prior as SnapMMD, whereas several interpolation/generative baselines cannot accommodate comparable inductive bias. Consequently, observed improvements may partly reflect unequal access to the mechanistic structure rather than the MMD objective or the learned, state/time-dependent diffusion.
2.  Volatility advantage may be overstated without tighter controls. Baselines are run with fixed, known diffusion (e.g., $\sigma=0.1$), while SnapMMD learns state/time-dependent volatility; the paper does not provide matched ablations using fixed $\sigma$ values calibrated to the learned diffusion’s empirical scale (e.g., mean over states/times). This leaves open whether forecast gains stem from better volatility calibration rather than the genuine benefits of state dependence.
3. Partial-observability claims rely mainly on small-scale simulations. The “mRNA+protein” advantage is demonstrated on synthetic data; real high-dimensional experiments (e.g., PBMC) do not include additional unobserved variables, so practical utility in complex biological settings remains under-substantiated.
4. Identifiability is not guaranteed and is acknowledged by the authors. Even full time series of marginals may not uniquely determine drift and diffusion; without strong priors, learned $(b_\theta,g_\theta)$ can be non-unique up to observational equivalence, limiting mechanistic interpretability.
5. Metric and kernel dependence may bias training/evaluation. The method and primary metric rely on RBF-MMD with median-heuristic bandwidth; the paper notes cases (Gulf of Mexico) where MMD prefers more diffuse point clouds despite inferior geometric fidelity, and reports discrepancies between MMD and EMD. This raises concerns about sensitivity to kernel choice and bandwidth in both optimization and evaluation.
6. Simulation budget and sample-imbalance handling lack principled guidance. Training minimizes a weighted sum of per-time MMDs using U-statistic estimates from M simulated paths per time, but the paper provides no principled selection or diagnostics for M or for mitigating time-wise sample-size imbalance beyond the fixed weighting scheme.
7. Baseline coverage and parity are incomplete. Empirical comparisons omit additional strong or structure-compatible baselines (e.g., neural-ODE variants using the same mechanistic drift, or other recent trajectory inference methods, e.g., JKOnet (Terpin et al., NeurIPS 2024)), which limits claims of superiority across design spaces and problem regimes.
8. Compute scalability remains a practical concern. The approach is not simulation-free; training cost scales with sample size and dimensionality. The paper acknowledges this but does not quantify scaling behavior in large real-world settings.

**Questions:**

1. Disentangling sources of gain. Could you discuss ablations that (i) remove the mechanistic drift prior (use fully-neural b_\theta), (ii) keep the prior but fix diffusion, and (iii) keep the prior and learn diffusion, so that improvements can be attributed separately to structural priors, the MMD objective, and state/time-dependent diffusion? Please report both forecasting and interpolation metrics.
2. Volatility calibration controls. What are the empirical summaries (e.g., mean/variance over (x,t)) of the learned $g_\theta(x,t)$? If baselines use a fixed $\sigma$, how do results change when $\sigma$ is set to the learned diffusion’s mean (or a time-only or piecewise-constant surrogate)? Please include visual/quantitative comparisons of trajectory spread vs. fixed-\sigma surrogates.
3. Identifiability scope. Under what parametric restrictions or priors can parts of $(b_\theta,g_\theta)$ be identified from marginals? Could you formalize identifiable functionals even if b,g are not fully unique?
4. Metric/kernel sensitivity. How sensitive are training and model selection to kernel choice and bandwidth (median heuristic) in the MMD?
5. Simulation budget M and imbalance. What principles guide the choice of the number of simulated paths M per time point ?
6. Coverage of recent SOTA. Could you include additional trajectory-inference baselines (e.g., JKOnet)?
7. Forecast horizon robustness. How does forecasting error grow with horizon length beyond the last snapshot?
8. Could you quantify runtime/memory vs. dimension and dataset size?

I believe this paper considers an important problem—forecasting from population snapshots. However, several aspects require more careful justification. I therefore lean toward a borderline rejection at this stage, but I would be open to reconsidering if the authors address the above issues with convincing rebuttals.

References
1. Terpin, Antonio, et al. "Learning diffusion at lightspeed."NeurIPS 2024.

The reviewer wrote this review. The LLM was used only to improve both grammar and clarity.

---

> ### Author Response · Authors · 2025-11-25
>
> We thank the reviewer for their comments and questions.
>
> We briefly restate our contributions and then address individual points. Forecasting stochastic dynamics from population-level snapshot data is a fundamental problem in many scientific domains, yet to our knowledge existing approaches are designed only for interpolation and offer no principled mechanism for extrapolating beyond the final observed time. Moreover, Schrödinger Bridge (SB)­-based methods require users to fix a volatility, often state independent, in advance, even though volatility is typically unknown and state-dependent. The difficulty stems from the SB formalizing the problem in path space while we never observe data in path space. Our method, SnapMMD, directly learns latent dynamics by matching the joint distribution of states and measurement times using a factored-kernel MMD loss. This enables accurate forecasting, allows volatility to be learned (including state dependence), naturally incorporates domain knowledge, handles missing state dimensions, and provides interpretable diagnostics via an $R^2$-style metric.
>
> **Confounded gains from structural priors vs. objective/volatility learning & disentangling sources of gain experiments.** We appreciate the reviewer’s concern but believe this reflects a misunderstanding of our experimental design and our goals. The ability to incorporate domain structure is not a confounder but a core motivation of our method and of much scientific modeling. SnapMMD is explicitly designed to leverage partial mechanistic knowledge through parametric or semiparametric SDE families when such knowledge exists — just as practitioners in genetics, ecology, and oceanography routinely do. In all experiments, we provide the same structural drift family to every baseline that is capable of using it; for example, SBIRR is run with the identical drift and volatility parameterization whenever applicable.
>
> Other baselines cannot incorporate comparable inductive bias by design. For instance, they require fixed reference drifts, cannot learn volatility, or assume state-independent diffusion. We do not believe it is “unfair” to compare to such approaches; rather, it is scientifically informative to demonstrate what becomes possible when one uses all the information available to practitioners. To analogize: if one had access to ten features, it would be inappropriate to discard nine of them merely because a competing algorithm can only use one. Inductive bias, structural knowledge, and temporal context are real sources of signal, and methods that can leverage them should be allowed to do so.
>
> Regarding the source of performance gains: our improvements stem from (i) combining domain-informed structure with a lightweight, sample-level MMD loss that avoids path-space operations like optimizing KL between path measures in SB based methods, and (ii) learning state- and time-dependent volatility, which SB-based methods cannot do. As shown in semiparametric repressilator and PBMC experiments, even partial biological scaffolds (e.g., production–decay structure) meaningfully improve learning compared to baselines that cannot represent the true volatility or integrate such contextual knowledge. Similarly, in the Gulf of Mexico, a vortex-inspired drift family — though not physically exact — provides a realistic inductive bias that clearly benefits forecasting in data-sparse regimes.
>
> In short, the experimental gains do not arise from an unfair comparison, but from SnapMMD’s principled ability to integrate partial mechanistic knowledge and flexible volatility learning — capabilities that prior SB and flow-matching approaches fundamentally lack. We are revising the manuscript to explicitly document, for each baseline, which structural components it can and cannot utilize.
>
> We also importantly note that the reviewer, in the questions box, asks for three ablation studies on this point. **But, importantly, we believe the ablations the reviewer requests are in fact already included in the paper.** If we interpret (i)–(iii) in the reviewer’s question correctly, the correspondence is:
> - (i) *Remove mechanistic prior.* This corresponds to our non-parametric variant in Sec. 4.6 & Appendix G.2, where the drift is learned fully from data with no domain-informed structure.
> - (ii) *Keep mechanistic prior but fix diffusion.* This corresponds to our “fixed-volatility” ablation reported in Sec. 4.6 & Appendix G.1, where we use the same drift family as SnapMMD but hold the diffusion constant.
> - (iii) *Keep mechanistic prior and learn diffusion.* This is precisely the full SnapMMD model presented throughout the main text.
>
> Because these experiments are already in the submission, it is unclear to us why this point was raised as if they were missing. If we are misunderstanding the reviewer’s intention, or if the reviewer had a different decomposition in mind, we are happy to clarify or run additional small-scale checks during the discussion period.

---

> > ### Author Response · Authors · 2025-11-25
> >
> > **Volatility advantage may be overstated without tighter controls & volatility calibration control.** We appreciate the reviewer raising this point, but we believe the concern reflects a misunderstanding of the volatility setting in snapshot-based Schrödinger bridge methods.
> >
> > (1) *Learning volatility is a core contribution, not a confounder.* In SB-based approaches, the diffusion coefficient must be fixed a priori, and in practice there is no principled guidance for how this value should be chosen. As we note in the paper, a poorly calibrated diffusion substantially reduces interpolation and completely prevents forecasting in SB formulations. The ability of SnapMMD to learn volatility (including its spatial and temporal dependence) is precisely one of its main advantages. While it is true that a better-calibrated fixed diffusion could hypothetically improve the performance of SB baselines, these baselines do not provide any mechanism to infer such a value. Using SnapMMD’s learned diffusion to calibrate SB methods would simply transfer the benefit of our method to them, and it is not the scope of this submission. To our knowledge, there is no established technique for choosing or learning the optimal fixed volatility in SB formulations; the recent work of Guan et al. (2025) highlights that inferring diffusion even in simplified Langevin settings is a nontrivial and active research question.
> >
> > (2) *State dependence is common in scientific systems, and SB methods fundamentally cannot represent it.* Many real-world dynamical systems — including those in genetics, ecology, and fluid flow — exhibit state-dependent or anisotropic volatility. Classical SB methods (and even more recent extensions) are structurally incapable of modeling such effects, because their reference dynamics must have fixed, isotropic noise. SnapMMD, by contrast, is explicitly designed to allow unknown, state-dependent volatility when present and to reduce to a constant-volatility model when the application demands it. If the true system has constant diffusion, SnapMMD can learn this; if it has state-dependent diffusion, SB baselines cannot represent it at all.
> >
> > Thus, the observed gains are not artifacts of unequal calibration: they arise because SnapMMD can model aspects of the dynamics that SB methods are structurally unable to express. This is a genuine modeling advantage, not a confounding factor.
> >
> > **Partial-observability claims rely mainly on small-scale simulations.** We agree that evaluating partial observability in real, high-dimensional biological settings is an important question. At the same time, we believe the synthetic mRNA+protein experiment remains scientifically meaningful: it directly demonstrates that SnapMMD can recover latent coordinates that SB-based baselines cannot model at all, and it isolates the effect of unobserved dimensions in a controlled setting where ground truth is known — something that real PBMC data cannot provide. That said, we agree that exploring partial observability on PBMC is a natural extension. We have begun running these experiments and will share results during the discussion period.

---

> > > ### Author Response · Authors · 2025-11-25
> > >
> > > **Identifiability is not guaranteed and is acknowledged by the authors & identifiability scope.** We would like to clarify that the identifiability issues the reviewer raises are inherent to the snapshot-only observation regime, not a limitation of SnapMMD. When trajectories are never observed, the temporal covariance structure is lost, and neither drift $b$ nor diffusion $\sigma$ can be uniquely determined from a sequence of marginal distributions — even if the entire time-series of marginals is available. This is precisely why we discuss these issues openly in Appendix H and cite emerging theoretical work. Identifiability failures arise from the data regime itself, not from methodological choices.
> > >
> > > Every method that operates in this regime — Schrödinger bridges, flow matching, neural ODE surrogates, neural SDEs, or potential-based models — faces the same structural obstruction. Avoiding it would require access to fundamentally different information (e.g., repeated trajectory observations, measurement models, controlled perturbations, or strong mechanistic priors).
> > >
> > > Regarding the reviewer’s follow-up question about formal identifiability conditions for parts of $b$ or $\sigma$: this is an active research area in its own right. Even for ODEs with sparse or noisy observations, basic identifiability took decades to fully characterize. For SDEs, the field is only beginning to develop rigorous results — for example:
> > >
> > > - Guan et al. (2025): identifiability of gradient fields with unknown but state-independent diffusion;
> > > - Guan et al. (2024): identifiability for linear SDEs with known diffusion;
> > > - Lavenant et al. (2024): identifiability of gradient flows with known diffusion.
> > >
> > > These results cover narrow, highly structured subclasses and illustrate the depth of the theoretical challenge. Fully characterizing identifiable functionals of $b$ or $\sigma$ in the general, partially observed, non-gradient, unknown and state dependent volatility setting considered in this paper is beyond what can reasonably be addressed in a single methods paper.
> > >
> > > Our contribution is therefore not to solve the theoretical identifiability problem in full generality, but to push the practical Pareto frontier within the unavoidable limitations of the snapshot regime. Classical SB-based approaches impose extremely restrictive assumptions (e.g., fixed known volatility, no latent variables). SnapMMD relaxes these assumptions — allowing unknown and state-dependent volatility and supporting unobserved dimensions — thereby enabling inference in a broader class of scientifically relevant systems despite the inherent non-identifiability of the underlying dynamics. We will expand Appendix H to summarize the above discussion and clearly distinguish the fundamental limitations of the data regime from the modeling flexibility introduced by SnapMMD.
> > >
> > > Guan, V., Janssen, J., Lanzetti, N., Terpin, A., Schiebinger, G., & Robeva, E. (2025). Langevin SDEs have unique transient dynamics. arXiv preprint arXiv:2505.21770.
> > >
> > > Guan, V., Janssen, J., Rahmani, H., Warren, A., Zhang, S., Robeva, E., & Schiebinger, G. (2024). Identifying drift, diffusion, and causal structure from temporal snapshots. arXiv preprint arXiv:2410.22729.
> > >
> > > Lavenant, H., Zhang, S., Kim, Y. H., & Schiebinger, G. (2021). Towards a mathematical theory of trajectory inference. Annals of Applied Probability.

---

> ### Author Response · Authors · 2025-11-25
>
> **Metric and kernel dependence may bias training/evaluation.** We agree that kernel and bandwidth choices can influence MMD-based objectives; this is true for all kernel methods and is well understood in the literature. In early development, we experimented with multiple bandwidth strategies and found that the median heuristic computed jointly across all marginals provided the most stable and consistent performance across the diverse datasets we study. Because practitioners in scientific domains usually lack ground truth or tuning signals, our goal was to adopt a robust, simple default rather than require dataset-specific hyperparameter searches.
>
> The reviewer is correct that, in the Gulf of Mexico dataset, MMD prefers a more diffuse point cloud. This is precisely why we (1) also report Earth Mover’s Distance where it is available and (2) provide direct visualizations of predicted distributions. These complementary checks are intended to prevent over-reliance on any one metric and to help practitioners interpret model behavior when different distance measures emphasize different aspects of the data.
>
> More broadly, kernel sensitivity is not unique to our method; it is an inherent aspect of using kernels for distributional matching, and ML practice routinely involves selecting defaults among many reasonable alternatives. There are indeed “millions of choices,” but a paper must balance completeness with usability, and our intention is to provide a method that a practitioner can reasonably apply out of the box. A systematic exploration of the entire kernel-design space would be highly interesting future work but is beyond the scope of a single paper. We will clarify this motivation in the revision during the discussion period.
>
> **Simulation budget and sample-imbalance handling lack principled guidance.** We thank the reviewer for this comment and we are happy to clarify this aspect of our method. First, the number of simulated trajectories M does not influence the weighting across time points: we simulate a fixed number of paths rather than changing per marginal, so the simulation budget does not create any imbalance in the loss. The only weighting comes from the empirical sample sizes at each time point.
>
> Second, weighting by empirical sample size is a feature rather than a bug. An empirical distribution with very few samples is a much noisier proxy for its underlying population distribution. Giving such marginals proportionally less weight is the statistically natural approach, and indeed an advantage over existing SB methods, which treat all time points uniformly regardless of sampling reliability.
>
> Regarding the choice of M: the U-statistic estimator for MMD is well behaved once M is moderately large (e.g., a few hundred). In practice, we choose M large enough (in the thousands) so that the U-statistic provides a stable approximation. We will add a short guideline to the paper to make this explicit. More elaborate variance-reduction or adaptive-budget schemes might be possible, but systematically exploring all such design knobs is beyond the scope of a single paper — we focus on providing a method that is robust and straightforward to apply in practice.
>
> **Forecast horizon robustness.** We appreciate this question. As noted in our response to other reviewers, we agree that evaluating multi-step forecasting is a natural extension. We expect forecasting error to gradually increase with horizon length, especially once trajectories are pushed into regions of state space that were not covered by earlier snapshots, where all methods — including SB-based ones — must necessarily rely on model extrapolation rather than observed data. This limitation is inherent to the snapshot-only setting, since no approach observes true trajectories beyond the training horizon. We are currently running additional experiments to quantify this effect more systematically in our benchmarks and will share the results during the discussion period.

---

> > ### Author Response · Authors · 2025-11-25
> >
> > **Baseline coverage and parity are incomplete.** We believe there may be a misunderstanding about which baselines are applicable in the snapshot-only observation regime that this paper addresses. In our setting, each trajectory is observed at exactly one time point, and there are no repeated observations of the same particle over time. This is the defining property of the problem and is precisely what distinguishes it from standard time-series or neural-ODE settings.
> >
> > *Neural ODE variants.* Neural ODE and adjoint-based models require access to multiple points along the same trajectory in order to integrate the ODE forward and compute supervised objectives. They are therefore not compatible with the snapshot regime, where individual trajectories cannot be paired across times. Adapting Neural ODEs to this setting would require introducing a full latent‐trajectory inference layer, which is a different modeling problem. For this reason, we do not view Neural ODEs as meaningful baselines for the marginal-only forecasting task.
> >
> > *JKOnet and related potential-based methods.* We appreciate the suggestion. The JKOnet framework is designed for potential-driven transport and is primarily suited for interpolation between one pair of marginals. While the theory allows, in principle, certain extrapolations, the current implementation and experiments in the JKOnet paper operate under fixed train/test marginals and would require nontrivial modification to support our forecasting setups.
> >
> > Furthermore, we attempted an evaluation following JKOnet* (the follow-up paper to JKOnet by Terpin et al. 2024) on the Repressilator system and obtained unrealistic trajectories, likely due to the mismatch between potential-based modeling and the oscillatory feedback dynamics involved. This aligns with the known structural limitations of potential-driven approaches in non-gradient-flow dynamical systems.
> >
> > More broadly, our goal is to compare against methods that are designed for the same data regime — learning dynamics from independent marginal snapshots with no trajectory supervision. Within this domain, we include relevant SB and flow matching methods. We believe that expanding to trajectory-supervised or potential-only frameworks would not provide meaningful insight into the marginal-only forecasting problem that this paper tackles.
> >
> > Terpin, A., Lanzetti, N., Gadea, M., & Dörfler, F. (2024). Learning diffusion at lightspeed. Advances in Neural Information Processing Systems, 37, 6797-6832.
> >
> > **Compute scalability remains a practical concern.** We appreciate the reviewer’s attention to computational considerations. We agree that SnapMMD is not simulation-free. But for the class of problems we target, simulation-free training is generally not possible without imposing very strong structural assumptions. If one wishes to compare a model’s predicted marginals to observed data, one must know what the model predicts; for general unknown-volatility SDEs, the only practical way to obtain these predictions is to simulate trajectories. Simulation-free approaches exist only under special conditions (e.g., known forward SDE in diffusion models, Brownian reference in balanced SB), none of which hold in the scientific domains we address. Avoiding simulation in fully general SDEs typically shifts difficulty into optimization (e.g., in physics-informed neural networks), often offsetting any gains.
> >
> > Regarding scalability:
> > - The factorized MMD loss reduces to a quadratic form in batch size $N$, with potential reduction to $O(N)$ using kernel approximations (e.g., random Fourier features).
> > - MMD has far more favorable sample complexity in high dimensions than Wasserstein distances, making it practical for datasets such as PBMC (30 dimensions).
> > - Computational cost is linear in the number of time points due to kernel factorization.
> > - The dominant cost is differentiating through the SDE solver, which is comparable to other gradient-based simulation methods; in practice, training times for SnapMMD were comparable to SBIRR, the closest structural baseline.
> >
> > Wall-clock times depend strongly on implementation details, hardware, solver tolerances, and batch sizes; reporting absolute numbers in isolation would therefore be misleading. If the reviewer feels that reporting these raw numbers is essential, we are happy to time all experiments and include them.
> >
> > More broadly, we note that compute scalability is only one dimension of contribution in ML research. A single paper cannot explore all kernel approximations, simulators, and adjoint variants, but it can provide a method that meaningfully advances the Pareto frontier in a scientifically important regime, which is our goal here.

---

### Official Review · Reviewer_uKeE · 2025-10-31

**Soundness:** 3
**Presentation:** 3
**Contribution:** 2
**Rating:** 4
**Confidence:** 5

**Summary:**

This paper studies forecasting stochastic dynamics from population snapshots (unpaired observations at discrete times). The main idea is to learn SDE parameters by matching the *joint* state–time distribution between model and data via an RKHS MMD. With a time–separable kernel, the joint MMD decomposes into a weighted sum of per–time MMDs, yielding a least–squares style fit across times. The method, SnapMMD, jointly learns drift and *state–dependent* volatility and can embed mechanistic structure through parametric SDE families. An RKHS $R^2$ diagnostic is introduced using a barycenter baseline. Experiments on synthetic and real systems report strong forecasting and competitive interpolation.

**Strengths:**

* **(S1) Volatility learning.** Ability to estimate state–dependent diffusion is a practical advantage in scientific systems with heterogeneous noise.

* **(S2) Structural priors.** Parametric SDE families let users encode mechanistic knowledge; ablations show gains when structure is informative.

* **(S3) Diagnostics.** The MMD–based $R^2$ is simple and useful for model selection.

**Weaknesses:**

* **(W1) Simulation cost and trajectory of the field.** The method is not simulation–free and requires path simulation and adjoint backprop through an SDE solver. The paper acknowledges this as a primary limitation. Meanwhile, the literature in 2024–2025 has rapidly advanced simulation–free alternatives for snapshot matching and forecasting; relative to that trend, this work doubles down on a simulation–based line with high compute burden.

* **(W2) Data efficiency and compounding error.** Training subsamples unpaired points from each time marginal. Across times, this can compound sampling noise and induce temporal mismatch; performance may degrade unless $N$ is very large and the distribution heterogeneity is not increasing over time.

* **(W3) Model misspecification sensitivity.** Performance depends on the chosen SDE family; fully neural SDEs underperform, indicating reliance on good inductive bias.

* **(W4) Horizon robustness.** Forecasting beyond short horizons is not stress–tested; failure modes with increasing extrapolation time are unclear.

* **(W5) Kernel and weighting sensitivity.** Results and $R^2$ depend on $K_y$ and bandwidth; the time–kernel choice induces squared snapshot–frequency weights, which may over–emphasize heavy–sampled times.

* **(W6) Identifiability under partial observation.** Without an explicit measurement model, recovering latent coordinates and diffusion can be weakly identified.

**Questions:**

1. **Compute and scaling.** Please report wall–clock and asymptotic scaling in state dimension $d$, number of times $|\mathcal{T}|$, and samples $N$. Any variance–reduction or multi–trajectory reuse to lower gradient noise?

2. **Simulation–free baselines.** How does SnapMMD compare against recent simulation–free methods on the same datasets and metrics, especially in compute and sample efficiency?

3. **Subsampling effects.** Can you quantify the bias/variance impact of unpaired subsampling across times, and whether stratified or control–variate schemes reduce temporal mismatch?

4. **Long–horizon stress test.** Please evaluate error as the forecast horizon increases and characterize failure modes.

5. **Kernel and weighting robustness.** Sensitivity of results and $R^2$ to $K_y$, bandwidth, and to replacing $\delta(t-t')$ with a normalization that yields linear (rather than squared) time weights.

---

> ### Author Response · Authors · 2025-11-25
>
> We thank the reviewer for their feedback, comments and questions.
>
> We briefly restate our contributions and then address individual points. Forecasting stochastic dynamics from population-level snapshot data is a fundamental problem in many scientific domains, yet to our knowledge existing approaches are designed only for interpolation and offer no principled mechanism for extrapolating beyond the final observed time. Moreover, Schrödinger Bridge (SB)­-based methods require users to fix a volatility, often state independent, in advance, even though volatility is typically unknown and state-dependent, which severely limits forecasting accuracy. The difficulty stems from the SB formalizing the problem in path space while we never observe data in path space. Our method, SnapMMD, directly learns latent dynamics by matching the joint distribution of states and measurement times using a factored-kernel MMD loss. This formulation enables accurate forecasting, allows volatility to be learned (including state dependence), naturally incorporates domain knowledge, handles missing state dimensions, and provides interpretable diagnostics via an $R^2$-style metric.
>
> **Fair contribution.** We appreciate the reviewer’s perspective and would like to clarify what we view as a meaningful contribution in the context of a single machine-learning paper. A widely accepted goalpost is a combination of: (1) demonstrable empirical gains on a well-defined task or the ability to handle a setting that prior work cannot, and (2) a clear explanation of why the method works, so that readers can trust the mechanism beyond the necessarily limited set of experiments in any one paper. The expectation is not that a method dominates every baseline in every conceivable dimension, but that it advances the Pareto frontier for a task of interest and provides insight into how it does so.
>
> In our view, the present paper meets these criteria.
>
> (1) We study an important and previously unaddressed task: forecasting stochastic dynamics from snapshot (marginal) observations, and we evaluate it across diverse domains (genetics, ecology, oceanography, and real immune-activation data). Because existing SB methods do not natively handle forecasting, we constructed several natural extensions of these baselines to enable extrapolation. SnapMMD consistently outperforms them in forecasting accuracy, while additionally handling missing observational dimensions — something SB-based approaches cannot accommodate.
>
> (2) Mechanistically, our method’s advantages arise from two key differences in how users can encode structure compared with prior work:
> - (a) SnapMMD allows practitioners to specify more realistic mechanistic families than what SB-based approaches permit the user to impose. In particular, we can accommodate unknown and state-dependent volatility and latent or partially observed dimensions. Classical SB formulations require fixing the volatility a priori (state-independent, known), which restricts the kinds of domain knowledge that can be incorporated at the modeling stage — even though the resulting bridge solutions may themselves be flexible. SnapMMD therefore expands the range of scientifically meaningful inductive biases that can be expressed and learned from data.
> - (b) SnapMMD fits a mechanistic SDE family directly to the marginal (state–time) distributions, instead of optimizing a KL over path space. This means the learned object is an actual time-consistent dynamical model whose parameters are shared across all times. As a result, once fitted, the model can be naturally simulated forward, enabling coherent forecasting, variable horizons, and principled handling of missing observed coordinates.
>
> These contributions hold independently of whether the approach is simulation-free; the scientific value is in the improved capability, the broadened applicability, and the transparent mechanism by which SnapMMD achieves these gains.

---

> > ### Author Response · Authors · 2025-11-25
> >
> > **Simulation cost, trajectory of the field, and simulation–free baselines.** We appreciate the reviewer’s concern and would like to clarify the role of simulation in this setting. To compare a model of stochastic dynamics with observed data, one must be able to evaluate (or approximate) the model’s predictive distribution. For general SDEs with unknown drift and volatility, simulation is effectively the only practical way to obtain these predictions with generality. “Simulation-free’’ approaches typically rely on special analytic structure (e.g., known forward SDE in diffusion models, closed-form Schrödinger potentials under Brownian references, or balanced Schrödinger Bridges formulations) that does not hold in the scientific systems we target, where both drift and volatility may be nonlinear, state-dependent, or the states being partially unobserved.
> >
> > In problems without such structure, avoiding simulation might simply shift the difficulty elsewhere — often into highly stiff or unstable optimization (e.g., physics-informed neural networks attempting to avoid PDE solvers). In our experiments, direct simulation with adjoint gradients offered a stable and tractable solution across domains as diverse as genetics, ecology, and oceanography.
> >
> > If the reviewer had specific recent “simulation-free’’ methods in mind that apply to general, unknown-volatility SDEs under snapshot observations, we would be happy to add a targeted discussion; to our knowledge, current approaches rely on assumptions that do not hold in our setting.
> >
> > **Data efficiency, compounding error, and subsampling effects.** We respectfully note that we do not understand the concern raised here. In the snapshot setting, every observation is inherently unpaired across time — this is the core challenge of the problem, not a methodological choice. No method in this setting has access to paired samples across times — the data simply do not contain trajectory correspondences. We agree that some SB-based implementations construct artificial pairings internally (e.g., via particle coupling or non-killing schemes) as part of their optimization procedure. These pairings are algorithmic constructs, not data-level correspondences, and are used to build paths in methods that optimize in path space.
> >
> > In contrast, SnapMMD does not optimize over paths and therefore does not require any such pairing mechanism. Because there is no notion of true cross-time pairing in the data, the fact that different marginals may have different sample sizes is not problematic: we do not subsample, and no pairing is needed or lost. For this reason, we do not see how the reviewer’s concepts of “paired points,” “temporal mismatch,” or “compounding sampling noise across times” would arise beyond the standard minibatch variance present in any neural estimator.
> >
> > Moreover, because we are unsure what statistical quantity or mechanism the reviewer is referring to, we are not able to provide a meaningful technical response. If the reviewer can clarify the intended notion (e.g., a specific estimator or a precise definition of “paired”), we would be happy to address it, but as written the comment does not seem applicable to snapshot-based marginal matching.

---

> > > ### Author Response · Authors · 2025-11-25
> > >
> > > **Model misspecification sensitivity.** We respectfully disagree with the premise of this comment. The need to specify a model family is not a limitation of our method but a fundamental feature of the snapshot–observation setting. Because individual trajectories — and hence the temporal covariance structure — are never observed, drift and volatility are not identifiable in full generality. As we discuss in Appendix H (and as emphasized in recent work such as Guan et al. 2025), any method for learning SDEs from marginal data must introduce structure or restrict the model class. This is a property of the problem, not of SnapMMD.
> > >
> > > Our contribution is precisely to relax the restrictive assumptions imposed by existing Schrödinger bridge approaches. Classical SB methods require users to fix a single reference dynamic with known, state-independent volatility and cannot accommodate latent dimensions. Even SBIRR, which relaxes the reference to a family informed by domain knowledge, still requires the user to fix the volatility a priori; the noise structure is not learned and must be assumed correct. SnapMMD, in contrast, allows unknown and state-dependent volatility, supports unobserved dimensions, and enables semi-parametric drifts that incorporate partial mechanistic knowledge when available. Our fully neural SDE ablation in Sec. 4.6 underperforms for exactly the reason identifiability theory predicts: removing all inductive bias makes the problem dramatically harder. This does not indicate fragility; it confirms that incorporating appropriate structure is beneficial and expected in scientific applications.
> > >
> > > Finally, allowing users to specify a family of plausible reference dynamics is a practical advantage. Across our collaborations in oceanography, genetics, and ecology, domain experts consistently report that they are comfortable specifying a structured family (e.g., vortex-like flows in the Gulf of Mexico or chemical network models in systems biology) but not a single fixed reference drift. Even when such families are only approximate—e.g., constant-curl vortex flows are not globally valid in real fluid mechanics—they provide useful inductive bias at the relevant scale, as our experiments show. SnapMMD is designed to leverage precisely this level of prior knowledge.
> > >
> > > Guan, V., Janssen, J., Lanzetti, N., Terpin, A., Schiebinger, G., & Robeva, E. (2025). Langevin SDEs have unique transient dynamics. arXiv preprint arXiv:2505.21770.
> > >
> > > **Horizon robustness and long-horizon stress test.** We appreciate this point. As with any method extrapolating in time, forecasting accuracy will naturally degrade as the extrapolation horizon increases—particularly once trajectories move into regions of state space that were not populated by earlier snapshots, where prediction necessarily relies on the inductive bias of the chosen model family. This limitation is inherent to the problem setting, since no population-snapshot method observes true trajectories beyond the training horizon. Nevertheless, we are currently running additional tests to quantify multi-step behavior in our settings, and we will report these results during the discussion period.
> > >
> > > **Kernel and weighting sensitivity.** The heavier weighting on more–sampled time points is a feature, not a bug. In MMD-based marginal matching, the goal is to fit population distributions, and empirical marginals with very small sample sizes are much noisier estimates of their underlying populations. It is therefore desirable that the objective naturally downweights these high-variance snapshots. Over-emphasizing extremely undersampled times would in fact distort the estimator, not improve it.
> > >
> > > Regarding kernel choice, we agree that kernel families and bandwidths can influence MMD-based objectives. During early development, we explored several bandwidth strategies and found that the median heuristic computed jointly across all marginals provided stable and consistent performance across diverse datasets. Since practitioners in scientific applications typically lack ground-truth or tuning signals, our goal was to adopt a robust default rather than require dataset-specific searches. In practice, we did not observe pathological sensitivity once bandwidths were within a reasonable range. We can add a brief clarification of this rationale in the revision, though a full exploration of kernel-weighting design is beyond the scope of this work.

---

> > > > ### Author Response · Authors · 2025-11-25
> > > >
> > > > **Identifiability under partial observation.** We respectfully note again that this concern reflects a fundamental property of the snapshot-observation setting, not a limitation of our proposed method. When trajectories are never observed and some coordinates are unmeasured, latent states and diffusion coefficients are weakly identifiable for any approach unless very strong assumptions or explicit measurement models are introduced. Crucially, none of the reviewer’s other suggestions (e.g., simulation-free solvers, longer rollouts, alternative kernels) would alter this identifiability structure. Partial observation is a structural limitation of the data regime itself. The relevant scientific question, therefore, is not whether one can eliminate these constraints entirely, but whether a method can meaningfully advance the Pareto frontier within them. In our view, SnapMMD does so: it broadens the admissible reference families, handles unknown and state-dependent volatility, accommodates unobserved dimensions, and achieves substantially improved forecasting performance across diverse domains.
> > > >
> > > > As with any ML paper, one cannot address every open theoretical direction at once, but we believe the contributions here are scientifically meaningful and materially useful for practitioners.
> > > >
> > > > **Compute and scaling.** We thank the reviewer for the question. We provide here a breakdown of the computational structure of SnapMMD; we are adding a short summary of these points in the revision.
> > > >
> > > > *Computational form of the loss.* The MMD loss in Eq. (3) evaluates to a closed-form quadratic expression over sample pairs. For a minibatch of $N$ samples per time, the computational cost scales as $O(N^2)$. As is standard in MMD-based models, this can be reduced to $O(N)$ using low-rank kernel approximations or random Fourier features if desired, though we did not find this necessary in our experiments.
> > > >
> > > > *Scaling with dimensionality.* MMD remains computationally and statistically stable in moderate-to-high dimensional settings, especially compared to alternatives such as Wasserstein distances, whose sample complexity scales poorly. In practice, our kernels are applied component-wise and we did not observe dimensionality-related instability.
> > > >
> > > > *Scaling with a number of times.* Because the kernel factorizes in the time dimension, each marginal contributes additively to the objective. The overall cost scales linearly with the number of observed times $T$.
> > > >
> > > > *Gradients through the SDE solver.* The dominant cost is differentiating through stochastic simulations. We use stochastic adjoint methods where appropriate, falling back on automatic differentiation otherwise; training times were comparable to neural baseline methods used in the paper. We did not observe instability or excessive gradient noise that would necessitate explicit variance-reduction schemes.
> > > >
> > > > As wall-clock time depends strongly on hardware, implementation details, solver choice, and batch sizes, we believe that reporting absolute numbers would not be very meaningful. In practice, on our machines, training SnapMMD was comparable to SBIRR, which is the closest baseline in terms of computational structure. If the reviewer considers precise timing crucial, we are happy to re-run and report detailed measurements.
> > > >
> > > > That said, we also note that many of the reviewer’s other points (e.g., on identifiability, inductive bias, and horizon behavior) are scientifically more central to the claims of this paper. We have focused our rebuttal on addressing those core issues thoroughly, and we would welcome further clarification or discussion on any of them.

---

### Official Review · Reviewer_hPV4 · 2025-11-04

**Soundness:** 3
**Presentation:** 2
**Contribution:** 3
**Rating:** 6
**Confidence:** 3

**Summary:**

The paper proposes a method for trajectory inference of *both* the drift vector field *and* the state- and time-dependent diffusion coefficients for an Ito SDE which approximates the temporal evolution of marginal distributions observed via independent 'snapshot' samples.   While Schrodinger Bridge methods are customarily applied to these problems, the authors propose a conceptually different approach by (i) parameterizing the drift & diffusion coefficients of the stochastic dynamics, (ii) calculating the MMD loss between generated and observed samples, and (iii)  backpropagating through the parameterized stochastic dynamics using the stochastic adjoint method.    This approach allows for more flexible incorporation of prior knowledge of the dynamics via the chosen dynamics parameterization, and appears to be more amenable to forecasting beyond the temporal horizon of observed samples.

**Strengths:**

Recent methods for trajectory inference have perhaps over-indexed to Schrodinger Bridge (SB) methods given their mathematical connections with popular diffusion models.   While ingredients of the proposed method are not new in themselves, overall, the authors provide more refined approach to the particular problem at hand.

Backpropagation through (stochastic) dynamics (Li et. al 2020, "Scalable Gradients for SDEs") may not be the method of choice for large-scale generative modeling, but allows the authors to directly incorporate prior knowledge using *informed variational parameterizations* in trajectory inference.
- Previous methods may introduce 'prior knowledge' via the reference drift of the SB problem or a state-cost in optimal control problems (Generalized SB, Neklyudov et. al 2023, "Wasserstein Lagrangian Flows"), but these approaches are arguably less direct and natural than the proposed approach (as the authors argue).

The authors demonstrate empirical benefits from learning the diffusion coefficient, which is indeed underexplored in the literature and is naturally accommodated in the proposed approach.
- Authors may be interested in (concurrent) Guan et. al arxiv 05.2025 "Gradient-flow SDEs have unique transient population dynamics" discussing identifiability of diffusion coefficient inference.

The authors frame the exposition around forecasting or extrapolation beyond the observed samples, which is clearly an important problem not naturally handled by existing methods.   Results in Fig 1-3 show improved performance in this respect, although I have questions below.


I appreciate the extensive experimental results in the Appendix, and look forward to the authors having extra space to incorporate some of these in a camera-ready version.

**Weaknesses:**

The exposition of the paper is not sufficiently clear in several crucial ways.

*Parametric Components / `Variational Family of SDEs'*

> L241: "We pick the candidate conditional fθ (·|t) based on domain knowledge by specifying parametric components of the SDE in every experiment, reflecting the partial mechanistic understanding available in scientific settings."

I understand this point to reflect a `variational family of SDEs', where the drift and diffusion have learnable coefficients.   These details may be difficult to specify due to different parameterizations across settings, but this is a major point in the exposition and understanding of the method.   This needs to be clarified with concrete equations or an example in the main text.
- Are *all* experiments within known variational families of SDEs?
- In which cases do we expect the parameters to be time-dependent? (see Questions below)


*Missing dimensions*
> L79-80:  Our framework offers the added benefit of enabling robust interpolation and forecasting even with incomplete state measurements, as in the protein expression example above.

> L250-253: "More precisely, since our loss (Eq. (4)) is defined over the observed state variables (together with time), the model is trained to match the marginal distribution of the observed variables along with time, without making any additional assumptions or imputations for the missing dimensions.  (*note: duplicate / repeated clauses*)

This point was not at all clear to me until inferring the parameterization point above and inspecting App F.8.     From the current exposition, the reader might wonder (i) why the unobserved dimensions are necessary at all? or (ii) how do they receive supervision from the data?
- Having established the informed variational family of SDEs in the main text, this point can be explained by the fact that the dynamics of observed dimensions are coupled with (parameters, dynamics) of unobserved dimensions within the given family (?)




*Schrödinger Bridge Example and Focus*

I found the reference to App B1 in L138-141 to be distracting, and did not find the eventual argument in App. B1 to be convincing.
- It is clear that the (multimarginal) SB problem is underdetermined beyond the final marginal.
- As the authors point out, extrapolation via the reference is only one possible resolution.
- The authors use extrapolation of learned vector fields (with $t > t^{\text{observed}}_{\text{max}}$?) for some baselines, which is  most natural but can be easily argued to lack any concrete mechanism for matching true extrapolating dynamics
- with a learned reference within a known family (SBIRR, see below), the extrapolation via the reference seems reasonable

That said, I was left wondering **what are the actual mechanisms by which the proposed method achieves improved extrapolation?** (see below).

**Questions:**

*Mechanisms for Extrapolation*
> L75-76:  "Our framework allows data to guide extrapolation beyond observed time"
> L68-69:  "[proposed method] shifts modeling focus from interpolation to accurate... forecasting"

At first glance, I was quite confused by these statements given that
- The method is never trained at extrapolating times
- The $\delta_0(\mathbf{y})$ conditional in Eq 2 is not specified but also never appears to be used (?)


Instead, it seems *much more likely* that the forecasting extrapolation benefits are due to accurate parameter estimation within an appropriate choice of variational family.

- If the parameters within the given variational family of SDEs were time-independent, then we would expect to learn these parameters well with a reasonable amount of data.   These would naturally extrapolate by simply running the parameterized dynamics from $X_T$
- If the parameters were time-dependent but only slowly changing over time, we might still expect to get reasonable performance by simulating using parameter estimates at time $T$ forward in time.
- by contrast, the solution to SB problems is fundamentally time-dependent in most interesting cases, which should threaten extrapolation (especially combined with time-conditioned NN vector fields).

I'm tentatively positive on the paper, but with the remaining concerns above as the emphasis and performance on forecasting did not fully compute for me.

*SB Exposition and Baselines* (more speculative)

Given my previous points, I was also distracted by the use of Schrödinger Bridge in the title given that this is not an SB method.
- I think exposition comparing with the most similar SB-method (SBIRR?) would be useful, where the known family of SDEs is used to update the reference path measure and we optimize over the first argument with a more flexible family.
    - the authors could emphasize why this cannot capture learned volatility
- Would a not-quite-SB baseline be to simply swap out MMD for the MLE KL ($\mathbb{E}$ over data, $\log$ prob over parameterized `reference' SDE family)?
    - This essentially swaps out MMD for MLE KL, and would seemingly accommodate learning the volatility using a likelihood approximation (as in SBIRR) and backprop through time/dynamics (as in this work).  May require working in discrete time.
- Again, in this case, extrapolation using learned dynamics within an appropriate family seems quite reasonable.

If any of this makes sense, it might help to navigate the design space of algorithms and motivate the choices made.   Again, not crucial, this is just my personal random walk from SB => SBIRR => this method.


**Minor comments / questions** (no need to respond):


- Clarify domain of $g_0$ (vector/diagonal matrix?  more general?)  Why are we using 0-subscripts?

- Is it ok to use $\hat{f}(y|t)$ notation in Eq. 2?   It's used in $f_\theta(y|t)$ anyway, and L164 is hard to read.

- The conditional/joint notation appears to be inconsistent in App. B1, L727-747.   It seems like you mean $q_{t_T, t_{T+1}} = \pi_T p_{t_{T+1}|t_T}$ instead of  $q_{t_T, t_{T+1}} = \pi_T p_{t_T, t_{T+1}}$ (i.e. $q_{t_{i},t_{i+1}}$ for joint distribution on LHS, whereas $p_{t_i, t_{i+1}}$ is used for conditionals in stated notation and RHS).

---

> ### Author Response · Authors · 2025-11-25
>
> We would like to sincerely thank the reviewer for their thoughtful and detailed evaluation of our work. We deeply appreciate the reviewer’s careful reading of the paper, their recognition of the conceptual distinctions our approach brings relative to prior Schrödinger-bridge–based methods, and their insight into the role of diffusion learning, prior-knowledge parameterization, and forecasting capability in this problem setting. We are grateful for the constructive strengths highlighted, many of which reflect core motivations behind our approach.
>
> We first briefly restate our contributions and then address each of the reviewer’s questions and suggestions below. Forecasting stochastic dynamics from population-level snapshot data is a fundamental problem in many scientific domains, yet to our knowledge existing approaches are designed only for interpolation and offer no principled mechanism for extrapolating beyond the final observed time. Moreover, Schrödinger Bridge (SB)­-based methods require users to fix a volatility, often state independent, in advance, even though volatility is typically unknown and state-dependent, which severely limits forecasting accuracy. The difficulty stems from the SB formalizing the problem in path space while we never observe data in path space. Our method, SnapMMD, directly learns latent dynamics by matching the joint distribution of states and measurement times using a factored-kernel MMD loss. This formulation enables accurate forecasting, allows volatility to be learned (including state dependence), naturally incorporates domain knowledge, handles missing state dimensions, and provides interpretable diagnostics via an $R^2$-style metric.
>
> **Parametric Components / ``Variational Family of SDEs".**  We thank the reviewer for highlighting this point. We agree that clearer exposition of the variational family is valuable. In the revision, we will add a concrete example in the main text illustrating how the drift and diffusion are parameterized in practice (e.g., semi-parametric repressilator or vortex-style flow), so that readers can see explicitly how domain knowledge is incorporated.
>
> *Are all experiments within known variational families of SDEs?* To apply SnapMMD, one must indeed specify a model family (i.e., a variational family of SDEs) within which inference is performed. However, this does not mean we assume the true data-generating process lies within that family. In fact, several of our experiments are intentionally misspecified.
> - In the repressilator simulations, we use a semi-parametric family: a mechanistic production–decay scaffold with a neural activation function parameterizing gene interactions. This family does not match the data-generating dynamics exactly.
> - In PBMC, we do not know the true biological dynamics, and the semi-parametric family we use is almost certainly misspecified — the point is to provide biologically plausible inductive bias without assuming a correct model.
> - In the Gulf of Mexico experiment, the vortex family is a coarse approximation: a true ocean flow is not a perfect Lamb–Oseen vortex with simple divergence, but oceanographers view vortex-like structures as meaningful priors at this spatial scale.
>
> Thus, while each experiment uses a defined variational family, we do not assume well-specification. SnapMMD is designed to remain effective under such partial and imperfect prior knowledge, and our ablation studies (e.g., fully neural drifts in Sec. 4.6) explicitly show behavior under weaker structure.
>
> *In which cases do we expect the parameters to be time-dependent?* This is a very interesting question. In principle, many scientific systems exhibit time-dependent drift or diffusion:
> - Ocean flows vary with tides, weather patterns, and seasonal forcing. Volatility in this case can depend on temperature and in turn time dependent.
> - Ecological systems can have time-varying interaction strengths (e.g., by time of the year) or diffusion-like dispersal depending on environmental conditions.
>
> In the present paper, we focus on state-dependent (and state-dependent and unknown) drifts and diffusions, because these were the most relevant for the datasets we studied. We have not yet explored time-dependent parameterizations in our experiments, but we agree this is an interesting and meaningful direction for future work, and our framework is fully compatible with such extensions.

---

> ### Author Response · Authors · 2025-11-25
>
> **Missing dimensions.** We thank the reviewer for this thoughtful observation — their interpretation is spot on. In our framework, unobserved dimensions influence the observed ones through the coupling inherent in the chosen variational family of SDEs. Because the drift and diffusion of the observed coordinates are functions of the full state (observed + unobserved), learning to match the marginal distribution of the observed coordinates implicitly constrains the dynamics of the latent coordinates. In other words, the supervision enters indirectly through the coupling structure of the family: the latent variables are adjusted so that, when the coupled system evolves, the observed coordinates match the data. No additional assumptions, imputations, or priors on the latent variables are required beyond what is encoded in the family itself.
>
> We agree that the current exposition can make this point clearer. In the revision, we will (i) provide more details in the main text on the protein–mRNA model to illustrate how the observed and unobserved dimensions interact, and (ii) remove the duplicated clause and tighten the description around L250–253. This should clarify why the latent dimensions are present, how they influence the observed dynamics, and how they receive supervision through the MMD loss.
>
> **Schrödinger Bridge Example & Focus.** We appreciate the reviewer’s thoughtful question. We agree that App. B1 can be streamlined and are revising it accordingly so it does not distract from the main exposition.
>
> Regarding the mechanism behind improved extrapolation, the key point is:
>  SnapMMD learns a dynamical model — a drift and diffusion inside a chosen SDE family — using all observed marginals, and extrapolation is performed by simulating this learned model forward in time. Thus, extrapolation inherits the inductive bias encoded in the variational family (e.g., production–decay structure, coupling, vortex-like flow). When this structure captures meaningful aspects of the underlying system, it provides coherent behavior outside the observed time window.
>
> A useful way to think about this is the analogy with model-based forecasting vs. smoothing:
> - smoothing approaches interpolate well but have no intrinsic mechanism for long-range extrapolation,
> - whereas model-based forecasting extrapolates via a learned dynamical law, guided by structural assumptions.
>
> SnapMMD falls in the latter category. Its extrapolation performance reflects the fact that it fits a flexible but structured SDE model, rather than relying on interpolation alone. We will clarify this mechanism more explicitly in the main text.
>
> **Mechanisms for extrapolation.** We thank the reviewer for this insightful line of questioning — we completely agree with the core ideas expressed here.
>
> First, we fully agree that the model is not trained on extrapolation times; we do not have observations beyond the final marginal, so forecasting is necessarily a form of temporal extrapolation. Eq. (2) appears in Proposition 3.1, where $\hat{f}(y;t)$  is simply the conditional distribution of states given time $t$ induced by the learned SDE; we are clarifying this in the revision.
>
> Regarding the mechanism: we agree entirely with the reviewer’s interpretation that forecasting benefits arise from accurate parameter estimation within an appropriate variational family. SnapMMD learns a drift and diffusion belonging to a chosen SDE family using all observed marginals; the structural assumptions built into that family (e.g., production–decay laws, coupling between coordinates, vortex-like flow) provide the “shape” along which extrapolation proceeds. In other words, the inductive bias encoded in the family is what makes extrapolation coherent. This aligns directly with the reviewer’s two cases:
> - If the parameters are time-invariant, learning them accurately from observed marginals naturally yields reasonable forward simulation.
> - If the parameters are time-dependent but slowly varying, learning their values near the boundary still provides a plausible extrapolation direction.
>
> We also agree with the reviewer that Schrödinger-bridge solutions are generally time-dependent and, being fit for interpolation, provide no supervisory signal for times beyond the observed horizon. This makes it difficult for SB-based objectives to restrict the time-varying dynamics in a way conducive to forecasting. Indeed, SB-forward variants (e.g., Shen et al. 2024), which explicitly constrain bridges to use a fixed forward-time model family, perform meaningfully better — consistent with the reviewer’s reasoning that structural modeling drives extrapolation.
>
> More broadly, forecasting from snapshots is fundamentally underdetermined; any method must rely on structural assumptions to extrapolate meaningfully. Our contribution is to provide a framework where such structure can be encoded directly and transparently through the SDE family, with the MMD objective fitting that model in marginal space.

---

> > ### Author Response · Authors · 2025-11-25
> >
> > **SB Exposition and Baselines (more speculative).** We thank the reviewer for this very thoughtful and constructive line of speculation — this “random walk” through SB → SBIRR → our method is actually an excellent way to understand the design space, and we appreciate the suggestion.
> >
> > First, we agree that SnapMMD is not an SB method, and we will adjust the exposition so that the SB discussion is framed purely as conceptual context rather than as a methodological prerequisite. We also agree that, among SB-based approaches, SBIRR is the closest conceptual neighbor, and expanding the comparison with SBIRR in the main text would help readers situate our contributions. In particular, we will emphasize that SBIRR still requires a fixed, known constant volatility and therefore cannot capture state-dependent or learn unknown diffusion, which is a key ingredient in the systems we study.
> >
> > Regarding the reviewer’s suggestion of swapping MMD for a likelihood-style term:
> >  we agree this is an appealing and principled idea. Nothing in our framework fundamentally relies on MMD; MMD is used primarily for computational reasons (closed-form, simulation-based gradients, stability under missing dimensions). In principle, one could replace MMD with any divergence or discrepancy between the empirical marginals and the distributions induced by the parameterized SDE family so long as they can be estimated using samples — including KL, score matching, or other likelihood-based objectives.
> >
> > However, there are practical considerations:
> > - KL is often a tighter divergence than MMD and would not require bandwidth tuning.
> > - But KL estimation for high-dimensional continuous distributions requires density estimation (e.g., via score-based methods or flow models), which can become computationally expensive or unstable.
> > - Moreover, likelihoods for continuous-time SDEs typically require discretization and involve stochastic integrals or Girsanov-style approximations, which introduce their own numerical challenges.
> >
> > We agree with the reviewer that exploring this “near-SB, near-likelihood” regime — essentially combining ideas from SBIRR, likelihood-based SDE fitting, and the differentiable-simulation framework used here — is an interesting direction. We would be very excited to pursue this in future work, and will include a discussion in the paper, but this is not the current focus for this paper.
> >
> > **Minor comments.** We thank the reviewer for these careful and constructive observations; we are addressing all of them in the revision.
> > - *Domain and notation for $g_0$​.* In all experiments $g_0$​ is diagonal, but the framework allows a fully general diffusion matrix. We use the subscript “0’’ only to denote the true diffusion in synthetic experiments; we will clarify this in the text.
> > - *Notation $\hat f(y\mid t)$ in Eq. (2).* We agree that conditional–on–time notation is the cleanest formulation conceptually. Our current notation was motivated by the fact that we do not observe samples at arbitrary continuous $t$, but we agree this can be made clearer. We will update the notation around Eq. (2) to improve readability and avoid ambiguity.
> > - *Conditional/joint notation in App. B.1.* We appreciate the reviewer catching this. The reviewer’s reading is correct — the expression should use the joint distribution $q_{t_i,t_{i+1}}$​​ on the LHS, consistent with the conditional notation used later. We are fixing this inconsistency and revising the exposition to ensure the intended meaning is unambiguous.
> >
> > We thank the reviewer once again for their thoughtful engagement with our work.

---

### Official Review · Reviewer_SBGD · 2025-11-07

**Soundness:** 3
**Presentation:** 3
**Contribution:** 2
**Rating:** 4
**Confidence:** 4

**Summary:**

This work introduces SnapMMD, a framework for reconstructing underlying dynamics from snapshot data by directly fitting the joint distribution of state and time using an RKHS MMD loss with a factorized kernel. This kernel choice enables the decomposition of this MMD into a least-squares sum over time steps; hence, training can be effectively interpreted as a weighted sum of time-marginal MMDs between simulated and empirical snapshots. Experimentation on synthetic and real datasets exhibit strong one-step-ahead forecasting and competitive or better interpolation.

**Strengths:**

- Problem & motivation: This work tackles forecasting from snapshot data, which is a central and integral problem in a variety of disciplines.  The authors clearly explain their motivation and why current SB-based methods, although strong for interpolation, often struggle in extrapolation.
- Clarity: The paper is well structured, readable, and easy to follow.
- Novelty: The theoretical analysis in the paper is sufficiently thorough. This work treats snapshots as a joint (state, time) distribution and optimizes a kernel MMD loss, resulting in an original and novel formulation.
- Experimental evaluation: The authors present an extensive experimental evaluation of their algorithm with consistently strong results in both extrapolation and interpolation across tasks.

**Weaknesses:**

- Model-class dependence. The approach assumes a (semi-)parametric family for the dynamics; both performance and identifiability hinge on this choice. Clearer guidance or guarantees on when drift/volatility are recoverable—especially under misspecification—would strengthen the claims.
- Sensitivity to reference dynamics. Following the previous point, the fully non-parametric setting (Sec. 4.6) exhibits a marked performance drop, suggesting that the method’s success is tied to a strong reference dynamics class. An ablation on reference capacity/inductive bias (and its interaction with kernel choices) would clarify robustness.

**Questions:**

- Multi-step forecasting. It would be interesting to see how error accumulates over multiple future steps (e.g., 2, 3, etc)? Do the authors have any intuition if the method is still stable if attempting to forecast multiple steps, and what future adjustments could improve longer horizon stability?

- Baselines. In addition to the current baseline methods, is it possible to compare your methodology to algorithms specialized in time series performing forecasting and imputation, such as [1], [2]? Additionally, it would be interesting if you could show similar strong one-step extrapolation capabilities in time series datasets. For this, you could use the synthetic ones used in [2].

- Reference dynamics sensitivity. It was shown that the fully non-parametric variant (Sec. 4.6) underperforms. Is it possible for your method to leverage external neural representations? For example, is it possible to use a pretrained model from models that solve the multi-marginal SB or the mmFlow Matching problem (e.g., [3, 4, 5]) as the reference? Does this mitigate the dependence on a parametric reference class? This could be particularly useful in applications where the reference dynamics are not known.

- Compute & stability. What are the compute costs and stability characteristics for long rollouts (e.g., step size, simulator variance, number of paths per time) across datasets?

- Kernel sensitivity. Could you perform an ablation study to demonstrate how sensitive the results are to kernel family and bandwidth?

---
References

[1] FM-TS: flow matching for time series generation

[2] Provably Convergent Schr¨ odinger Bridge with Applications to Probabilistic Time Series Imputation

[3] Deep Momentum Multi-Marginal Schrödinger Bridge

[4] Multi-Marginal Stochastic Flow Matching for High-Dimensional Snapshot Data at Irregular Time Points

[5] Modeling complex system dynamics with flow matching across time and conditions.

---

> ### Author Response · Authors · 2025-11-25
>
> We thank the reviewer for their insightful feedback and thoughtful comments and questions. We are excited that the reviewer found our paper readable and easy to follow, the method original and novel, and the experimental evaluation extensive and strong in both extrapolation and interpolation.
>
> We briefly restate our contributions and then address individual points. Forecasting stochastic dynamics from population-level snapshot data is a fundamental problem in many scientific domains, yet to our knowledge existing approaches are designed only for interpolation and offer no principled mechanism for extrapolating beyond the final observed time. Moreover, Schrödinger Bridge (SB)­-based methods require users to fix a volatility, often state independent, in advance, even though volatility is typically unknown and state-dependent, which severely limits forecasting accuracy. The difficulty stems from the SB formalizing the problem in path space while we never observe data in path space. Our method, SnapMMD, directly learns latent dynamics by matching the joint distribution of states and measurement times using a factored-kernel MMD loss. This formulation enables accurate forecasting, allows volatility to be learned (including state dependence), naturally incorporates domain knowledge, handles missing state dimensions, and provides interpretable diagnostics via an $R^2$-style metric.
>
> **Model class dependence.** We thank the reviewer for raising this point. We fully agree that the choice of a model family matters — but this choice is inherent to the snapshot-observation setting rather than a limitation of SnapMMD. When only marginal snapshots are observed, individual trajectories (and therefore the temporal covariance structure) are never seen. As we discuss in Appendix H, no method can identify drift or diffusion uniquely without imposing structural assumptions. This is a fundamental limitation of the data regime, not of any particular algorithm. Recent theory (e.g., Guan et al., 2025; Lavenant et al., 2024) makes this point explicit even in gradient-flow SDEs under strong regularity assumptions.
>
> Where methods differ is in what structural assumptions they force the practitioner to make. SB approaches require users to fix a reference drift (or a family of reference drifts, in SBIRR) as well as to fix the volatility exactly (known, state-independent). In many scientific domains this latter assumption is unrealistic — volatility is typically unknown, often state-dependent, and may involve latent variables. SnapMMD, instead, allows unknown and potentially state-dependent volatility and accommodates latent or unobserved state dimensions, while still fitting a coherent mechanistic SDE model to all marginals.
>
> We also agree that in practice one cannot perfectly specify these families. This is precisely why we evaluate SnapMMD under misspecification. As shown in semiparametric repressilator and PBMC experiments, even partial biological scaffolds (e.g., production–decay structure) meaningfully improve learning compared to baselines that cannot represent the true volatility or integrate such contextual knowledge. Similarly, in the Gulf of Mexico, a vortex-inspired drift family — though not physically exact — provides a realistic inductive bias that clearly benefits forecasting in data-sparse regimes.
>
> Finally, our ablation with a fully neural SDE family shows degraded performance — consistent with the reviewer’s intuition and with identifiability theory. When the family is too unconstrained, many distinct drift/volatility pairs can explain the same marginals, and estimation becomes ill-posed. This illustrates the necessity of meaningful structural scaffolds in this data regime and reinforces the motivation behind SnapMMD’s design.
>
> Guan, V., Janssen, J., Lanzetti, N., Terpin, A., Schiebinger, G., & Robeva, E. (2025). Langevin SDEs have unique transient dynamics. arXiv preprint arXiv:2505.21770.
>
> Lavenant, H., Zhang, S., Kim, Y. H., & Schiebinger, G. (2021). Towards a mathematical theory of trajectory inference. Annals of Applied Probability.

---

> > ### Author Response · Authors · 2025-11-25
> >
> > **Sensitivity to reference dynamics.** We may be misunderstanding the intent of this comment, but we interpret it as asking whether SnapMMD’s performance relies critically on a strong inductive bias or specific reference dynamics. Our ablations in Sec. 4.6 were designed precisely to study this: we compare (i) domain-informed parametric models, (ii) semiparametric models with neural residuals, and (iii) fully neural drifts/diffusions. As expected, performance decreases as we deliberately remove useful mechanistic structure. We view this as a feature rather than a fragility, since scientific applications often can provide meaningful inductive bias that improves forecasting. If this was not the reviewer’s intended concern, we are very happy to clarify further or add the specific ablation the reviewer has in mind.
> >
> > **Multistep forecasting.** We appreciate this suggestion. We agree that multi-step forecasting is an interesting extension, and we expect performance to gradually degrade as the forecast horizon increases — especially once particles are pushed into regions of state space not covered by earlier snapshots, where extrapolation necessarily relies on the model family. This behavior is fundamental to any population-snapshot method, since no approach observes true trajectories beyond the training horizon. We are currently testing additional experiments to quantify this effect in our settings, and we will share these results during the discussion period.
> >
> > **Baselines.** We thank the reviewer for the suggestions. Our understanding is that [1] and [2] address a different problem setting: both assume access to multiple evaluations along each individual trajectory, enabling classical time-series forecasting and imputation. In contrast, the core challenge in our setting is that each particle is observed exactly once; there is no temporal continuity per particle, and the covariance structure between times is fundamentally unobserved. Methods designed for trajectory-level time series therefore do not operate under the same data limitations and are not directly applicable without substantial modification.
> >
> > Regarding the broader point about additional baselines, we note that — while there is always an open-ended universe of possible comparisons — the scientific question is whether an additional baseline provides new insight relative to the ones we already evaluate. Our baselines include state-of-the-art Schrödinger-bridge and flow-matching methods that are designed for population-snapshot dynamics; these are precisely the methods that fail at forecasting due to fixed-volatility or reference-drift constraints, which our approach overcomes. We are, of course, happy to incorporate a brief discussion clarifying why classical time-series models are not suitable in the snapshot regime.
> >
> > **Reference dynamics sensitivity.** We appreciate this thoughtful question. We interpret it as asking whether external neural models (e.g., pretrained multi-marginal SB or flow-matching models) could serve as a learned reference to reduce dependence on a parametric family. Conceptually, this is certainly possible: one could imagine a two-step pipeline where an external method first proposes a candidate family of dynamics, and SnapMMD then fits within that family. However, this does not eliminate the need for structural assumptions. As we discuss in Appendix H, with snapshot data the drift and volatility are not identifiable without restricting the model class; this is a property of the problem, not of our method. Any learned representation still encodes a reference dynamic — just implicitly. For example, DMSB uses Brownian motion as a reference on the lifted space of state and momentum, and flow-matching methods encode reference structure through their specific construction of the vector field. Our view is that SnapMMD provides a more transparent and controllable way to incorporate such a structure, rather than burying it inside a first-stage model. If an application provides a pretrained model that meaningfully captures domain-specific inductive bias, SnapMMD can certainly leverage it as a reference family; but this does not change the fundamental need for some reference dynamics due to identifiability constraints.

---

> > > ### Author Response · Authors · 2025-11-25
> > >
> > > **Compute & Stability.** We thank the reviewer for raising this point. In our current experiments, we do not perform long-horizon rollouts; all evaluations involve forecasting to a single held-out future time. For this setting, the relevant computational details are already provided in the paper: we briefly discuss the computational aspects of the MMD objective in Appendix D, and for each experiment we report the SDE solver settings — such as step size, integration length, and number of simulated trajectories — in the corresponding sections of Appendix F. Under these conditions (typically 200–400 simulated paths per time, fixed solver steps, and stochastic adjoint gradients), we did not observe numerical instabilities. It is not entirely clear to us what additional analysis the reviewer has in mind. Since our method does not currently target long-horizon simulation, we hesitate to speculate, but we would be happy to elaborate on any specific aspect the reviewer is interested in.
> > >
> > > **Kernel sensitivity.** We agree that kernel and bandwidth choices can influence MMD-based objectives. In early development, we experimented with several bandwidth selection strategies, and found that the median heuristic computed across all marginals provided stable and consistent performance across the diverse datasets in our paper. Since practitioners in real scientific applications typically lack ground truth or tuning signals, our goal was to adopt a robust default rather than require dataset-specific bandwidth search. In practice, we did not observe pathological sensitivity once the bandwidth was within a reasonable range. We are adding a brief note in the revision clarifying this design choice, though we emphasize that exploring the full kernel-design landscape is beyond the scope of this paper.

---

### Meta-Review · Area_Chair_7Rs4 · 2026-01-07

**Summary:**

This work considers the problem of reconstructing and forecasting dynamics from unpaired snapshots of some latent stochastic dynamics. This is an important problem and actively considered by the ICLR community. The authors' approach is to directly fit the joint distribution of state and time using an RKHS MMD loss with a factorized kernel. This kernel choice enables the decomposition of this MMD into a least-squares sum over time steps, and training can therefore be effectively interpreted as a weighted sum of time-marginal MMDs between simulated and empirical snapshots. The effectiveness of the method was demonstrated experimentally on synthetic and real datasets. Reviewers and I agree this is an interesting idea. Meanwhile, reviewers also raised significant concerns about technical details - for example, a common question is about the model class dependence as the approach is semi-parametric; unfortunately, these concerns seemed to have not fully resolved by the end of the discussion period. In addition, it was suggested that `past SB papers have not addressed forecasting', but experimental demonstrations were restricted to comparisons to methods only from a single research group, hence insufficiently supporting the claim. Recognizing the potential of the idea, I encourage the authors to take the discussions into consideration and re-submit a revised version.

**Reviewer Concerns:**

Just a guess: they probably won't be fully convinced.

**Reviewer Scores:**

Just a guess: they probably won't increase by too much.

---

### Decision · Program_Chairs · 2026-01-26

Reject